# Rare genetic variants confer a high risk of ADHD and implicate neuronal biology

Ditte Demontis[1,2,3,4,28 ✉], Jinjie Duan[1,2,3,28], Yu-Han H. Hsu[4,5], Greta Pintacuda[4,5], Jakob Grove[1,2,3], Trine Tollerup Nielsen[1,2,3], Janne Thirstrup[1,2,3], Makayla Martorana[4,5], Travis Botts[4,5], F. Kyle Satterstrom[5,6], Jonas Bybjerg-Grauholm[2,7], Jason H. Y. Tsai[1,2,3], Simon Glerup[1], Martine Hoogman[8,9,10], Jan Buitelaar[8,9], Marieke Klein[8,9], Georg C. Ziegler[11], Christian Jacob[12], Oliver Grimm[13], Maximilian Bayas[13], Nene F. Kobayashi[13], Sarah Kittel-Schneider[14,15], Klaus-Peter Lesch[16,17,18], Barbara Franke[8,9,19], Andreas Reif[13,20], Esben Agerbo[2,21,22], Thomas Werge[2,23], Merete Nordentoft[2,24], Ole Mors[2,25], Preben Bo Mortensen[2,21,22], Kasper Lage[4,5,23], Mark J. Daly[5,6,26,27], Benjamin M. Neale[5,6] & Anders D. Børglum[1,2,3 ✉]

Attention deficit hyperactivity disorder (ADHD) is a childhood-onset neurodevelopmental disorder with a large genetic component[1]. It affects around 5% of children and 2.5% of adults[2], and is associated with several severe outcomes[3–11]. Common genetic variants associated with the disorder have been identified[12,13], but the role of rare variants in ADHD is mostly unknown. Here, by analysing rare coding variants in exome-sequencing data from 8,895 individuals with ADHD and 53,780 control individuals, we identify three genes (*MAP1A*, *ANO8* and *ANK2*; $P < 3.07 \times 10^{-6}$; odds ratios 5.55–15.13) that are implicated in ADHD. The protein–protein interaction networks of these three genes were enriched for rare-variant risk genes of other neurodevelopmental disorders, and for genes involved in cytoskeleton organization, synapse function and RNA processing. Top associated rare-variant risk genes showed increased expression across pre- and postnatal brain developmental stages and in several neuronal cell types, including GABAergic (γ-aminobutyric-acid-producing) and dopaminergic neurons. Deleterious variants were associated with lower socioeconomic status and lower levels of education in individuals with ADHD, and a decrease of 2.25 intelligence quotient (IQ) points per rare deleterious variant in a sample of adults with ADHD ($n = 962$). Individuals with ADHD and intellectual disability showed an increased load of rare variants overall, whereas other psychiatric comorbidities had an increased load only for specific gene sets associated with those comorbidities. This suggests that psychiatric comorbidity in ADHD is driven mainly by rare variants in specific genes, rather than by a general increased load across constrained genes.

Attention deficit hyperactivity disorder (ADHD) is a neurodevelopmental disorder that affects around 5% of children and 2.5% of adults worldwide[2]. The disorder is linked to a variety of serious outcomes, including higher risks of substance-use disorder[3,4], accidents[5], premature death[6], unemployment[7], incarceration and crime[8], suicide[9] and metabolic conditions[10,11]. Gaining insight into the biological mechanisms that drive the disorder is crucial for understanding how it develops and how it could be treated in the future.

A large proportion of ADHD risk can be explained by genetics, with an estimated twin heritability of 77–88%[1]. Large genome-wide association studies (GWASs) have found that common genetic variants explain 14–22% of the overall variation in liability[12,13]. The most recent GWAS of ADHD identified 27 genome-wide significant loci and estimated that around 7,300 common variants explain 90% of the single-nucleotide polymorphism (SNP) heritability of ADHD[13]. ADHD is thus highly polygenic, with a considerable proportion of risk explained by common

genetic variation; however, an investigation of rare variants is also necessary to explain more of the heritability. We have previously established that rare deleterious variants in evolutionary constrained genes have a role in ADHD, at a level comparable with what has been found in autism[14].

Although rare coding variants might explain only a minor part of the overall liability, they can confer substantial risk individually and, in contrast to common variants, they often directly pinpoint the causal gene affected and the probable functional consequence, providing clues to the underlying aetiology of ADHD.

Here we present results from a whole-exome-sequencing study of ADHD and identify three significant genes with an increased load of rare deleterious variants in individuals with ADHD, compared with control individuals. We provide insights into the genetic architecture and neurobiological mechanisms involved in ADHD, by linking identified rare-variant risk genes to gene-expression data from brain tissues

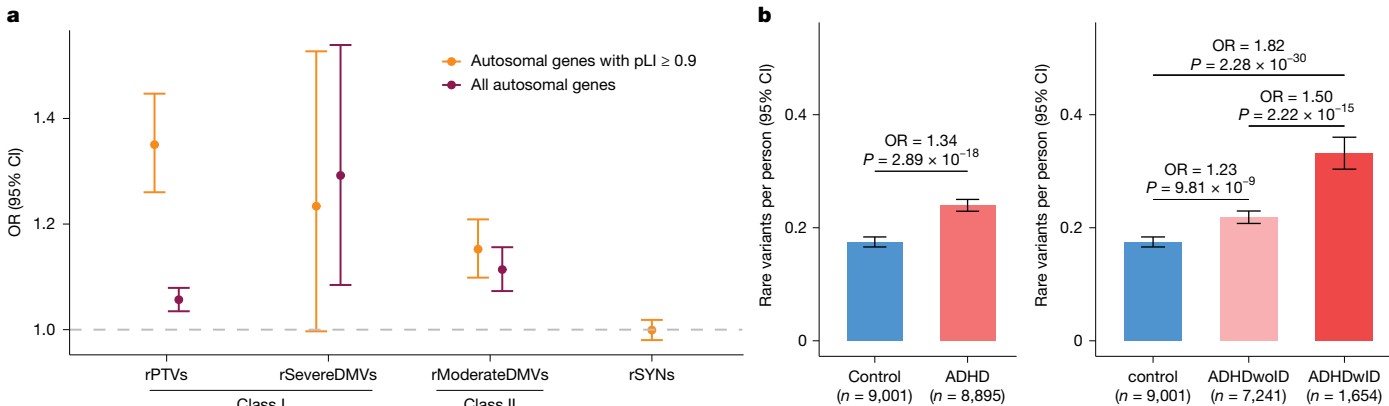

**Fig. 1 | ADHD risk across rare-variant categories and mean load. a**, Odds ratio (OR) of rPTVs, rSevereDMVs, rModerateDMVs and rare synonymous variants (rSYNs) in all genes (marked in yellow) and in constrained genes (pLI ≥ 0.9; marked in red) in individuals with ADHD (n = 8,895) and in control individuals (n = 9,001). Dots represent OR point estimates, and error bars indicate the corresponding 95% confidence intervals (CIs). Owing to the similar effect sizes of rPTVs and rSevereDMVs in constrained genes (pLI ≥ 0.9) these variants were grouped into class I variants, and rModerateDMVs were

categorized as class II variants. Note: the count of rSYNs in each individual is used as a covariate in the analyses, and thus it is not possible to test for differences in rSYN load across all autosomal genes. **b**, Number of class I variants (rPTVs + rSevereDMVs) in constrained genes (pLI ≥ 0.9) per person on average. ADHD, individuals diagnosed with ADHD regardless of any comorbidities; ADHDwoID, individuals diagnosed with ADHD but not ID; ADHDwID, individuals diagnosed with both ADHD and ID. ORs and two-sided P values were calculated using logistic regression in **a**,**b**.

and cell types, and through analyses of protein–protein interaction (PPI) networks of the identified rare-variant risk genes. We evaluate the load of rare deleterious variants across various comorbidities, and show that rare deleterious variants affect socioeconomic status (SES) and cognition in individuals with ADHD.

## Sequencing of individuals from the iPSYCH cohort

We analysed whole-exome-sequencing data from 8,895 individuals with ADHD and 9,001 control individuals from the Danish iPSYCH[15,16] (Methods, Supplementary Table 1 and Extended Data Fig. 1), doubling the sample size from our previous study[14].

We focused on rare variants with an allele count no higher than five across iPSYCH (17,896 individuals; Supplementary Figs. 1 and 2) and a subset of individuals with European (non-Finnish) ancestry from the Genome Aggregation Database (gnomAD) who had not been diagnosed with a psychiatric disorder[17] (44,779 individuals). There was a high comorbidity with intellectual disability (ID) among the included individuals (18.4%; Supplementary Table 1) and, consequently, the effect of co-occurring ID was evaluated by doing analyses both with and without comorbid ID.

## Effects across functional categories

Rare variants were grouped on the basis of their functional effect on the encoded protein, and their load in ADHD compared with that in control individuals was assessed for all autosomal genes (18,866 genes) and autosomal genes with a probability of being loss-of-function intolerant (pLI ≥ 0.9) (2,811 genes)[18], hereafter referred to as constrained genes. We found a significantly increased burden of rare protein-truncating variants (rPTVs) in ADHD compared with control individuals in all genes (odds ratio (OR) = 1.06, 95% confidence interval (CI) = [1.04, 1.08], P = 2.41 × 10^−7), and a further increased load in constrained genes (OR = 1.35, CI = [1.26, 1.45], P = 1.52 × 10^−17; Fig. 1a and Supplementary Table 2). In line with observations in schizophrenia[19], the latter effect size was similar to what was observed for rare severe damaging missense variants (rSevereDMVs; defined as variants with a missense badness, PolyPhen-2 and constraint (MPC)[20] score > 3) in all genes (OR = 1.29, CI = [1.09, 1.54], P = 4.11 × 10^−3; Fig. 1a and Supplementary Table 2). Consequently, rPTVs and rSevereDMVs were grouped together (referred to as class I variants) in the gene-discovery analysis. The burden of rare

missense variants predicted to have a moderate effect on protein function (rModerateDMVs; 2 ≤ MPC score ≤ 3) was significantly increased in ADHD, but with a lower effect size (OR_all genes = 1.11, CI = [1.07, 1.16], P = 1.43 × 10^−8; Fig. 1a and Supplementary Table 2) than was observed for class I variants; these were therefore analysed separately (referred to as class II variants). For comparison, there was no increased load of rare synonymous variants in ADHD in constrained genes (Fig. 1a).

Class I variants in constrained genes were identified in around one out of five individuals with ADHD (Fig. 1b, Supplementary Table 3 and Supplementary Fig. 3), indicating that highly deleterious variants did not contribute to disease risk in most individuals with ADHD.

No differences in variant load were observed between male and female individuals with ADHD (Supplementary Fig. 4), suggesting that there is a similar overall burden of rare variants in the two sexes, in line with what is observed for common variants[21,22].

## ADHD gene discovery

To increase the power to identify rare-variant risk genes for ADHD, the control group was expanded by combining iPSYCH controls with 44,779 individuals from gnomAD[17]. In total, we analysed 8,895 individuals with ADHD and 53,780 control individuals, assessing only genomic regions with high-quality data across iPSYCH and gnomAD samples. To ensure that a potential signal of rare deleterious variants in ADHD was not driven by a generally higher rate of variants in iPSYCH samples, we only included genes that had a higher rate of rare synonymous variants in control individuals than in ADHD (15,603 genes analysed; 3,263 genes excluded).

We performed a gene-based burden test to identify genes with an increased burden of class I or class II variants using a two-tailed Fisher's exact test. Because we focused on variants with a deleterious effect on protein function, we expected these to be increased in ADHD compared with controls. Therefore, and owing to our variant filtering strategy (Methods), the analysis was restricted to include only genes with a higher rate of class I variants (3,698 out of 15,603 genes) or class II variants (1,026 out of 15,603) in ADHD than in controls. For genes with both class I and class II variants (347 genes), a higher number of both types of variants in ADHD compared to controls was required, and the combined effect was estimated in a meta-analysis (Methods).

We identified three significant genes: *MAP1A* (P = 1.02 × 10^−6, OR = 13.31), *ANO8* (P = 1.90 × 10^−6, OR = 15.31) and *ANK2* (P = 2.72 × 10^−6, OR = 5.55) (Fig. 2a and Supplementary Table 4). The results for *MAP1A*

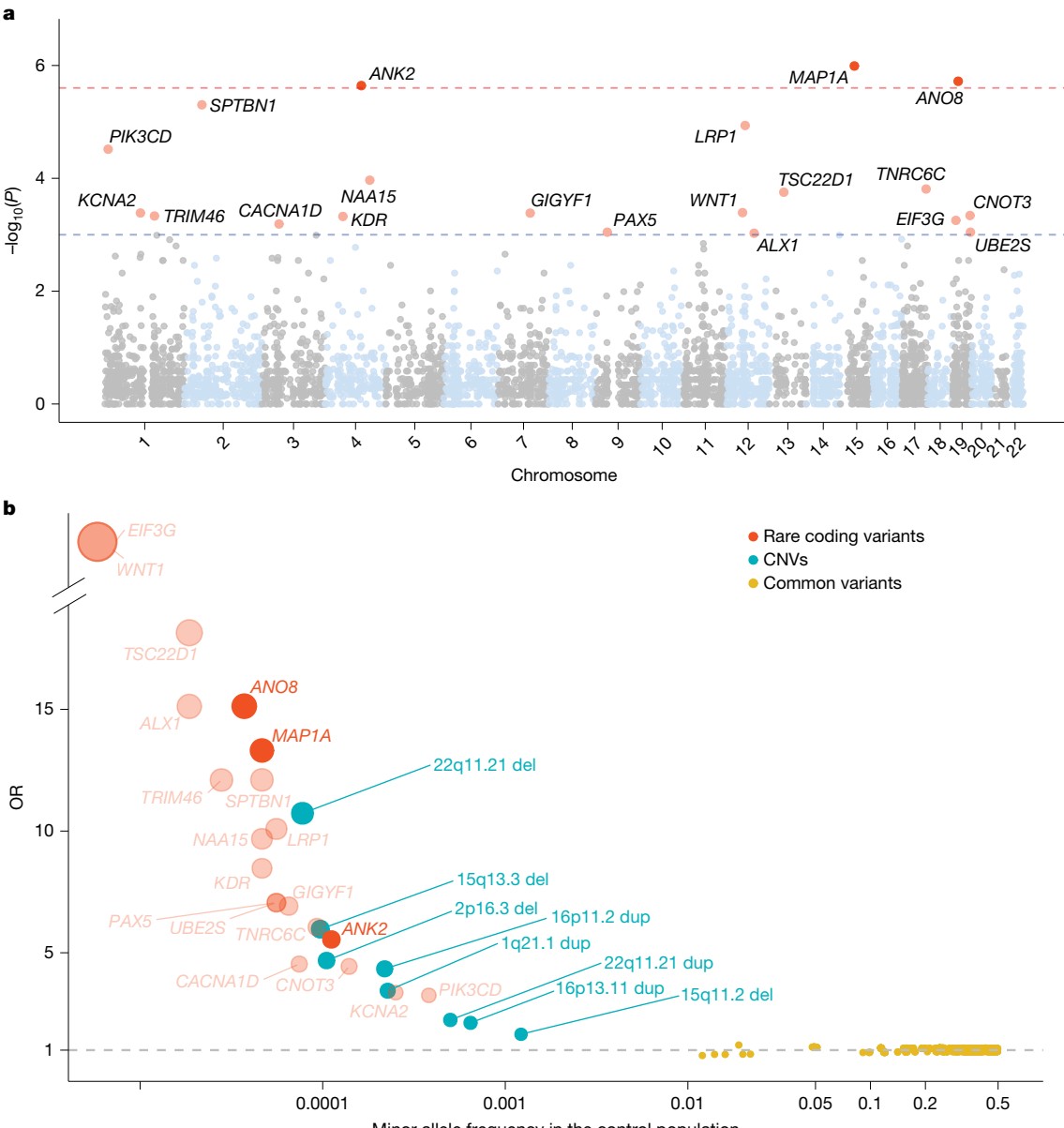

**Fig. 2 | Association of rare coding variants with ADHD. a**, Manhattan plot with $-\log_{10}(P)$ on the $y$ axis for gene-based associations from a two-sided Fisher's exact test of counts of rare class I and II variants in 8,895 individuals with ADHD and 53,780 controls. The chromosome position of genes is shown on the $x$ axis. Genes that met the threshold for exome-wide significance (two-sided $P < 3.07 \times 10^{-6}$, implying significance correcting for 16,297 tests) are highlighted in dark red; genes with two-sided $P < 0.001$ are in orange. The red dashed line denotes exome-wide significance; the blue dashed line represents two-sided $P \le 0.001$. **b**, Genetic architecture plot. Dots represents ORs from this study and the most

recent GWAS[13] and CNV study[25] of ADHD. The minor allele frequency in the control cohorts is shown on the $x$ axis. Dot colour represents variant type (rare, common or CNV) and dot size reflects the strengths of the associations. The 20 genes from this study with a gene-based two-sided burden $P < 0.001$ are shown in red; the exome-wide-significant genes are solid red and the remainder are translucent red. Two genes (*WNT1* and *EIF3G*) have an infinite OR, as rare deleterious variants were only observed in individuals with ADHD and none in control individuals.

and *ANO8* were driven entirely by class I variants (only rPTVs); for *ANK2*, the result stemmed from both class I and class II variants (rPTVs and rModerateDMVs). Details on the phenotypes of individuals with class I or class II variants in the three risk genes can be found in the Supplementary Information and Supplementary Fig. 5. The ORs implied that rare deleterious variants in these genes confer a risk that is much higher than that observed for common variants[12,13], and higher than the risks observed for copy-number variants (CNVs) in iPSYCH[23,24] and other studies[25] (Fig. 2b). Out of the top 20 genes ($P < 1 \times 10^{-3}$; Supplementary Table 4), 16 are constrained.

We examined the generalizability of our findings in another European sample, consisting of 1,078 individuals who had been clinically

diagnosed with persistent ADHD and 1,738 controls. Overall, class I variants were significantly enriched in ADHD, with further enrichment when restricting to constrained genes (OR = 1.24, CI = [1.07, 1.45], $P = 0.005$; Supplementary Fig. 6 and Supplementary Table 5). Owing to the small sample size, we had no power to discover significant genes, but the effect-size point estimate was higher for the top gene set from our discovery analysis than for constrained genes (OR = 1.42, CI = [1.08, 1.85], $P = 0.012$; Supplementary Table 6; see also Supplementary Information), suggesting that the identified top genes overall confer more ADHD risk than do constrained genes. It is noteworthy that the number of deleterious variants in *MAP1A* and *ANK2* was higher in individuals with ADHD than it was in

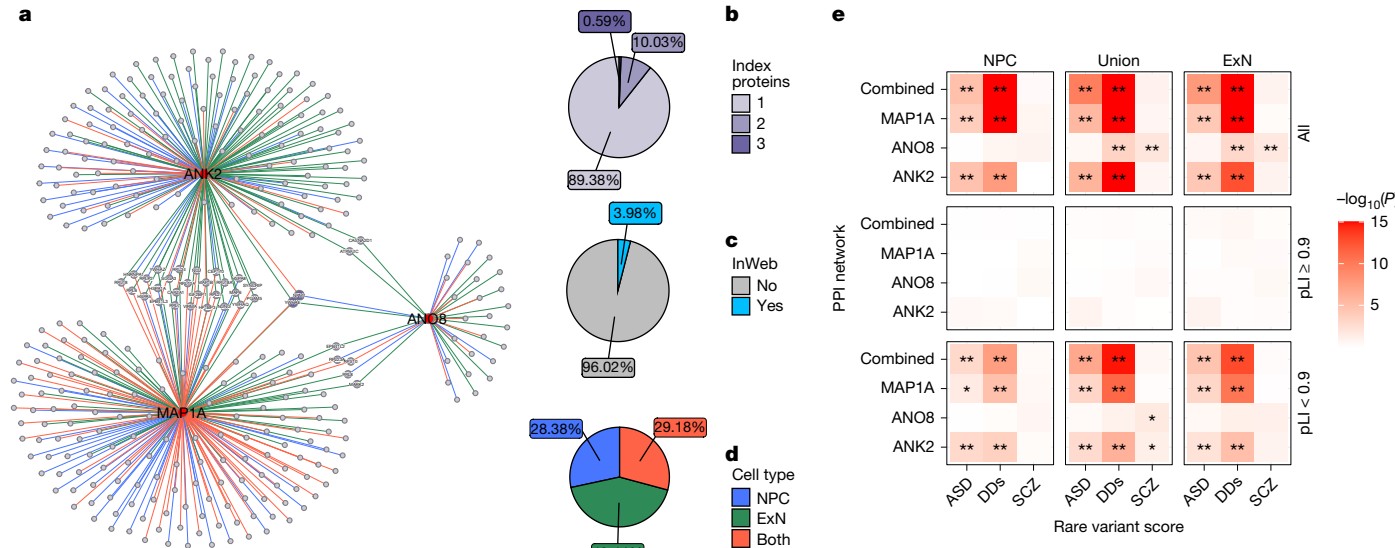

**Fig. 3 | Enrichment of rare-variant risk genes in PPI networks. a**, PPI network derived from three IP–MS experiments. Nodes represent index proteins (MAP1A, ANK2 and ANO8) and their interactors (purple); colour intensity and size of the interactor nodes scale with interactor frequency (that is, number of linked index proteins). Line colour indicates cell type: NPCs in blue, ExNs in green and both cell types in orange. **b**, Distribution of interactor frequency in the network. **c**, Distribution of InWeb versus newly reported interactions in the network.

**d**, Distribution of interactions across cell types. **e**, Results from one-tailed Kolmogorov–Smirnov tests of enrichment in the PPI networks of rare-variant risk genes associated with autism (ASD)[28], developmental disorders (DDs)[28] or schizophrenia (SCZ)[19]. *Kolmogorov–Smirnov test one-sided $P < 0.05$, **FDR $< 0.05$, minimum $P$ was capped at $1 \times 10^{-15}$ for visualization purposes. Results for all genes with available pLI score[48] (all), constrained genes (pLI $\geq 0.9$) and non-constrained genes (pLI $< 0.9$) are shown separately.

control individuals (no rare deleterious variants observed in *ANO8*; Supplementary Table 7).

## X-chromosome analyses

Significant differences in the load of deleterious variants on the X chromosome were observed only for class II variants in ADHD with ID (Extended Data Fig. 2a–d and Supplementary Table 8), compared with controls, and in the sex-stratified analysis comparing male individuals with ADHD to control individuals (Supplementary Fig. 7), but not after excluding comorbid ID. No genes on the X chromosome were associated with ADHD after Bonferroni correction (Supplementary Table 9).

## Rare burden heritability

The variability in the phenotype explained by rare variants revealed a burden heritability of 2.5% (s.e. = 0.7%) for class I variants and 0.1% (s.e. = 0.3%) for class II variants for ADHD on the liability scale, using a population prevalence of 5% (Supplementary Table 10). When excluding comorbid ID, the burden heritability decreased to 1.43% (s.e. = 0.74%) and 0.26% (s.e. = 0.27%) for class I and class II variants, respectively. These estimates are in line with findings for schizophrenia (1.7% (s.e. = 0.3%)) and bipolar disorder (1.8% (s.e. = 0.3%))[26]. Rare synonymous variants showed no evidence of non-zero burden heritability for ADHD. The three significant genes (*MAP1A*, *ANO8* and *ANK2*) explained 5.2% (s.e. = 3.4%) of the class I burden heritability, suggesting that other ADHD risk genes implicated by rare coding variants remain to be identified.

## Linking ADHD risk genes to biology

The three identified risk genes might point to a larger set of genes and biological mechanisms involved in ADHD through their protein inter-action partners, as reported in other disorders[27]. To examine this, we performed immunoprecipitation–mass spectrometry (IP–MS) for proteins encoded by the three genes (hereafter referred to as index

proteins) in human induced pluripotent stem (iPS)-cell-derived neural progenitor cells (NPCs) and excitatory neurons (ExNs) to generate their PPI networks (Supplementary PPI Tables 1–5). Across the two cell types, we identified 184, 35 and 158 interaction partners for MAP1A, ANO8 and ANK2, respectively; 36 were linked to more than one index protein and thus could point to convergent biology (Fig. 3a–d). Of the interacting proteins, 48 have previously been implicated in neurode-velopmental disorders by genetic studies[13,19,28] (Supplementary PPI Table 6). Furthermore, the MAP1A-, ANK2- and combined network of all three index proteins were significantly enriched (false discovery rate (FDR) < 0.05, one-tailed Kolmogorov–Smirnov tests) for rare-variant risk genes associated with autism spectrum disorder (ASD) and devel-opmental disorders (DDs) in both NPCs and ExNs, compared with other protein-coding genes expressed in the neuronal cell model (Fig. 3e and Supplementary PPI Tables 7 and 8). The ANO8 network was enriched for rare-variant risk genes associated with DD and with schizophrenia in ExNs. The networks were not enriched for ADHD or other neurode-velopmental risk genes identified by common variants (Supplementary PPI Table 9).

In addition, most networks were strongly enriched for proteins encoded by constrained genes (Supplementary PPI Table 8). The constrained network genes were generally not further enriched for disease risks, compared with other constrained genes, whereas the non-constrained genes in some networks showed stronger enrichment, compared with the rest of the non-constrained genome (Fig. 3e and Supplementary PPI Table 10).

For each of the three index proteins, the union (NPCs + ExNs) PPI-network genes were significantly enriched among several gene sets. For MAP1A, top findings included genes encoding RNA binding ($P = 2.32 \times 10^{-99}$) and cytoplasmic ribosomal proteins ($P = 2.09 \times 10^{-85}$); for ANO8, genes expressed in cell junctions ($P = 2.23 \times 10^{-11}$) and synapses ($P = 7.09 \times 10^{-8}$); and for ANK2, genes encoding the actin cytoskeleton ($P = 2.77 \times 10^{-39}$) and cell junction proteins ($P = 1.17 \times 10^{-30}$) (Supplementary Table 11). For all three PPI networks, a high proportion of the genes mapped to genes with synaptic annotations in SynGO[29] (MAP1A 52.7%; ANK2 44.30%; and ANO8 57.14%) and showed significant

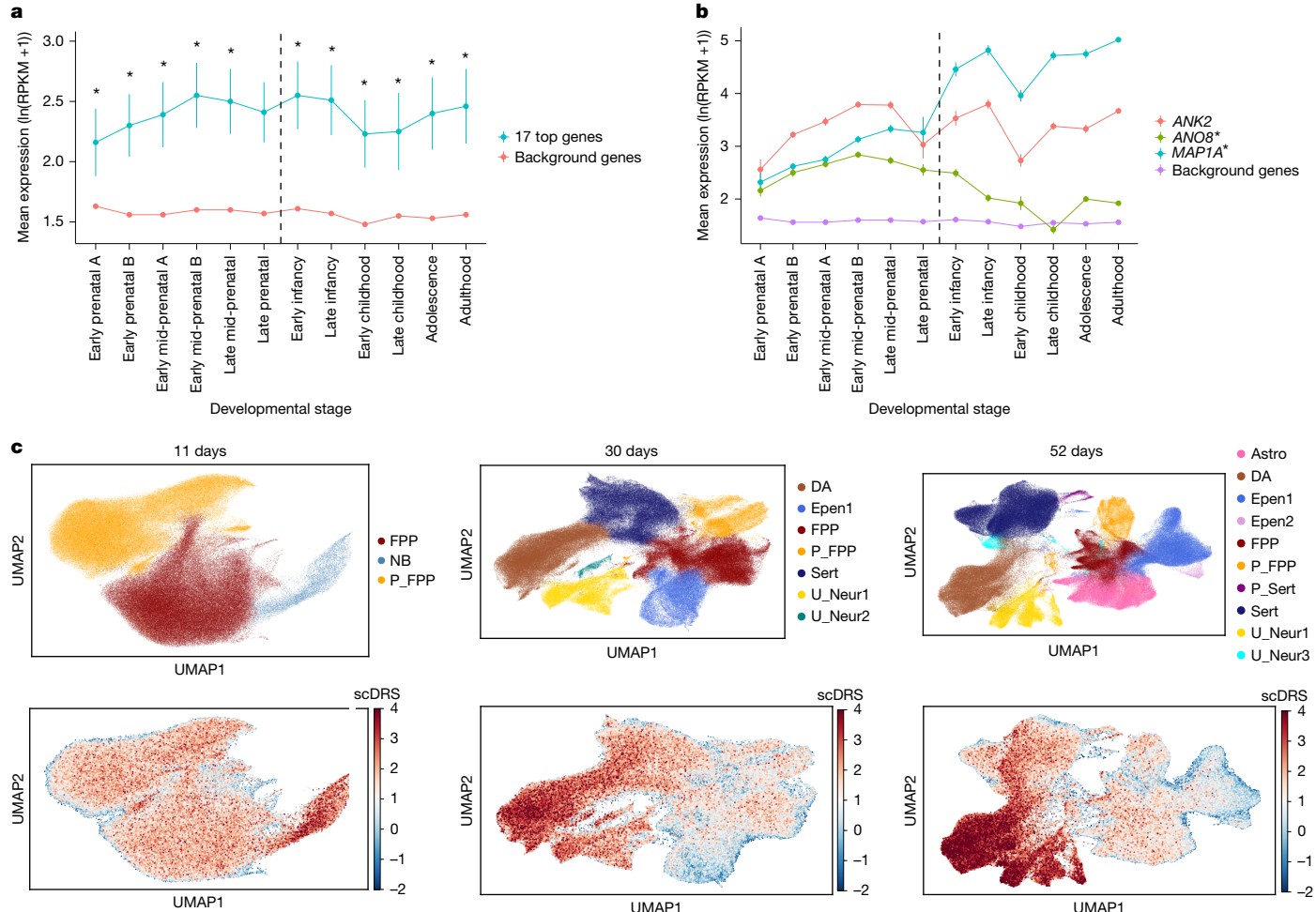

**Fig. 4 | Expression of ADHD risk genes across brain developmental stages and cell types. a**, Mean expression, expressed as ln(reads per kilobase million (RPKM) + 1)), of the 17 ADHD risk genes and background genes across neocortical brain developmental stages in BrainSpan. Background genes (22,402 genes or transcripts) include genes expressed in BrainSpan, except for the 17 ADHD risk genes. Asterisks indicate a significant difference between the two gene sets at a given developmental stage using a two-sided paired *t*-test or Wilcoxon rank test; $P = 4.17 \times 10^{-3}$ was considered significant, correcting for 12 brain developmental stages. Vertical lines represent standard error. **b**, Expression of the three significant genes across brain developmental stages, and expression of background genes (22,416 genes or transcripts). Asterisks next to the gene names at the right side indicate significant differences in mean prenatal and postnatal expression using a two-sided two-sample *t*-test; $P = 2.2 \times 10^{-16}$ (*MAP1A*) and $P = 2.2 \times 10^{-16}$ (*ANO8*); two-sided $P = 1.67 \times 10^{-2}$ was considered significant, correcting for three comparisons. Vertical lines represent standard

error. Sample sizes for each neocortical brain developmental stage are in Supplementary Table 12. **c**, Top, uniform manifold approximation and projection (UMAP) of scRNA-seq data[32], showing clustering of cell types from human iPS cell cultures developed towards midbrain neuronal cell types. Results are shown for cells developed for 11, 30 and 52 days: astrocyte-like (Astro), dopaminergic neurons (DA), ependymal-like 1 (Epen1), ependymal-like 2 (Epen2), floor-plate progenitors (FPP), neuroblasts (NB), proliferating floor-plate progenitors (P_FPP), proliferating serotonergic-like neurons (P_Sert), serotonergic-like neurons (Sert), unknown neuron 1 (U_Neur1), unknown neuron 2 (U_Neur2) and unknown neuron 3 (U_Neur3). Bottom, scDRS for each cell, with the strength indicated by the bar on the right. Red or blue indicates a positive or a negative score, respectively, reflecting increased or decreased expression of ADHD rare-variant risk genes compared with the distribution of expression of control gene sets in a cell.

enrichment among genes involved in several synaptic processes (Supplementary Table 11). This was especially the case for the MAP1A PPI network, in which 37 and 49 genes (out of 184) mapped to presynaptic and postsynaptic functions, respectively ($P_{\text{presynaptic ribosome}} = 3.02 \times 10^{-51}$; $P_{\text{postsynaptic ribosome}} = 6.94 \times 10^{-68}$).

In addition, the top 20 ADHD risk genes were enriched among genes expressed in the main axon ($P = 4.8 \times 10^{-7}$) and the initial segment of the axon ($P = 1.4 \times 10^{-6}$), and among genes involved in channelopathies ($P = 2.4 \times 10^{-7}$).

Because ADHD has a common-variant risk component that affects genes expressed in the brain[13], we evaluated the expression of the top associated rare-variant risk genes across neocortical brain developmental stages using BrainSpan data (see 'Data availability'). ADHD risk genes showed significantly higher mean expression in 11 out of 12

brain developmental stages (prenatal to adult) when compared with the average gene expression, and in 10 out of 12 stages when compared with neuronally expressed genes (Supplementary Table 12, Fig. 4a and Supplementary Fig. 8). This contrasts with common-variant risk genes, which were enriched only among genes expressed prenatally[13].

Overall, there was no difference in the mean expression of the top genes prenatally and postnatally ($P = 0.76$), but *MAP1A* was expressed significantly more highly postnatally than prenatally ($P = 2.2 \times 10^{-16}$) and the opposite was observed for *ANO8* ($P = 2.2 \times 10^{-16}$; Fig. 4b).

## Linking ADHD risk genes to cell types

Single-cell disease relevance scores (scDRSs)[30] were used to link risk genes to cell types. A higher scDRS indicates a stronger deviation of

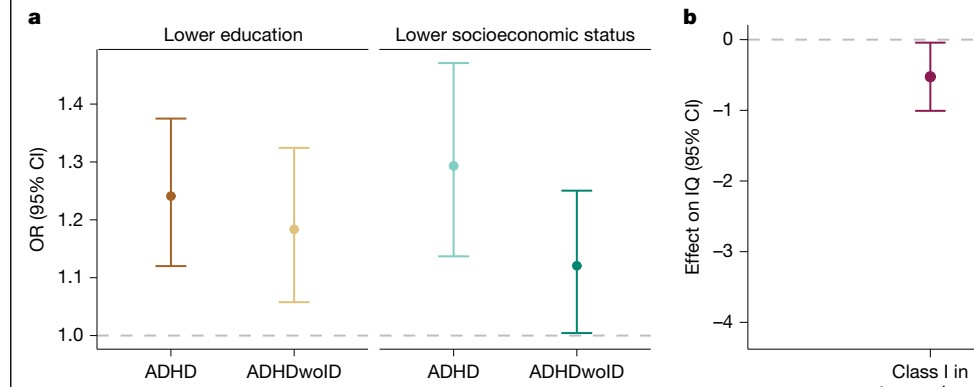

**Fig. 5 | Association of rare variants with education, socioeconomic status and IQ. a**, OR (from logistic regression) of lower education and lower SES in individuals with ADHD who have one or more rPTVs in constrained genes (pLI ≥ 0.9), compared with individuals with ADHD who do not have rPTVs in constrained genes. The analysis was also performed for individuals with ADHD without intellectual disability (ADHDwoID) who have one or more rPTVs in constrained genes, compared with individuals with ADHD without intellectual disability (ADHDwoID) who do not have rPTVs in constrained genes. ADHD with lower education, $n$ = 6,488; ADHD with higher education, $n$ = 1,436; ADHDwoID with lower education, $n$ = 5,297; ADHDwoID with higher education,

$n$ = 1,253; ADHD with lower SES, $n$ = 3,110; ADHD with higher SES, $n$ = 3,223; ADHDwoID with lower SES, $n$ = 2,509; ADHDwoID with higher SES, $n$ = 2,778). Dots represent OR point estimates, and error bars indicate the corresponding 95% CIs. **b**, Association of IQ and the number of ultra-rare class I variants in the German ADHD clinical sample ($n$ = 962). Results for all autosomal genes and constrained genes (pLI ≥ 0.9). The effect on IQ is the β coefficient from linear regression. Dots represent β coefficient point estimates, and error bars indicate the corresponding 95% CIs. The dotted line represents an OR of 1 in **a**, and a β of 0 in **b**.

the expression of risk genes from the expression of control gene sets in a cell. Common risk variants have previously been linked to genes with high expression in dopaminergic neurons in prenatal human midbrain[13]. In analyses of the same prenatal human midbrain single-cell RNA sequencing (scRNA-seq) dataset[31], we identified a significant association (that is, an increased scDRS across a cell type) between rare-variant risk genes and the dopaminergic neurons classified as type 1 in the previous study ($P$ = 9.99 × 10$^{-4}$), consistent with the common-variant findings[13]. Furthermore, we found a significant signal for GABAergic neurons ($P$ = 9.99 × 10$^{-4}$) and medial neuroblasts ($P$ = 2 × 10$^{-3}$) (Supplementary Fig. 9 and Supplementary Table 13). The results could be influenced by variables that we were not able to correct for in this older data (Methods), and thus we validated our findings in a more recent, larger dataset of developing midbrain neuronal cell types, derived from human iPS cells[32] generated from 215 healthy donors. Here we identified a significantly increased scDRS in both 30-day-old ($P$ = 9.99 × 10$^{-4}$) and 52-day-old ($P$ = 9.99 × 10$^{-4}$) dopaminergic neurons (Fig. 4c and Supplementary Table 13).

We also investigated the scDRS across 382 cell-type clusters representing a spectrum of neurons from the entire human brain[33], with no significant findings (Supplementary Table 13). In this dataset, the dopaminergic neuronal cluster (scDRS $P$ = 0.054), did not reflect the strong signal observed for the iPS-cell-derived dopaminergic neurons, which could be due to differences in neuronal age across datasets (neurons from post-mortem brains[33] versus iPS cells[32]).

Our results add to the emerging evidence that genetic risk variants, both common[13] and rare, influence ADHD by their effects on genes expressed in neurons, specifically GABAergic neurons and younger stages of dopaminergic neurons.

## Impact on socioeconomic outcomes

In common-variant analyses, a strong negative genetic correlation has been found between ADHD and cognition-related phenotypes, including educational attainment[13]. Here we examined the association of rPTVs in constrained genes with education level and SES among individuals with ADHD, by linking rare-variant data to data in the Danish registers. rPTVs had a negative effect on education level in individuals with ADHD; rPTVs in constrained genes were significantly associated with

finishing only primary school (OR = 1.24, s.d. = 5.23 × 10$^{-2}$, $P$ = 3.68 × 10$^{-5}$; Fig. 5a and Supplementary Table 14). Epidemiological studies have also consistently linked ADHD to decreased SES[34,35]. We found that rPTVs in constrained genes were significantly associated with low SES, defined by social security payment, early retirement benefit and/or unemployment for more than six months (OR = 1.28, s.d. = 0.07, $P$ = 9.09 × 10$^{-5}$). When excluding individuals with ID, the results remained significant for education and nominally significant for SES (Fig. 5a).

Overall, individuals with ADHD without ID with one or more rPTVs in constrained genes were around five to seven times more likely to have finished only primary school and to have lower SES, compared with controls in the iPSYCH cohort. This increased to six to eight times when including individuals with ID (Supplementary Fig. 10a,b).

We also evaluated the impact of ultra-rare class I variants on cognition in the clinical ADHD sample, in which we identified a decrease of 2.25 intelligence quotient (IQ) points per rare deleterious variant in constrained genes (β = −2.25, $P$ = 0.02; Fig. 5b and Supplementary Table 15). Our results add to researchers' understanding of ADHD, showing that rare deleterious variants, in line with common ADHD risk variants, have a negative effect on cognitive performance and life outcomes.

## Joint effect of common and rare variants

When we analysed GWAS data on common variants in ADHD, we found that 4 out of the top 20 rare-variant risk genes were nominally significantly associated with ADHD, including *MAP1A* ($P$ = 0.005, Supplementary Table 16), but that overall the gene set of the 20 genes was not enriched in common-variant associations.

The joint effect of rare and common variants in ADHD was evaluated by combining data on class I variants with ADHD polygenic risk scores (ADHD-PGSs) capturing the common-variant risk load. When considering individuals without class I variants in constrained genes, the ADHD-PGS was significantly associated with an increased risk of ADHD (OR$_{bin5\_vs\_bin1}$ = 6.52; s.e. = 0.06; Extended Data Fig. 3 and Supplementary Table 17), and the risk increased across PGS bins following a linear pattern (Extended Data Fig. 3). When considering individuals with one or more class I variants in constrained genes, the risk for ADHD was significantly higher within all PGS bins, when compared with individuals without class I variants of the same bin (except pentile 4; Extended Data

Fig. 3 and Supplementary Table 17). The risk increased following a linear pattern almost parallel to what was observed for individuals without class I variants, indicating that common and rare variants act additively on ADHD risk (Extended Data Fig. 3), in line with observations in other complex traits[26]. As shown in Extended Data Fig. 3, having one or more class I variants in constrained genes increased the risk of ADHD at a level comparable to a 20% increase in the common-variant polygenic risk load.

## Rare-variant load across comorbidities

We evaluated the burden of class I and II variants in individuals with ADHD who have also been diagnosed with ID ($n = 1,654$), autism ($n = 2,730$), schizophrenia ($n = 410$), substance-use disorders (SUDs; $n = 1,200$), disruptive behaviour disorders (DBDs; $n = 1,036$) or other psychiatric disorders combined ($n = 5,420$), referred to as multimorbidities, and compared this with the burden in individuals with ADHD who do not have these comorbid conditions (sample sizes in Supplementary Table 18). Class I and class II variants were significantly increased in constrained genes in ADHD with ID, compared with ADHD without ID (Extended Data Fig. 4a and Supplementary Table 19). ADHD without ID still showed a significantly higher load than controls, consistent with previous findings[14] (Supplementary Table 19 and Extended Data Fig. 4e). On the basis of these results, individuals with ID were excluded in analyses of other comorbidities. In analyses of comorbid autism, schizophrenia, SUD, DBDs and multimorbidities, there were no differences between individuals with comorbidities and individuals without comorbidities in the load of class I and class II variants across constrained genes (pLI ≥ 0.9; Extended Data Fig. 4b–d,g,h and Supplementary Table 19). In addition, we evaluated the load of class I and class II variants across comorbid subgroups in seven gene sets (listed in Supplementary Table 20) representing genes involved in autism, schizophrenia and neurodevelopmental disorders, which were identified in previous studies of rare variants (Supplementary Information). All gene sets, except one, showed an increased load of class I variants in ADHD with ID, as compared with ADHD without ID (Extended Data Fig. 5a and Supplementary Table 21). When compared with controls, ADHD without ID showed an increased load of class I variants across four of the gene sets, and two gene sets remained nominally significant for ADHD without multimorbidities (Extended Data Fig. 5b,c and Supplementary Table 21). These results suggest that rare deleterious variants in risk gene sets defined on the basis of constrained genes or disorders have a role in ADHD, even in individuals who are less severely affected; that is, with no ID or psychiatric comorbidities.

The seven rare-variant risk gene sets showed an increased load with some specificity towards comorbidity for the disorders to which they were most related to, but not for other co-occurring conditions. To specify, there were no significant findings in analyses of comorbidity with DBDs or SUD (Extended Data Fig. 5g,h), but when comparing ADHD comorbid with schizophrenia to ADHD without schizophrenia, we found a nominally significant increased load of class I variants in rare-variant schizophrenia risk genes[19] (OR = 7.27, CI = [1.67, 31.67], $P = 0.0083$; Extended Data Fig. 5e and Supplementary Table 21). ADHD comorbid with autism also showed a nominally significant increased load of class I variants in the three gene sets previously identified in rare-variant studies of autism, with the highest OR observed for the 'ASD_FDR0.001' gene set[28] (OR = 1.94, CI = [1.15, 3.30], $P = 0.014$; Extended Data Fig. 5d and Supplementary Table 21).

No strong impact of class II variants on comorbidity risk was identified—at least, not in the investigated gene sets (Extended Data Figs. 4b–d,g,h and 5d–h and Supplementary Tables 19 and 21).

Overall, our results suggest that the contribution from class I variants to the risk of psychiatric comorbidities (other than ID) in ADHD is, to some extent, driven by variants in sets of specific risk genes related to the comorbidity being considered, rather than being a result of a more general increased load across highly constrained genes.

## Rare variants across ADHD and autism

In our previous study of rare variants in ADHD and autism[14], we did not find any differences in the distributions of the genes affected by rPTVs across the two disorders. We have now substantially increased the sample size for both ADHD and autism, and re-examined whether the distributions of constrained genes with rare class I, class II and synonymous variants in ADHD only ($n = 5,536$) and autism only ($n = 7,554$) still have the same underlying distribution, by applying a C-alpha[36] test as used previously[14]. We found no significant differences between ADHD and autism risk genes for any of the variant groups analysed (Supplementary Table 22), when considering both individuals with ID and those without ID. By contrast, when comparing ADHD-only with controls, significant differences in the distributions of genes affected by both class I ($P = 8.09 \times 10^{-9}$) and class II ($P = 0.003$) variants were observed, and the same was observed when comparing autism-only to controls for class I variants ($P = 0.014$; Supplementary Table 22). These results suggest that there is a substantial sharing of rare-variant risk genes across ADHD and autism, and that deleterious variants might affect shared neurodevelopmental processes.

## Discussion

In this study, we have advanced researchers' understanding of the role of rare coding variants in ADHD risk and its comorbidities, revealed their association with major life outcomes and implicated pathophysiological components in ADHD rare-variant risk.

We identified three significant rare-variant risk genes for ADHD (*MAP1A*, *ANO8* and *ANK2*). *MAP1A* encodes a protein involved in the assembly of microtubules[37], and thus it could be hypothesized that disruption of this gene confers ADHD risk through dysfunction of the cytoskeleton, affecting synapse formation and function in neurons[38,39]. Notably, rPTVs in *MAP1A*'s sister gene *MAP1B* have been reported to cause ID, autism and extensive brain-wide deficits in white matter[40]. Both *ANK2* and *ANO8* encode proteins involved in calcium-ion transport across the plasma membrane[41,42], suggesting that neuronal synaptic channelopathies are involved in ADHD. This idea was also supported by our gene-set enrichment results, and is in line with what was proposed for autism[43].

Around 50% and around 30% of the association signal, for *MAP1A* and for *ANO8*, respectively, was driven by rare deleterious variants in individuals diagnosed with ADHD only (without comorbid schizophrenia, ID or autism), whereas the *ANK2* signal was driven mainly by ADHD with co-occurring autism or ID. This is consistent with *ANK2* being a known rare-variant risk gene in autism[28]. *MAP1A* and *ANO8* have not been associated with ASD or other neurodevelopmental conditions in other studies[19,28], although *MAP1A* was significantly associated when combining rare deleterious variants in both ADHD and ASD, but not significant when we examined the disorders separately in our previous study[14]. A discussion on the potential effect of comorbid autism on the results can be found in the Supplementary Information.

When evaluating the top 20 ADHD risk genes identified here, 9 have been implicated with autism and/or other neurodevelopmental disorders (FDR < 5%)[28], which supports the validity of our findings. Sixteen of the genes are evolutionarily constrained (Supplementary Table 4), which suggests that many of the genes we identify in rare-variant studies are genes involved in fundamental neurodevelopmental processes that are likely to be shared across disorders. That said, it is noteworthy that among our top genes are also genes that have not previously been linked to psychiatric disorders by rare variants and might be more specific to ADHD.

A previous study[44] identified *KDM5B* as a potential rare-variant risk gene for ADHD (FDR = 0.04). Although *KDM5B* was not significantly associated with ADHD in this larger dataset and was not among our

top 20 genes, it did show a moderate association signal, with an OR of 2.93 ($P = 1.23 \times 10^{-3}$).

Rare deleterious variants in constrained genes had the highest effect on ADHD risk, as reported previously for ADHD[14], autism[28,45] and schizophrenia[19]. Of note, we did not observe an increased load of rare deleterious variants in constrained genes in individuals with comorbid conditions, compared with those who had only ADHD (when excluding ID); likewise, we were not able to identify differences in the distribution of genes affected by rare deleterious variants in ADHD and autism. However, we did identify an increased load in ADHD with comorbid conditions in specific gene sets representing neurodevelopmental risk genes. This suggests that comorbid conditions—other than ID—are not associated with a general increased rare-variant risk load in constrained genes, but rather, with an increased load in a smaller set of specific risk genes. So, when sample size increases, we might be able to detect rare-variant risk genes that are more disorder specific.

Exploring the biological implications of the identified rare-variant risk genes, the broader PPI networks of the three significant genes were enriched for autism, DD and schizophrenia rare-variant risk genes, which supports the validity of our findings and reinforces the conclusion that rare-variant risk genes and their interacting proteins are most likely to affect gene networks involved in fundamental neurodevelopmental processes. Enrichment pointed towards several biological mechanisms or cell components that could potentially be affected in individuals with ADHD, including both presynaptic and postsynaptic functions, which were identified for all three networks. Our analyses also implicated dopaminergic neurons in ADHD. It has been hypothesized that dysregulation of dopamine in the brain has a role in ADHD, owing to the observation that stimulant medications, such as methylphenidate, often treat the symptoms of ADHD successfully[46]. Methylphenidate blocks the dopamine transporter, which increases neurotransmitter concentrations in the synaptic cleft[47].

In addition, we found that the top 20 risk genes had a high mean expression across all stages of brain development, suggesting that the genes have key roles in both the development and the function of the brain across lifespan. In accordance with this, we identified both immature neurons (medial neuroblasts) and mature GABAergic and dopaminergic neurons as enriched for the expression of rare-variant risk genes (as mentioned above). This suggests that a diverse set of neuronal dysfunctions could be involved in ADHD. When combining the effects of rare and common variants, rare variants were found to act additively with the common-variant PGS. Neuronal cell types, including dopaminergic neurons, have also been highlighted in analyses of common ADHD risk variants[13], and the biological mechanisms affected might be (partly) shared across common and rare variants. The convergence in affected biological mechanisms by variants across the allele spectrum might be a common feature of neurodevelopmental psychiatric disorders, given that this has also been observed for schizophrenia[19].

Assessing pertinent socioeconomic outcomes, we found that individuals with ADHD who carry one or more rare deleterious variants have significantly poorer educational attainment and lower SES than do people with ADHD who do not carry these variants (also when excluding comorbid ID). Compared with the general Danish population, individuals with ADHD who carry one or more deleterious variants were more than five to seven times as likely to experience these lower educational or socioeconomic outcomes. Likewise, ultra-rare deleterious variants had a significantly negative effect on IQ in adults with ADHD. Thus, rare deleterious variants have an impact not only on ADHD diagnosis but also on important life outcomes, which is consistent with the observation that an increased load of common ADHD risk variants is associated with decreased performance across a range of cognitive measures[13]. These findings show that ADHD is a polygenic disorder with a genetic component that influences certain cognitive domains, and this can present challenges for some individuals, particularly in academic settings. This insight could promote the development of supportive measures and innovative approaches designed to create better learning environments for individuals with ADHD.

In summary, we show that genes carrying rare deleterious risk variants have a considerable effect on ADHD, its comorbid conditions and key outcomes. We reveal the biological implications of rare-variant risk, and show that rare- and common-variant risk act additively in ADHD. While the risk genes described here explain only a fraction of the overall rare-variant risk, our study provides a path forward for identifying the many other risk genes that have yet to be discovered to further our understanding of ADHD pathophysiology.

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

[1]Department of Biomedicine–Human Genetics, Aarhus University, Aarhus, Denmark. [2]Lundbeck Foundation Initiative for Integrative Psychiatric Research (iPSYCH), Aarhus, Denmark. [3]Center for Genomics and Personalized Medicine, Aarhus, Denmark. [4]Novo Nordisk Foundation Center for Genomic Mechanisms of Disease, Broad Institute of MIT and Harvard, Cambridge, MA, USA. [5]Stanley Center for Psychiatric Research, Broad Institute of MIT and Harvard, Cambridge, MA, USA. [6]Analytic and Translational Genetics Unit, Department of Medicine, Massachusetts General Hospital and Harvard Medical School, Boston, MA, USA. [7]Center for Neonatal Screening, Department for Congenital Disorders, Statens Serum Institut, Copenhagen, Denmark. [8]Donders Institute for Brain, Cognition and Behaviour, Radboud University, Nijmegen, The Netherlands. [9]Department of Medical Neuroscience, Radboud University Medical Center, Nijmegen, The Netherlands. [10]Department of Psychiatry, Radboud University Medical Center, Nijmegen, The Netherlands. [11]Department of Psychiatry, Psychosomatics and Psychotherapy, Center of Mental Health, University Hospital Würzburg, Würzburg, Germany. [12]Department of Psychiatry and Psychotherapy, Medius Hospital of Kirchheim, Kirchheim unter Teck, Germany. [13]Department of Psychiatry, Psychosomatic Medicine and Psychotherapy, University Hospital Frankfurt, Goethe University, Frankfurt am Main, Germany. [14]Department of Psychiatry and Neurobehavioural Science, University College Cork, Cork, Ireland. [15]APC Microbiome Ireland, University College Cork, Cork, Ireland. [16]Division of Molecular Psychiatry, Center of Mental Health, University Hospital Würzburg, Würzburg, Germany. [17]Department of Child and Adolescent Psychiatry, Psychosomatics and Psychotherapy, Center of Mental Health, University Hospital Würzburg, Würzburg, Germany. [18]Department of Psychiatry and Neuropsychology, School for Mental Health and Neuroscience (MHeNs), Maastricht University, Maastricht, The Netherlands. [19]Department of Human Genetics, Radboud University Medical Center, Nijmegen, The Netherlands. [20]Fraunhofer Institute for Translational Medicine and Pharmacology (ITMP), Frankfurt am Main, Germany. [21]National Centre for Register-Based Research (NCRR), Business and Social Sciences, Aarhus University, Aarhus, Denmark. [22]Centre for Integrated Register-based Research at Aarhus University (CIRRAU), Aarhus University, Aarhus, Denmark. [23]Mental Health Centre Sct. Hans, Capital Region of Denmark, Institute of Biological Psychiatry, Copenhagen University Hospital, Copenhagen, Denmark. [24]Mental Health Centre Copenhagen, Capital Region of Denmark, Copenhagen University Hospital, Copenhagen, Denmark. [25]Psychosis Research Unit, Aarhus University Hospital, Aarhus, Denmark. [26]Department of Medicine, Harvard Medical School, Boston, MA, USA. [27]Institute for Molecular Medicine Finland, University of Helsinki, Helsinki, Finland. [28]These authors contributed equally: Ditte Demontis, Jinjie Duan. ✉e-mail: ditte@biomed.au.dk; anders@biomed.au.dk

## Methods

The iPSYCH study was approved by the Scientific Ethics Committee in the Central Denmark Region (case number 1-10-72-287-12) and the Danish Data Protection Agency. In accordance with Danish legislation, the above-mentioned ethics committee waived the need for specific informed consent in biomedical research based on existing biobanks. iPSYCH was approved by the ethics committee in 2012, with subsequent amendments in 2013, 2015 and 2018. More details can be found at https://ipsych.dk/en/data-security/health-research-and-ethical-approval. New Danish legislation (effective from January 2024) introduces the possibility for participants to opt out of studies that are exempt from active informed consent. After consulting with the Ethics Committees and patient organizations, iPSYCH contacted all participants (around 140,000) in the iPSYCH cohort in June 2025 and offered the possibility of opting out of new genetic studies initiated henceforth. Overall, 1.8% of the iPSYCH participants chose to opt out, and their data will be deleted from the active research database. Data included in finalized and ongoing studies will not be removed.

The clinical data were approved by the ethics committee at the University of Würzburg in Germany. In the Netherlands, they were approved by the regional ethics committee (Commissie Mensgebonden Onderzoek: CMO Regio Arnhem–Nijmegen; protocol III.04.0403 and 2014/290; ABR: NL47721.091.14) and the Institutional Review Board of the Radboud University Medical Center. Participants were included at the Department of Psychiatry at the Radboud University Nijmegen Medical Centre. All participants in the German and Dutch samples provided signed informed consent in accordance with the Declaration of Helsinki.

### Samples

**iPSYCH.** The individuals selected for exome sequencing were part of the iPSYCH cohort, which has been described in detail elsewhere[15,16]. In short, the study base includes all singleton births to mothers who were living in Denmark between 1 May 1981 and 31 December 2008, where the child was alive and resided in Denmark at their one-year birthday ($n = 1,657,449$). All individuals diagnosed with major psychiatric disorders by the end of 2016 according to the ICD10 criteria (ADHD (1.8% in the study base), autism, bipolar disorder, schizophrenia, major depressive disorder or post-partum depression ($n = 93,608$)) were identified in the study base using information in the Danish Psychiatric Central Research Registry[49] (and the Danish Patient Registry[50] for some disorders). In addition, 50,000 randomly selected population-based controls from the study base were selected. Subsequently, biological material for genotyping was obtained from the Danish Neonatal Screening Biobank (DNSB)[51]. The DNSB has stored residual biological material from screening of newborns for rare metabolic disorders since May 1981, and includes material from practically all births in Denmark since then. A subsample of 34,544 individuals was selected for whole-exome sequencing, and from these, we included individuals with an ICD10 diagnosis of ADHD (F90) in the Danish Psychiatric Central Research Registry[49] and the Danish Patient Registry[50] given before or during 2016; individuals with no diagnosis of the major psychiatric disorders (autism (ICD10: F84.0, F84.1, F84.5, F84.8 and F84.9), bipolar disorder (ICD10: F30–F31), schizophrenia (ICD10: F20) or major depressive disorder (ICD10: F32–F33)) were included as controls. The samples were included in iPSYCH in 2012–2016 and the sequencing was performed in 2012–2018.

For the comorbidity analyses, we identified individuals with the following ICD10 diagnosis codes in the Danish Psychiatric Central Research Registry: ID (ICD10: F70, F71, F72, F73, F78 and F79); ASD (as above); schizophrenia (as above); DBDs (including conduct disorder and oppositional defiant disorder (ICD10: F91 and F90.1)); SUDs (ICD10: F10.1–9, F11.1–9, F12.1–9, F13.1–9, F14.1–9, F15.1–9, F16.1–9, F17.1–9, F18.1–9 and F19.1–9) and multi-comorbidities, which, besides the comorbidities already listed, included comorbid anxiety (ICD10: F40.0–F40.2, F41.0–F41.1, F42 and F43.0–F43.1), tic disorder (ICD10: F95), bipolar disorder (ICD10: F30–F31), major depressive disorder (ICD10: F32–F33), anorexia nervosa (ICD10: F50.0), DDs (ICD10: F80–F83) and antisocial personality disorder (ICD10: F60.2).

**Clinical samples.** The clinical samples consisted of adults (over 18 years old) with persistent ADHD diagnosed according to the Diagnostic and Statistical Manual of Mental Disorders IV (DSM-IV) criteria. None of the individuals was diagnosed with ID. They were recruited as part of the International Multicenter Persistent ADHD Collaboration (IMpACT) at two sites: Radboud University Medical Center, the Netherlands, and University Hospital Würzburg, Germany. In analyses of these clinical samples, we used control samples from individuals recruited together with the clinical cases at the IMpACT site at Radboud University Medical Center (ADHD-screened controls), and from 1,766 German control individuals who were recruited from the German MI Family Study[52] and the Angio-Lub study; the latter samples were whole-exome-sequenced by the MIGen Exome Sequencing Consortium: Lubeck Heart Study (dbGaP accession number phs000990/DS-CVD, https://dbgap.ncbi.nlm.nih.gov/beta/study/phs000990.v1.p1/). The German controls consisted of 870 individuals with cardiovascular disease and 896 without. The total set of samples described in this section is referred to as the 'clinical sample' in the remainder of this manuscript.

**GnomAD.** All references to gnomAD refer to release 2.1.1 exomes from a subset of gnomAD consisting of individuals with non-Finnish European ancestry, and no diagnosis of psychiatric and neurological disorders ($n = 44,779$) (see 'Data availability').

### Exome sequencing and quality control

**iPSYCH.** In this study, we applied the same methods for quality control (QC) described in our previous study[14] to an updated dataset including new exome-sequenced individuals. To recap, the sequencing of 34,544 individuals from the iPSYCH cohort was performed using the Illumina Nextera Capture kit and the Illumina HiSeq sequencer. The sequencing process was performed in waves, including a pilot wave and three more substantial production waves. After sequencing, raw data were processed using the Genome Analysis Toolkit (GATK) v.3.4 to generate a variant call format (VCF) v.4.1 file. As per Danish regulations, American College of Medical Genetics (ACMG) v3.2 genes[53] were removed from the dataset. Next, we did thorough, multiple-round quality checks on the samples and the genetic variations using Hail 0.1. Samples were removed if they lacked complete phenotype information; if the imputed sex was inconsistent with the reported sex in the registries; if duplicates or genetic outliers were identified by principal component analysis (PCA); if they had an estimated level of contamination greater than 5%; or if they had an estimated level of chimeric reads higher than 5%. In addition, one of each pair of related samples was removed if the pairwise pi-hat value was 0.2 or higher.

Genotypes were removed if they did not pass GATK variant quality score recalibration (VQSR) or had a read depth lower than 10 or higher than 1,000. QC was done in Hail 0.1. Homozygous alleles were removed if they had reference calls with a genotype quality lower than 25, homozygous alternate alleles with PL(HomRef) (that is, the phred-scaled likelihood of being homozygous reference) < 25 or less than 90% of reads supporting the alternate allele. Heterozygous alleles were removed if they had PL(HomRef) < 25 or less than 25% of reads supporting the alternate allele, less than 90% informative reads (that is, number of reads supporting the reference allele plus number of reads supporting the alternate allele < 90% of the read depth) or a probability of the allele balance (calculated from a binomial distribution centred on 0.5) less than $1 \times 10^{-9}$. After these filters were applied, variants with a call rate lower than 90% were removed, then samples with a call rate lower than 95%, and then variants with a call rate lower than 95% were

removed. After QC, 28,448 individuals and 1,362,971 variants remained for further analysis.

Subsequently, we selected for this study the individuals diagnosed with ADHD and controls as described above, resulting in 8,895 cases and 9,001 controls (see Supplementary Table 23 for sample of individuals with other ancestries not included in this study). Finally, we defined rare variants ($n$ = 565,053) as those with an allele count no higher than five across our dataset ($n$ = 17,896) and the exome subset of gnomAD used in this study ($n$ = 44,779).

**Clinical samples.** Biological samples collected at the Dutch and German IMpACT sites were sequenced at BGI, Shenzhen, China. The coding regions of the DNA were targeted using BGI´s exome-capture kit (developed by Beijing Genomics Institute targeting 58.8 Mb) and paired-end sequenced on the Illumina HiSeq 2000 platform, with an average sequencing depth of 50×. The exome-sequencing data obtained from dbGaP were generated at the Broad Institute of Harvard and MIT using Illumina's ICE Capture reagent kit, and sequencing was performed on an Illumina HiSeq 2000 or 2500. Bam files for all samples (clinical cases and dbGaP controls) were reprocessed together, adhering to the same QC approach described above for the iPSYCH samples. After this QC process, 2,816 samples, consisting of 1,078 individuals with ADHD and 1,738 controls, remained for further analysis. We were not able to obtain a group of completely homogenous individuals after the exclusion of genetic outliers based on PCA (Supplementary Fig. 11), and thus, to minimize the effect of population stratification in subsequent analyses, we aimed at getting as close to de novo mutations as possible; we therefore included only ultra-rare variants, defined as singletons not in the non-psychiatric subsample of individuals in the ExAC[18] database. We could not filter based on presence in gnomAD because the dbGaP data (that is, the controls) are included in this database.

### Effects of variant categories on ADHD
Quality controlled variants were functionally annotated using SnpEff v.4.3[54,55], and SnpSift[54] was used to annotate information derived from dbNSFP[56]. If a variant had more than one annotation, only the most severe annotation was considered. PTVs were defined by being annotated as frameshift, splice-site or stop-gained, and predicted with a loss of function (LOF) flag by SnpEff. For missense variants, we annotated the potential effect of the variant on protein function using the MPC score[20].

In iPSYCH data, the burden of different rare-variant categories (rPTVs; rSevereDMVs, rModerateDMVs and rSYNs) in ADHD cases compared with controls was tested using logistic regression. Following the same approach, the load of rare variants across variant categories in female individuals with ADHD ($n$ = 2,265) was compared with male individuals with ADHD ($n$ = 6,630), both with and without individuals with ID (1,827 female individuals without ID; 5,414 male individuals without ID). The burdens in all genes and genes stratified by their pLI score were evaluated. Covariates included in analyses of iPSYCH samples were: birth year, sex, the first ten principal components (PCs) from PCA (performed after excluding non-European samples), total number of variants, number of rare synonymous variants, percentage of exome target covered at a read depth of at least 20, mean read depth at sites within the exome target passing VQSR, and sequencing wave (one-hot encoded).

The burden of different types of variant categories in gene sets was also tested for the clinical samples following the approach described above but restricting to ultra-rare variants and including the following covariates: sex, total number of variants, number of ultra-rare synonymous variants, number of all variants and PC1–PC10 from PCA (performed after excluding individuals with non-European genetic ancestry).

### Gene-based burden tests
To increase power for gene discovery, we combined iPSYCH data with a subset of 44,779 individuals from gnomAD[17] (as defined above). Only variants in high-confidence regions for the two datasets were included, defined as regions in which at least 80% of the samples in both datasets had at least 10× sequencing coverage (based on analysis of bam files for the Danish samples and on coverage summary tables for gnomAD). To avoid biases caused by variations in call rates between cases (entirely iPSYCH) and controls (83.3% gnomAD), all autosomal genes with higher rates of rare synonymous variants in cases than controls were excluded (15,603 out of 18,866 autosomal genes remained; 3,263 excluded).

Variants were grouped depending on their impact on ADHD: class I variants include rPTVs and rare missense variants with MPC > 3 (rSevereDMVs); class II variants include missense variants with $2 \leq MPC \leq 3$ (rModerateDMVs). Gene-based burden analysis was performed with a two-tailed Fisher's exact test. Only genes with an increased load of class I (3,698 genes) and/or class II (1,026 genes) variants in cases compared with controls were considered. We did not consider genes with higher numbers of class I and II variants in controls, because such observations could be caused by a generally higher number of variants in controls, owing to the previous filtering step in which only genes with a higher number of rare synonymous variants in controls than in cases were retained. A small number of genes (347) had an increased load of both class I and class II variants in cases, and for these genes we also estimated the combined impact of class I and class II variants in a meta-analysis using the weighted $z$-score method. Weights were the ratio of the standardized effect sizes observed for the classes in enrichment analysis of constrained genes. For the 347 genes, we used the minimum $P$ value across the three analyses done for these genes. In total, we tested 15,603 genes (347 genes were tested three times) resulting in 16,297 tests. We considered a gene significantly associated with ADHD if the $P$ value was lower than $0.05/16,297 = 3.07 \times 10^{-6}$.

As described above, genes with an increased rate of rare synonymous variants in individuals with ADHD relative to controls were excluded to avoid biases when combining iPSYCH and gnomAD data. When inspecting the QQ-plot of rare synonymous variants (Supplementary Fig. 12), the plot revealed the anticipated pattern of inflation that corresponds to our gene selection strategy.

In the clinical sample, we performed a gene-based burden test of class I variants using emmaxCMC[57,58] implemented in EPACTS v.3.3.0 (https://genome.sph.umich.edu/wiki/EPACTS). This method allows for the incorporation of a kinship matrix to account for potential remaining population stratification among individuals. We generated the kinship matrix using 'epacts make-kin --min-maf 0.01 --remove-complex'. In addition, we included sex, the number of ultra-rare synonymous variants, the number of all variants and PC1–PC10 from PCA (performed after excluding non-European samples) as covariates.

### Gene-set analyses in the clinical sample
We tested for increased load of ultra-rare class I and class II variants in ADHD compared with controls in the clinical sample. We used different $P$-value thresholds from the gene-based burden test in iPSYCH + gnomAD samples to define three ADHD rare-variant risk gene sets: $P < 1 \times 10^{-3}$ (20 genes), $P < 5 \times 10^{-2}$ (316 genes) and $P < 1 \times 10^{-1}$ (583 genes) (gene sets are listed in Supplementary Table 20). The control group included 870 with cardiovascular disease and 896 without, so we also tested for the potential difference in ultra-rare class I variants across the two groups in the three ADHD gene sets. Only class I variants were tested, because these were the only type of variants with a tendency to be overrepresented in clinical ADHD cases versus controls (Supplementary Information, Supplementary Fig. 13 and Supplementary Table 24). The gene sets were tested using logistic regression with sex, number of ultra-rare synonymous variants, number of all variants, and PC1–PC10 from PCA (performed after excluding non-European samples) as covariates.

### X-chromosome analyses
Given the distinct regions of the X chromosome, we applied region-specific thresholds to define rare variants: in the pseudoautosomal

regions (PARs), rare variants were defined as those with an allele count of five or less following what was done for the autosomes. In non-pseudoautosomal regions (nonPARs), the rare-variant threshold was adjusted to an allele count of three of less to account for the hemizygosity in male individuals. This adjustment was derived using a scaling factor based on the relative contributions of male individuals and female individuals in iPSYCH (6,002 female and 11,894 male individuals) and subset of gnomAD used in this study (19,916 female and 24,863 male individuals):

$$5 \times \frac{(\text{num}_{\text{males}} + 2 \times \text{num}_{\text{females}})}{2 \times (\text{num}_{\text{males}} + \text{num}_{\text{females}})} = 3.53$$

Variants were categorized into class I, class II and synonymous variants, using the same criteria applied for autosomal variants. In the burden test of male individuals, the number of alleles for nonPAR variants was counted as two to account for hemizygosity.

Logistic regression was performed on three gene sets stratified by their pLI score, comparing the following groups of individuals: ADHD versus controls; ADHD without ID versus controls; ADHD with ID versus controls; and ADHD with ID versus ADHD without ID. Covariates included birth year, sex, the first ten PCs from PCA (non-European samples excluded), total variant count, rare synonymous variant count, exome target coverage (percentage ≥ 20×), mean read depth at target sites and sequencing batch (one-hot encoded). In addition, sex-stratified analyses were performed using the same approach.

For gene discovery on the X chromosome, we applied the same approach as for autosomes when combining iPSYCH data with gnomAD. We excluded genes with higher numbers of synonymous variants in individuals with ADHD than in controls and retained genes with a higher rate of class I variants in cases than in controls (78 genes included) or a higher number of class II variants (44 genes). We performed gene-based burden analysis separately for male individuals and female individuals, in which we, for each gene, compared the number of individuals with ADHD carrying at least one class I or class II variant to the number found for control individuals, using a two-tailed Fisher's exact test. Notably, six genes had both increased class I and II variants in cases compared with controls; for these genes, a meta-analysis was used to combine the effect of class I and class II variants using the weighted $z$-score method. The weight was determined as the ratio of the standardized effect sizes observed for class I and class II variants in the enrichment analysis of constrained genes on chromosome X (weight = 1.03/1.86 = 0.55). In total, we performed 78 × 3 tests for class I variants (female, male and combined), 44 × 3 tests for class II variants and 6 tests for combining class I and II, resulting in a total of 372 tests. The Bonferroni correction threshold for statistical significance was set at $P < 1.34 \times 10^{-4}$.

### Rare burden heritability

We used BHR[26] to estimate the burden heritability and the contribution of a gene set to the burden heritability (namely, the burden heritability enrichment of a gene set). We used BHR to estimate the heritability of ADHD explained by the load of rare class I, class II and synonymous variants.

Variant-level summary statistics associated with ADHD, including allele count and allele frequency from the iPSYCH exome data, were used as input for BHR. The method regresses gene burden test statistics (based on the variant category being evaluated) against burden scores that correspond to the combined allele frequency. The slope of the regression represents the burden heritability and confounding factors such as population stratification, are controlled through the intercept.

### PPI-network analyses

The ANK2 PPI networks were derived from published IP–MS datasets included in Table S2 of a previous study[27]. All significant proteins with log$_2$-transformed fold change (FC) > 0 and FDR ≤ 0.1 in the ANK2_WH, ANK2_CNCR1 and ANK2_CNCR2 datasets were defined as the ANK2 interactors in ExNs; the significant proteins in the ANK2_WT dataset were defined as the ANK2 interactors in NPCs.

The MAP1A and ANO8 PPI networks in NPCs and ExNs were derived from IP–MS experiments performed in this study. We evaluated the expression of cell-type marker genes using single-nucleus RNA-seq (snRNA-seq) and performed immunofluorescence staining on NPCs and fully differentiated ExNs to confirm their identities as immature neural progenitors and upper-layer prefrontal cortex neurons, respectively (results shown in Supplementary Fig. 14). Details on cell culture and differentiation, snRNA-seq, immunofluorescence, protein extraction, immunoprecipitation, immunoblotting, mass spectrometry and IP–MS data analysis can be found in the Supplementary Information. Consistent with the ANK2 networks, we defined significant proteins with log$_2$-transformed FC > 0 and FDR ≤ 0.1 in each IP–MS experiment to be the interactors of the index protein. The resulting IP–MS datasets are provided in Supplementary PPI Tables 1–4.

We parsed the interactors identified across all IP–MS datasets into 12 PPI networks grouped by index proteins and cell-type specificity (Supplementary PPI Table 5). For each index protein, we merged all interactors identified in the same cell type into an 'NPC' or 'ExN' network, or in either cell type, into a 'Union' network. We also merged interactors for all three index proteins into combined NPC, ExN and Union networks accordingly.

We also annotated unique interactors identified across all IP–MS datasets with the following information (Supplementary PPI Table 6): (1) name and number of associated baits (index proteins) in NPCs, ExNs or either cell type; (2) whether the interactor had been implicated in genetic association studies of ADHD ($P < 0.001$ in this study, or by common-variant risk genes listed in Supplementary Table 7 of Demontis et al.[13]), autism, DDs, neurodevelopmental disorders (Supplementary Table 11 of Fu et al.[28]) or schizophrenia (Supplementary Table 5 of Singh et al.[19] at various significance thresholds).

For subsequent network enrichment analyses, we compared the network genes against a background set of protein-coding genes expressed in neurons (hereafter, 'neuronal background'). To define the neuronal background, we re-analysed RNA-seq data derived from the same ExN cellular model used in this study (day-21 and day-51 data from the Gene Expression Omnibus (GEO) GSE178896 dataset)[27]. We first performed transcript quantification from FASTQ files using Salmon (v.1.10.2)[59] and GENCODE (v.43)[60] reference files. We summarized the quantification results to gene-level counts using tximport (v.1.26.1), then removed non-protein-coding genes and low-count genes using the filterByExpr function in edgeR (v.3.40.2). This resulted in a list of 13,018 neuronal background genes (Supplementary PPI Table 7) to be used in downstream analyses.

To perform rare-variant enrichment analysis for the PPI networks, gene-based association scores were obtained from exome-sequencing studies of autism and DDs (FDR_TADA_ASD and FDR_TADA_DD columns in Supplementary Table 11 of Fu et al.[28]) and schizophrenia ('P meta' column in Supplementary Table 5 of Singh et al.[19]). Loss-of-function constraint scores (pLI scores) were obtained from gnomAD v.2.1.1[17]. For each phenotype and each PPI network, we performed a one-tailed Kolmogorov–Smirnov test to assess whether the network genes had more significant scores than the neuronal background genes. Because the PPI networks were significantly enriched for loss-of-function constrained genes, we also repeated the analysis for constrained (pLI ≥ 0.9) and non-constrained (pLI < 0.9) genes separately. That is, we used one-tailed Kolmogorov–Smirnov tests to compare the network genes with pLI ≥ 0.9 to other neuronal background genes with pLI ≥ 0.9, and vice versa for the network genes with pLI < 0.9.

A description of the common-variant enrichment analysis can be found in the Supplementary Information.

## Enrichment analyses

We tested for enrichment of top associated rare-variant risk genes (20 genes with $P < 1×10^{-3}$ in ADHD iPSYCH cases versus iPSYCH + gnomAD controls) among genes in the following gene sets: (1) gene sets related to gene ontology analysed using the Gene Ontology (GO) knowledge-base[61,62] (GO biological processes v.2022-07-01, 9,290 gene sets; GO cellular component ontology v.2022-07-01, 1,581 gene sets; GO molecular function ontology v.2022-07-01, 2,997 gene sets); (2) genes related to synapse function using synaptic annotations based on published, expert-curated evidence for 1,602 genes in SynGO[29] (v.20231201 release 1.2); the list of brain-expressed genes provided by SynGO was used as background; (3) gene sets related to biological pathways (BioCarta 2016, 237 gene sets; KEGG 2021, 320 gene sets; Reactome 2022, 1,818 gene sets; canonical pathway gene sets derived from the WikiPathways pathway database 2021, 622 gene sets)[29]; and (4) gene sets related to diseases (PheWeb v.2019, 1,161 gene sets; PhenGenI Association v.2021, 950 gene sets; GWAS catalogue v.2021, 17,37 gene sets; DisGeNET v.6.0, 9,828 gene sets; OMIM Disease, 90 gene sets). Enrichment analyses of the latter two (that is, pathway- and disease-related gene sets) were performed using Enrichr[63]. All of the enrichment analyses were done using a one-sided Fisher´s exact test and the within-database correction for multiple testing was done using the Benjamini−Hochberg method. A gene set was considered significant if the within-database corrected $P$ value was less than $P = 0.0038$ (0.05 divided by the number of databases [0.05/13 = 0.0038]).

In addition, the genes encoding proteins in the union PPI networks (NPCs + ExNs) for each of the three index proteins (MAP1A, ANO8 and ANK2) were tested for enrichment among gene sets related to (1) gene ontology, (2) synapse function and (3) biological pathways, as described above.

## Expression of risk genes in the brain

Expression of the top 20 rare-variant risk genes ($P < 0.001$) across neocortex developmental stages was evaluated using bulk RNA-seq data v.10 (gene-level RPKMs) obtained from BrainSpan (www.brainspan.org). The samples represent an age span from post-conceptual week 8 to 40 years of age, and were grouped into brain developmental stages as defined previously[64]. Following a previous study[45], we analysed neocortical regions. Samples with poor quality (RIN ≤ 7) were removed. Genes were defined as expressed if the RPKM was at least one in at least 80% of the samples for at least one neocortical region in one major temporal epoch.

After filtering, the BrainSpan dataset contained expression data from 324 samples (information about the number of samples analysed for each developmental stage can be found in Supplementary Table 12). We focused on the 20 ADHD risk genes, of which 17 had expression data in BrainSpan. Gene expression was natural logarithm-transformed ($\log(\mathrm{RPKM} + 1)$) and a two-sided paired $t$-test was used to test for differential expression of the set of the 17 ADHD risk genes against a background gene set (22,402 genes or transcripts) for all developmental stages merged and across developmental stages ($P = 4.17×10^{-3}$ was considered significant correcting for 12 brain developmental stages). A two-sided paired $t$-test was also performed to determine differential expression of the 17 genes prenatally and postnatally. Likewise, each of the three exome-wide-significant genes (MAP1A, ANK2 and ANO8) were tested for differences in pre- and postnatal expression. Because the sample sizes were small for some developmental stages, we tested to see whether the data were normally distributed. If this assumption was violated, a non-parametric Wilcoxon signed-rank test was performed instead.

## Linking risk genes to cell types

We used scDRS[30] to link rare-variant risk genes to cell types. First, we used a hypothesis-based approach that focused on midbrain neurons, motivated by findings linking common variants[13] to midbrain dopaminergic neurons. We used four datasets from two published snRNA-seq and scRNA-seq studies (see 'Data availability'): one scRNA-seq dataset from a study (study I) of gene expression in prenatal human brain cells[31] derived from ten human embryos 6–11 weeks old (1,695 cells analysed); and three snRNA-seq datasets from a study (study II) of gene expression in developing midbrain neuronal cell types[32]. These data were generated from 215 pluripotent stem cell (iPS cell) lines, each derived from a single healthy donor (88 male and 127 female individuals), for differentiation towards midbrain neuronal cell types. The three datasets represent cells developed for 11 days (253,381 cells analysed), 30 days (250,923 cells analysed) and 52 days (303,856 cells analysed). We used processed scRNA-seq and snRNA-seq count data and cell-type annotations generated as described in the two papers[31,32]; for study I, the data were downloaded as a collapsible expression format (CEF) file and converted into a hierarchical data format version 5 annotated data (h5ad) file format using the Python package AnnData (see 'Data availability'); for study II, h5ad file formats were available. For study II, the 11-day and 30-day datasets (downloaded) were used without any modifications, whereas the 52-day dataset was filtered to remove cells that were treated with rotenone (following what was done in the published study[32]). In addition, the raw read counts from study I (ref. 31) and the 52-day data from study II (ref. 32) were filtered so that only genes represented in a minimum of 30 cells were included.

Furthermore, we used a hypothesis-free approach, in which we evaluated the scDRS in 382 cell-type clusters (2,480,956 cells analysed) representing a spectrum of neurons from the entire human brain (study III; ref. 33). This dataset consists of snRNA-seq data of neurons desiccated from four adult post-mortem human brains (three male individuals and one female individual) from 105 locations across the forebrain (cerebral cortex, hippocampus, cerebral nuclei, hypothalamus and thalamus), midbrain and hindbrain (pons, medulla and cerebellum). We used a h5ad file with processed data (see 'Data availability') and cell cluster definitions reported previously[33].

For the dataset from study I (ref. 31) and the 52-day dataset from study II (ref. 32), there were no generated UMAPs for visualizing cell-type clusters. To generate UMAPs for these data, we did the following, using Python and the single-cell analysis libraries Scanpy and AnnData: for each of the four scRNA-seq datasets, the raw counts were filtered to contain genes represented in a minimum of 30 cells. Afterwards, the raw counts were normalized and log-transformed, and highly variable genes were identified using 'highly_variable_genes' in Seurat[65] implemented in Scanpy with default settings. The expression data for the highly variable genes were scaled and used as input in PCA generated using the singular value decomposition (SVD) solver method in the 'arpack' algorithm implemented in Scanpy. PCs from the PCAs were then used to compute a neighbourhood graph generated using the Scanpy function 'pp.neighbors', which afterwards was embedded and visualized as a UMAP constructed with the Scanpy function tl.umap. For study I, the neighbourhood graph was constructed with the first 20 PCs and 'n_neighbors' set to 100. For study II, the neighbourhood graph was constructed with the first 40 PCs and 'n_neighbors' set to 30 for each of the 3 datasets. The parameters were set based on visual inspections.

We used the $P$ values from the top 100 most associated genes from the gene-discovery analysis of iPSYCH + gnomAD samples as input for scDRS analysis, which was done separately for the above-described data from studies I, II and III. The number of genes was set to 100, which is the recommended minimum number of genes for the method. The method computes, on the basis of the expression of these 100 most significant genes, one raw disease score per cell. In addition, 1,000 raw control scores were computed for each cell. The control scores were generated using Monte Carlo sampling, in which 1,000 gene sets were produced, each consisting of 100 genes with similar gene size, mean gene expression and expression variance to that observed for the disease gene set.

The computation of scores was corrected for relevant covariates (see below) and the number of genes in each of the cells. The normalized disease score was compared with the empirical normalized control score distribution to estimate a P value that quantifies the association between the disease genes and their expression in individual cells. These scores were used in the downstream scDRS cell-type-level analysis to test the association between the scores and predefined cell types using a t-test and the top 5% quantile of the disease score of cells from the given cell type. In addition, a heterogeneity test using the Geary's C statistic[66] was performed to determine heterogeneity in disease score within cell types. Both the compute-score and the perform-downstream functions were run with default settings on raw counts.

We included donor ID as a covariate in analyses of studies II and III. Information on donor ID was not available for study I, and we were therefore not able to correct for the potential effects of factors captured by the donor ID covariate; that is, differences in biological variance between individuals, potential technical variation linked to donor samples or unequal donor representation among cells. The day of cell collection (collected from week 6 to week 11) was also used as a covariate in analyses of data from study I. Sex was used as a covariate in the analyses of data from study II. Information about the sex of the donors was obtained from the Human Induced Pluripotent Stem Cell Initiative (HipSci) data browser (see 'Data availability') and merged with the downloaded data. Results were considered significant after within-study Bonferroni correction, correcting for the number of cell types analysed.

### Gene-set analyses across comorbidities

The loads of class I and class II variants in individuals with ADHD comorbid with other disorders (ID, autism, schizophrenia, DBDs, SUD and multi-comorbidities; see diagnosis codes above), compared with ADHD without comorbidities, were evaluated in iPSYCH samples for sets of autosomal genes grouped by their pLI score: $pLI \geq 0.9$ (2,811 genes), $0.5 < pLI < 0.9$ (1,332 genes), $pLI \leq 0.5$ (14,267 genes), and for 7 gene sets related to other psychiatric disorders and ID: (1) 'SCZ_Pval2.14e-6' includes the 10 genes significantly associated with schizophrenia at $P < 2.14 \times 10^{-6}$, identified from exome-sequencing data[19]; (2) 'SCZ_Qval0.05' includes a broader set of 32 genes with a nominal association with schizophrenia at $q < 0.05$, identified from exome-sequencing data[19]; (3) 'ASD_Pval2.5e-6' includes 60 genes significantly associated with ASD at $P < 2.5 \times 10^{-6}$, identified from sequencing data[67]; (4) 'ASD_FDR0.001' includes 72 genes significantly associated with ASD at FDR < 0.001, identified from exome-sequencing data[28]; (5) 'ASD_FDR0.05' includes 183 genes with a nominal association with ASD at FDR < 0.05 based on exome-sequencing data[28]; (6) 'NDD_FDR0.001' includes 373 genes significantly associated with neurodevelopmental disorder (NDD) at FDR < 0.001, identified from exome-sequencing data[28]; and (7) 'NDD_FDR0.05' includes 662 genes with a nominal association with NDD at FDR < 0.05, identified from exome-sequencing data[28] (gene sets are listed in Supplementary Table 20). The analyses were done using logistic regression and the same covariates as were used in analyses of the effects of variant categories on ADHD in iPSYCH samples (described above). The load of synonymous variants was also evaluated as a sanity check, because we expected no different load for this variant category. We considered tests with $P < 0.05/7 = 7.14 \times 10^{-3}$ as significant (seven gene sets tested).

### Effects of rare variants on education and SES

A copy of the whole-exome-sequencing data (after QC and functional annotation) on iPSYCH ADHD cases was transferred to Statistics Denmark (no controls due to restriction on file size), to link rare variants to variables only available at the secured servers at Statistics Denmark. In these analyses, rare variants were defined as singletons in iPSYCH ADHD cases. Individuals with ADHD with at least one rPTV (nonsense, frameshift and essential splice-site variants) in constrained genes ($pLI \geq 0.9$) were compared with individuals with ADHD without rPTVs in constrained genes; rPTVs were used as exposure, and high versus low SES or education were used as outcomes in logistic regression analysis using the glm() function in R and for two-tailed Fisher's exact test using the fisher.test() function in R[68]. We corrected for gender, birth year, first ten PCs generated from ancestry PCA, number of rare synonymous variants, percentage of target with coverage greater than 20×, mean read depth at sites within the exome target passing VQSR, total number of variants, and sequencing wave. As control tests, we analysed rare synonymous variants in constrained genes.

SES information was obtained from the Income Statistics Register[69]; data were available for 5,297 individuals with ADHD who were over 16 years old. The 'low SES' group was defined as individuals receiving social security payment, receiving early retirement benefit and/or having been unemployed for more than six months ($n = 3,110$); the 'high SES' group consisted of the remaining individuals ($n = 3,223$). Education level was obtained from the Danish Population Education Register[70]. Low education was defined as finishing only primary school, which is nine years of schooling in Denmark ($n = 6,488$); high education was defined as an education beyond primary school ($n = 1,463$).

We used two-tailed Fisher´s exact test to estimate the risk of low SES or low education among individuals with 'ADHD with one or more rPTVs on constrained genes' or individuals with 'ADHD without rPTVs in constrained genes' against 21,413 population-based controls (over 16 years old).

We tested for associations of ultra-rare deleterious variants with measures of IQ in adults with ADHD in the German clinical sample ($n = 962$ individuals). Intellectual function was assessed with the Mehrfachwahl Wortschatz Intelligenztest (MWT-B)[71] test, and none of the recruited individuals had an IQ lower than 80. We used linear regression with the 'lm' function in R to assess the correlation between IQ and the number of ultra-rare class I variants in all autosomal genes and constrained genes ($pLI \geq 0.9$). The analysis was adjusted for sex, first ten PCs from PCA, number of ultra-rare synonymous variants and total number of variants.

### Overlap with common-variant risk loci

Common-variant gene-based associations were calculated using MAGMA[72] and summary statistics from our previous GWAS meta-analysis of ADHD[13]. Association was tested using the SNP-wise mean model and linkage disequilibrium correction was based on estimates from samples with European ancestry from phase 3 of the 1000 Genomes Project[73]. No window around genes was used. Gene-based results were subsequently used in a MAGMA competitive gene-set analysis to test for the enrichment of common-variant associations in two gene sets: (1) rare-variant risk genes with $P < 0.001$ (20 genes); and (2) rare-variant risk genes with $P < 0.005$ (62 genes).

### Joint effect of common and rare variants

For the individuals in iPSYCH, common-variant data are also available. These data were included in our previous GWAS meta-analysis of ADHD, which contains detailed information on data generation[13]. In short, the samples were genotyped using Illumina's PsychChip (iPSYCH1 samples) or Illumina's Global Screening Array (iPSYCH2 samples). QC, imputation and association analysis were done using the bioinformatics pipeline Ricopili[74]. After stringent QC, individuals and variants were included according to the following parameters: subject call rate > 0.95, autosomal heterozygosity deviation (|Fhet| < 0.2), variant call rate > 0.98, difference in variant missingness between cases and controls < 0.02 and SNP Hardy–Weinberg equilibrium (HWE) ($P > 10^{-6}$ in controls and $P > 10^{-10}$ in cases). Imputation was done separately for iPSYCH1 and iPSYCH2 samples using EAGLE v.2.3.5[75] and Minimac3[76], and the Haplotype Reference Consortium[77] panel v.1.0 was used as reference.

The PGS was constructed by splitting the relatedness pruned GWAS dataset (25,895 with ADHD; 37,148 controls) in 50 random subsets of

roughly even size. For each of these, a GWAS was run on the complimentary 49 subsets using 10 PCs as covariates, and the results were meta-analysed with PGC and deCODE ADHD GWAS summary statistics described elsewhere[13]. The resulting summary statistics were then used to generate PGSs in the index set using the SBayesR algorithm implemented in LDAK[78]. Finally, the scores from the 50 subsets, which together cover the full dataset, were assembled into one dataset.

We binned the PGSs into pentiles, and ten dummy variables were generated identifying the individuals in each pentile bin who have at least one class I variant or none, respectively. Logistic regression of ADHD status on the nine dummy variables was performed to estimate the impact on ADHD risk in each PGS pentile using individuals in the first pentile with no class I variants in constrained genes as reference. This was done separately for individuals with no class I variants in constrained genes and for individuals with at least one class I variant in constrained genes. The regression was adjusted for individual birth year, total number of variants, number of rare synonymous variants, percentage of exome target covered at a read depth of at least 20, mean read depth at sites within the exome target passing VQSR, sequencing wave, and the first ten PCs.

### C-alpha test
To assess whether the rare variants identified in ADHD and ASD come from the same underlying gene distribution, we performed C-alpha tests on the iPSYCH exome data, including both ADHD and ASD. This comparison was designed to be simultaneous. To start with, we identified individuals diagnosed with either ADHD or ASD by the end of 2016, along with a control group in the quality-controlled iPSYCH data ($n$ = 28,448). Subsequently, we defined and classified the rare variants into class I, class II and synonymous categories.

We performed a C-alpha test between ADHD without ASD (ADHD only) and ASD without ADHD (ASD only), regardless of ID comorbidity. In addition, we stratified the samples by the presence or absence of ID to perform C-alpha tests between ADHD only and ASD only. A separate set of C-alpha tests compared the single disorders with the control group.

The C-alpha test[36], implemented in the AssotesteR R package (http://cran.r-project.org/web/packages/AssotesteR/index.html), was used to evaluate the similarities between ADHD and ASD, and between controls and either ADHD or ASD. Each pairwise comparison for class I, class II and synonymous variants underwent 10,000 permutations, allowing us to verify the asymptotic $P$ value against the permutation-based $P$ value.

### Reporting summary
Further information on research design is available in the Nature Portfolio Reporting Summary linked to this article.

## Data availability
iPSYCH data, including gene-based summary statistics data, are available from the authors after approval by the iPSYCH Data Access Committee and can only be accessed on the secured Danish server (GenomeDK; https://genome.au.dk) because the data are protected by Danish legislation. With regard to the clinical samples, which include samples from the International Multicenter persistent ADHD Collaboration (IMpACT) at Radboud University Medical Center, the Netherlands, and University Hospital Würzburg, Germany, the ethical permissions do not allow sharing of the data outside the secured Danish server and can only be used to study ADHD. For data access and correspondence, please contact one of the corresponding authors. The response time will be within two weeks. The IP–MS data for MAP1A and ANO8 have been deposited to MassIVE with identifier MSV000098548. Other data sources are as follows: BrainSpan: www.brainspan.org; gnomad v.2.1.1: https://gnomad.broadinstitute.org/downloads#v2-lof-curation-results; gnomAD release 2.1.1: https://gnomad.broadinstitute.org/downloads#v2; La Manno et al.[31] scRNA-seq data: GSE76381 (GSE76381_EmbryoMoleculeCounts.

cef.txt.gz); Jerber et al.[32] snRNA-seq datasets: https://zenodo.org/record/4651413#.ZAcbxXbMJEZ; Siletti et al.[33] snRNA-seq data: CZ CELLxGENE platform at https://datasets.cellxgene.cziscience.com/f9ecb4ba-b033-4a93-b794-05e262dc1f59.h5ad; Pintacuda et al.[27] RNA-seq data: GSE178896; HipSci data browser: https://www.hipsci.org/#/lines; Python package AnnData: https://anndata.readthedocs.io/en/latest/generated/anndata.AnnData.html; and EPACTS: https://genome.sph.umich.edu/wiki/EPACTS.

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

**Acknowledgements** D.D. was supported by the Novo Nordisk Foundation (NNF20OC0065561, NNF21SA0072102, NNF23OC0083657), the Lundbeck Foundation (R344-2020-1060) and the European Union (EU)'s Horizon 2020 research and innovation programme under grant agreement 965381(TIMESPAN). The iPSYCH team was supported by grants from the Lundbeck Foundation (R102-A9118, R155-2014-1724 and R248-2017-2003), the National Institute of Mental Health (NIMH) of the National Institutes of Health (NIH) (1R01MH124851-01 to A.D.B.), the EU's Horizon Europe programme under grant agreement 101057385 (R2D2-MH to A.D.B.) and the Universities and University Hospitals of Aarhus and Copenhagen. The Danish National Biobank

resource was supported by the Novo Nordisk Foundation. High-performance computer capacity for handling and statistical analysis of iPSYCH data on the GenomeDK HPC facility was provided by the Center for Genomics and Personalized Medicine and the Centre for Integrative Sequencing (iSEQ), Aarhus University, Denmark (grant to A.D.B.). K.-P.L. was supported by ERA-NET NEURON under grant 01EW1902 (DECODE!), the EU's Horizon 2020 research and innovation programme under grants 728018 (Eat2beNICE), 101086453 (Aqua-Synapse) and 953327, the Deutsche Forschungsgemeinschaft (DFG KFO 125, SFB 581, SFB TRR 58 and GRK 1253) and the Bundesministerium für Bildung und Forschung (BMBF 01GV0605). For the Dutch IMpACT sample, we acknowledge support from the European College of Neuropsychopharmacology network 'ADHD Across the Lifespan'. M.H. was supported by the Netherlands Organisation for Scientific Research (NWO; grants Veni 91619115 and Vidi 09150172210069). B.F. was supported by funding from the EU's Horizon 2020 programme (H2020/2014–2020) under grant agreement 847879 (PRIME) and the NIMH/NIH under award number R01MH124851. M.K. was supported by personal grants from the Dutch Research Council (NWO/ZonMW; Rubicon grant 45219212 and Veni grant 09150162010073). K.L. was supported by the Stanley Center for Psychiatric Research, the US NIMH (U01MH121499), the Simons Foundation Autism Research Initiative (award 735604), the Lundbeck Foundation (R350-2020-963) and the Novo Nordisk Foundation (NNF21SA0072102). S.G. was supported by the Lundbeck Foundation 'ADHD of Mice and Men' (R108-2012-10957). J.G. was supported by NIH (1R01NS131433-01). G.P. was supported by the Simons Foundation Autism Research Initiative (SFARI) Bridge to Independence (BTI) award (00002804). We thank the Broad Institute for generating high-quality sequence data supported by funds from the National Human Genome Research Institute (NHGRI) (grant U54 HG003067) with E. Lander as PI. One of the datasets used in this manuscript was obtained from dbGaP at http://www.ncbi.nlm.nih.gov/gap through dbGaP accession number phs000990. We thank the Center for the Development of Therapeutics at the Broad Institute for assisting with immunofluorescence imaging.

**Author contributions** Analysis: D.D., J.D., Y.-H.H.H., G.P., J.G., T.T.N., J.T., M.M., T.B., F.K.S. and J.H.Y.T. Providing and/or processing samples and/or data: J.B.-G., S.G., M.H., J.B., M.K., G.C.Z., C.J., O.G., M.B., N.F.K., S.K.-S., K.-P.L., B.F., A.R., E.A., T.W., M.N., O.M., P.B.M., K.L., M.J.D., B.M.N. and A.D.B. Writing: D.D., J.D., Y.-H.H.H., G.P., J.G., T.T.N. and A.D.B. Study supervision: D.D., K.L., B.M.N. and A.D.B. Study direction: D.D. and A.D.B. All authors contributed to critical revision of the manuscript.

**Competing interests** D.D. has received speaker fees from Takeda and Medice Nordic. A.D.B. has received speaker fees from Lundbeck. M.H. has received a speaker fee from Medice. S.K.-S. has received speaker honoraria from Medice, Takeda and Janssen. B.F. has received educational speaking fees and travel support from Medice. The remaining authors declare no competing interests.

**Additional information**
**Correspondence and requests for materials** should be addressed to Ditte Demontis or Anders D. Børglum.

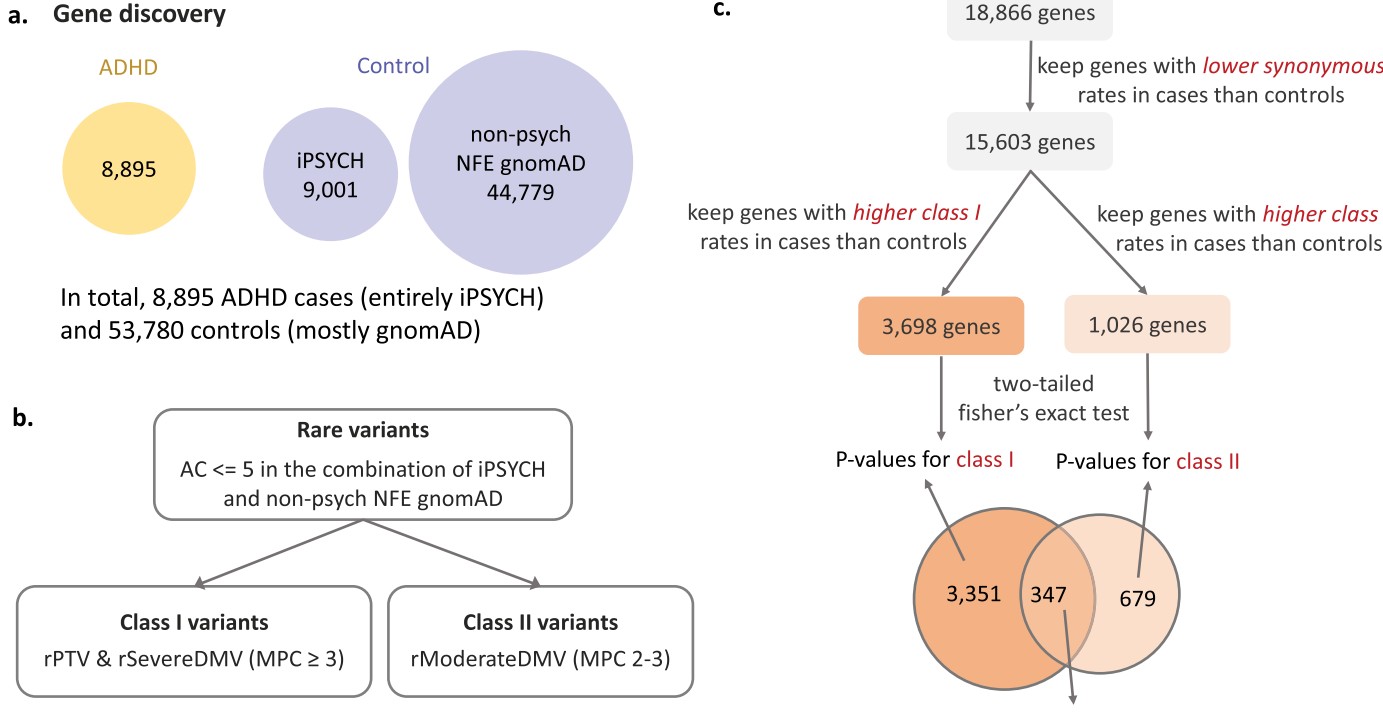

**a.** **Gene discovery**

ADHD

Control

8,895

iPSYCH 9,001

non-psych NFE gnomAD 44,779

In total, 8,895 ADHD cases (entirely iPSYCH) and 53,780 controls (mostly gnomAD)

**b.**

Rare variants

AC <= 5 in the combination of iPSYCH and non-psych NFE gnomAD

Class I variants

rPTV & rSevereDMV (MPC ≥ 3)

Class II variants

rModerateDMV (MPC 2-3)

**c.**

18,866 genes

keep genes with *lower synonymous* rates in cases than controls

15,603 genes

keep genes with *higher class I* rates in cases than controls

keep genes with *higher class II* rates in cases than controls

3,698 genes

1,026 genes

two-tailed fisher's exact test

P-values for class I

P-values for class II

3,351 | 347 | 679

min(combined P-value, P-value for class I, P-value for class II)

**Extended Data Fig. 1 | Overview of study design.** Study design and analytical approach for gene discovery. **a**, Sample sizes used in the gene-discovery analysis (NFE = non-Finnish European). **b**, Definition of rare-variant classes. **c**, Overview of the analytical approach for gene discovery. The numbers of genes illustrated correspond to autosomal genes. The same approach was applied to genes on the X chromosome, and the corresponding results are described in the main text.

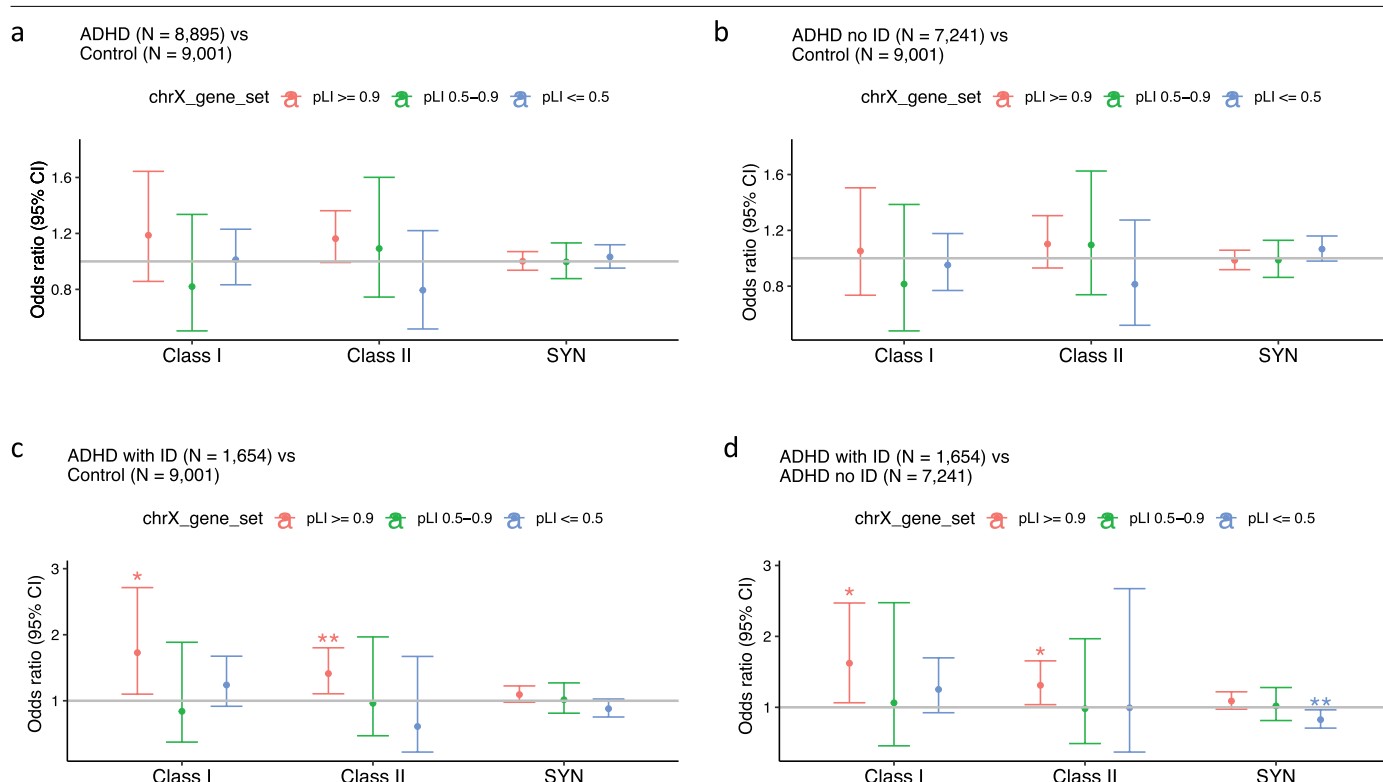

**Extended Data Fig. 2 | ADHD risk across variant categories on the X chromosome.** Burden (odds ratio; OR) of class I, class II and rare synonymous variants (SYN) in chromosome X genes stratified by their pLI score. Results are from logistic regression (results in Supplementary Table 8). **a**, ADHD compared to controls. **b**, ADHD with no intellectual disability (ID) compared to controls. **c**, ADHD with ID compared to controls. **d**, ADHD with ID compared to ADHD without ID. *Indicates nominal significant association $P$ < 0.05, **Indicates significant association after Bonferroni correction correcting for three gene sets ($P$ values less than $P$ = 0.0167 are considered significant).

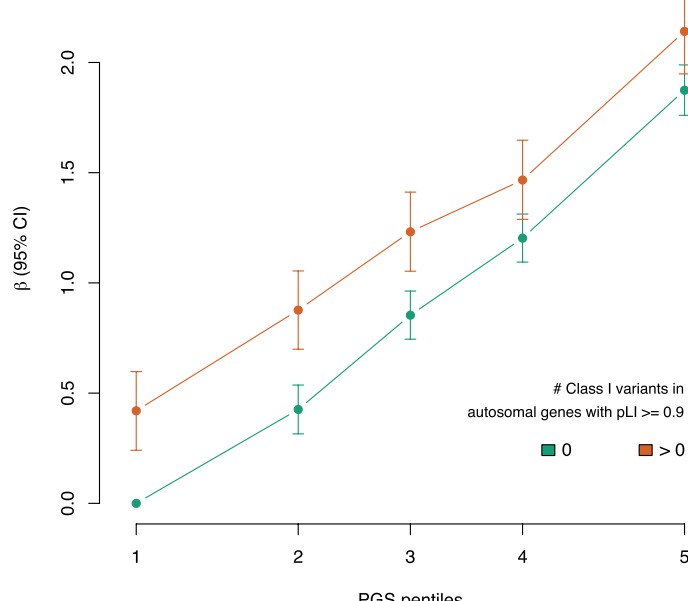

**Extended Data Fig. 3 | Combined effect of common and rare variants on ADHD risk.** Risk for ADHD is given on the y-axis (Dots represent beta coefficient point estimates from logistic regression, and error bars indicate the corresponding 95% confidence intervals [CI]) across ADHD-PGS pentiles on the x-axis, in individuals with no rare class I variants in constrained autosomal genes (pLI ≥ 0.9; marked in green) (n = 14,634 individuals) and in individuals with 1 or more class I variants in constrained autosomal genes (pLI ≥ 0.9; marked in orange) ($n$ = 3,262 individuals). The beta (β) score across groups is relative to the risk in ADHD-PGS pentile one of individuals with no rare class I variants in constrained autosomal genes.

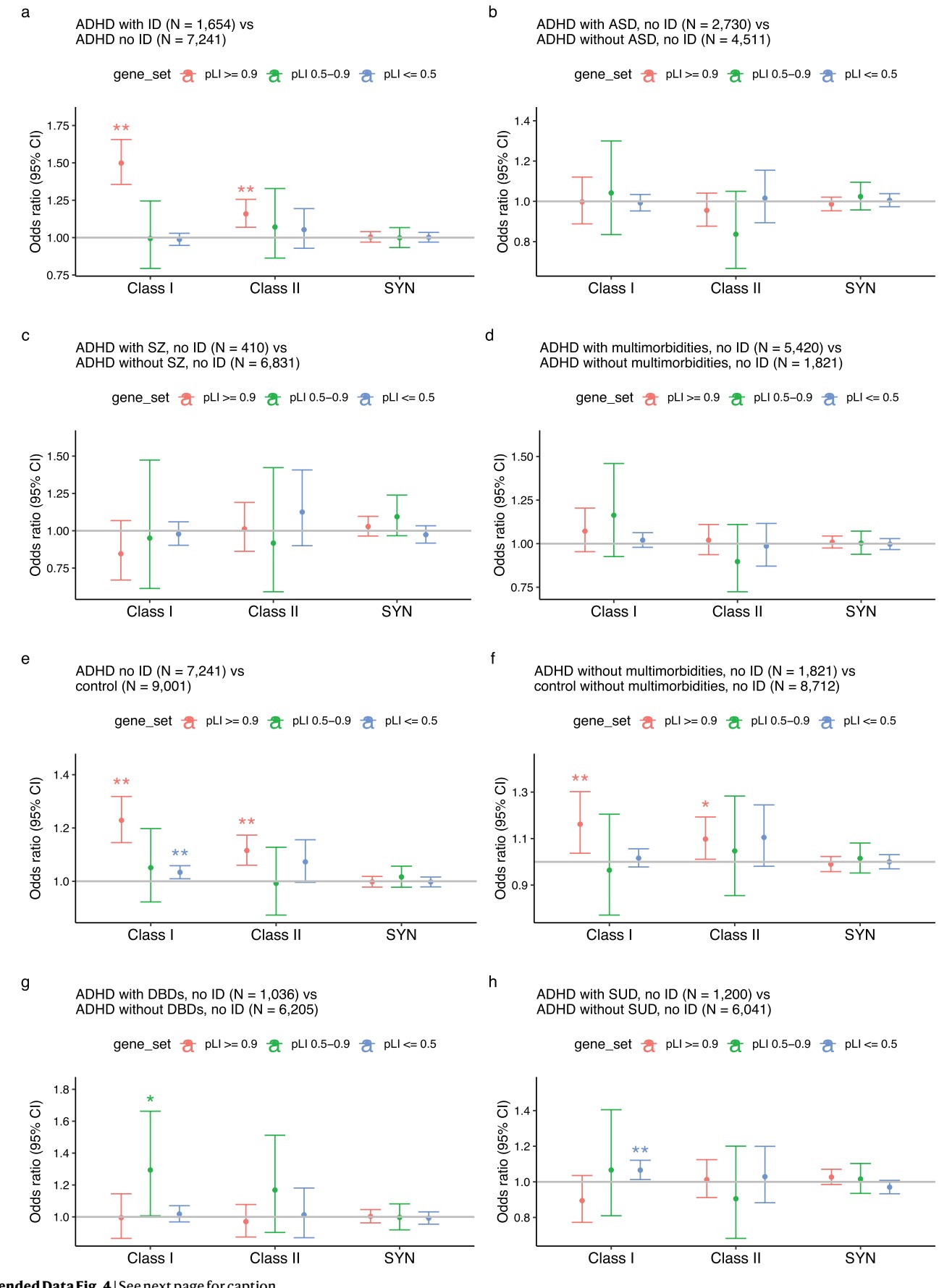

**Extended Data Fig. 4 | See next page for caption.**

**Extended Data Fig. 4 | Effect of rare variants on comorbidity in ADHD across pLI bins.** Burden (odds ratio; OR) of rare class I, class II and rare synonymous variants (SYN) in individuals with ADHD and comorbidities compared to individuals without the comorbid condition being analysed across pLI bins. Results are from logistic regression (detailed results in Supplementary Table 19). **a**, ADHD with intellectual disability (ID) compared to ADHD with no ID. **b**, ADHD comorbid with autism spectrum disorder (ASD) compared to ADHD without ASD. **c**, ADHD comorbid with schizophrenia (SZ) compared to ADHD without SZ. **d**, ADHD with multimorbidities compared to ADHD without multimorbidities. **e**, ADHD with no ID compared to controls. **f**, ADHD without multimorbidities compared to controls without multimorbidities. *Indicates nominal significant association $P < 0.05$, **Indicates significant association after Bonferroni correction (correcting for three gene sets; $P$ values less than $P = 0.0167$ are considered significant).

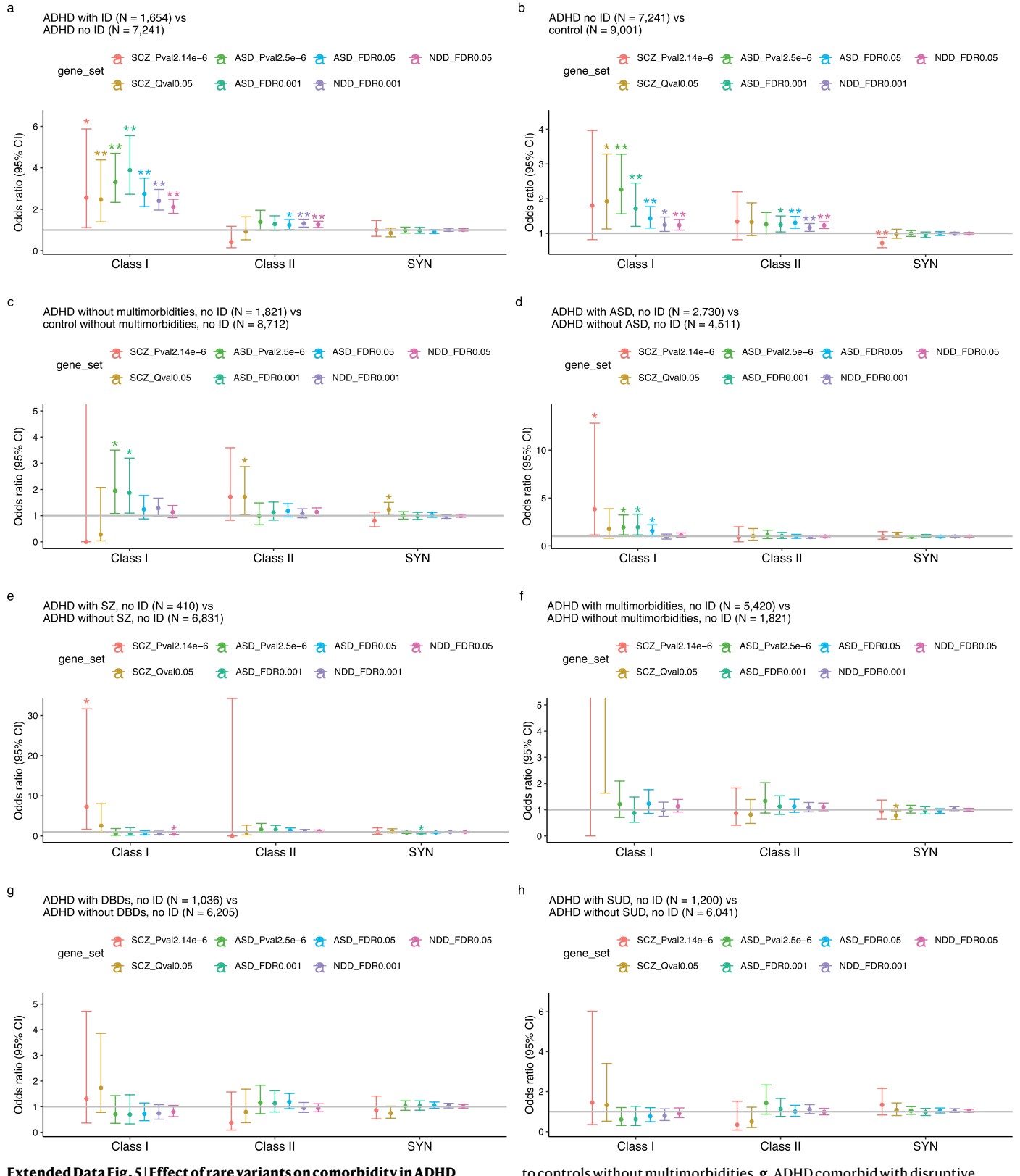

**Extended Data Fig. 5 | Effect of rare variants on comorbidity in ADHD across brain-disorder gene sets.** Burden (odds ratio; OR) of rare class I, class II and rare synonymous variants in individuals with ADHD and comorbidities compared to individuals without the comorbid condition being analysed. **a**, ADHD with intellectual disability (ID) compared to ADHD with no ID. **b**, ADHD comorbid with ASD compared to ADHD without ASD. **c**, ADHD comorbid with schizophrenia (SZ) compared to ADHD without SZ. **d**, ADHD with multimorbidities compared to ADHD without multimorbidities. **e**, ADHD with no ID compared to controls. **f**, ADHD without multimorbidities compared to controls without multimorbidities. **g**, ADHD comorbid with disruptive behaviour disorders (DBS) compared to ADHD without DBS. **h**, ADHD comorbid with substance use disorders (SUD) compared to ADHD without SUD. The loads in gene sets related to autism, schizophrenia and developmental disorders have been evaluated. Results are from logistic regression (Sample sizes can be found in Supplementary Table 18, detailed results in Supplementary Table 21). *Indicates nominal significant association $P < 0.05$, **Indicates significant association after Bonferroni correction correcting for seven gene sets. $P$ values less than $P = 7.14 \times 10^{-3}$ are considered significant.

                           Anders D. Børglum

# Reporting Summary

## Statistics

For all statistical analyses, confirm that the following items are present in the figure legend, table legend, main text, or Methods section.

| n/a | Confirmed | |
|---|---|---|
| ☐ | ☒ | The exact sample size (*n*) for each experimental group/condition, given as a discrete number and unit of measurement |
| ☐ | ☒ | A statement on whether measurements were taken from distinct samples or whether the same sample was measured repeatedly |
| ☐ | ☒ | The statistical test(s) used AND whether they are one- or two-sided<br>*Only common tests should be described solely by name; describe more complex techniques in the Methods section.* |
| ☐ | ☒ | A description of all covariates tested |
| ☐ | ☒ | A description of any assumptions or corrections, such as tests of normality and adjustment for multiple comparisons |
| ☐ | ☒ | A full description of the statistical parameters including central tendency (e.g. means) or other basic estimates (e.g. regression coefficient) AND variation (e.g. standard deviation) or associated estimates of uncertainty (e.g. confidence intervals) |
| ☐ | ☒ | For null hypothesis testing, the test statistic (e.g. *F*, *t*, *r*) with confidence intervals, effect sizes, degrees of freedom and *P* value noted<br>*Give P values as exact values whenever suitable.* |
| ☒ | ☐ | For Bayesian analysis, information on the choice of priors and Markov chain Monte Carlo settings |
| ☒ | ☐ | For hierarchical and complex designs, identification of the appropriate level for tests and full reporting of outcomes |
| ☒ | ☐ | Estimates of effect sizes (e.g. Cohen's *d*, Pearson's *r*), indicating how they were calculated |

*Our web collection on statistics for biologists contains articles on many of the points above.*

## Software and code

Policy information about availability of computer code

| Data collection | Whole-exome seqeuncing was done using Illumina Nextera capture kit and the Illumina HiSeq sequencer. Alignment of sequence reads to the reference genome and and calling of genotypes were done using BWA v0.7.12 and GATK v.3.4. QC was done in Hail v. 0.1 (https://github.com/hail-is/hail), annotation of variants was done using SnpEff v. 4.3t and SnpSift v. 4.3t. |
|---|---|
| Data analysis | Quality control of sequencing data was done in Hail 0.1. Logistic regression analyses were done in R 4.1.3. Gene based burden test in the clinical cohort was done using EPACTS v.3.3.0 Genetic risk enrichment for overlap with common variants were done using MAGMA (v1.09). Limma-based47 (v3.54.2) two-tailed two-sample moderated t-test in the Genoppi R package (development branch, v1.1.0) was used to calculate enrichment statistics in IP data. Enrichment in PPI-networks of rare-variant associations was performed with a one-tailed Kolmogorov-Smirnov (KS) test. A two-sided paired t-test was used to test for differential expression in BrainSPAN v10 data. UMAPs were generated using Python v3.10.0 and the single-cell analysis libraries Scanpy v1.9.1 and AnnData v0.8.0. The "arpack" algorithm implemented in Scanpy was used to perform PCA. scDRS v1.0.2 was calcuated using the code that can be found here: https://github.com/martinjzhang/scDRS. ). Imputation of common variants was done using EAGLE v2.3.5 and Minimac3, and the Haplotype Reference Consortium panel v1.0 as reference. Polygenic scores were generated using the SBayesR algorithm implemented in LDAK v5.2. The c-alpha test implemented in the 'AssotesteR' v0.1-10 R package was used to test for differences in rare variant distributions.. Burden heritability was estimated using the BHR code v0.5.0-alpha that can be found here: https://github.com/ajaynadig/bhr?tab=readme-ov-file. Transcript quantification was performed using Salmon v1.10.2 with GENCODE v43 reference files. Gene-level counts were summarized using tximport v1.26.1, and lowly expressed or non-protein-coding genes were filtered using the filterByExpr function in edgeR v3.40.2. |

For manuscripts utilizing custom algorithms or software that are central to the research but not yet described in published literature, software must be made available to editors and reviewers. We strongly encourage code deposition in a community repository (e.g. GitHub). See the Nature Portfolio guidelines for submitting code & software for further information.

# Data

Policy information about availability of data

All manuscripts must include a data availability statement. This statement should provide the following information, where applicable:
- Accession codes, unique identifiers, or web links for publicly available datasets
- A description of any restrictions on data availability
- For clinical datasets or third party data, please ensure that the statement adheres to our policy

iPSYCH data are available from the authors after approval by the iPSYCH Data Access Committee and can only be accessed on the secured Danish server (GenomeDK https://genome.au.dk) as the data are protected by Danish legislation. For data access and correspondence please contact: Ditte Demontis (ditte@biomed.au.dk) or Anders D. Børglum (anders@biomed.au.dk).

The IP-MS data for MAP1A and ANO8 have been deposited to MassIVE with identifier MSV000098548.

URLs
BrainSpan: www.brainspan.org
gnomad v2.1.1: https://gnomad.broadinstitute.org/downloads#v2-lof-curation-results
gnomAD release 2.1.1: https://gnomad.broadinstitute.org/downloads#v2
La Manno et al. scRNA-seq data (GSE76381_EmbryoMoleculeCounts.cef.txt.gz) were obtained from: https://www.ncbi.nlm.nih.gov/geo/query/acc.cgi?acc=GSE76381
Jerber et al. snRNA-seq datasets were obtained (version v310.5281/zenodo.4651413) from: https://zenodo.org/record/4651413#.ZAcbxXbMJEZ
Silletti et al. snRNA-seq data were obtained from the CZ CELLxGENE platform:
https://datasets.cellxgene.cziscience.com/f9ecb4ba-b033-4a93-b794-05e262dc1f59.h5ad
Pintacuda, G. et al. RNA-seq data were obtained from GEO: GSE178896: https://www.ncbi.nlm.nih.gov/geo/query/acc.cgi?acc=GSE178896
HipSci data browser: https://www.hipsci.org/#/lines

# Research involving human participants, their data, or biological material

Policy information about studies with human participants or human data. See also policy information about sex, gender (identity/presentation), and sexual orientation and race, ethnicity and racism.

| | |
|---|---|
| Reporting on sex and gender | Samples were removed if the sex, derieved based on genetic information was inconsistent with the reported sex in the registries. The study included: females with ADHD (N = 2,265); males with ADHD (N = 6,630); female controls (N = 3,737); male controls (N = 5,264). No difference in the number of class I, class II, or synonymous variants was observed between males and females with ADHD, either before or after excluding individuals with ID. |
| Reporting on race, ethnicity, or other socially relevant groupings | Individuals were grouped according to ICD10 codes obtained from the Danish registries, to test for the impact of rare genetic variants on risk of ADHD and co-morbidities. Individuals with no diagnosis of major psychiatric disorders were grouped as controls. |
| Population characteristics | Participants were born between 1981 and 2008. Psychiatric diagnoses were conferred by the end of 2016 using data from the Danish Psychiatric Central Research Registry and the Danish Patient Registry. Among individuals diagnosed with ADHD (N = 8,895), the mean age at diagnosis was 12.6 years (standard deviation = 6.26). |
| Recruitment | 34,544 individuals from the iPSYCH cohort were selected for whole-exome sequencing, and from these, we included individuals with an ICD-10 diagnosis of ADHD (F90) in the Danish Psychiatric Central Research Registry and the Danish Patient Registry given before or in 2016; individuals with no diagnosis of major psychiatric disorders (autism, bipolar disorder, schizophrenia, and major depressive disorder) were included as controls. Since the information was drawn from national registries and the blood samples were pulled from the Danish Newborn Screening Biobank there is no bias from self-selection. |
| Ethics oversight | The iPSYCH study was approved by the Scientific Ethics Committee in the Central Denmark Region (Case No 1-10-72-287-12) and the Danish Data Protection Agency. In accordance with Danish legislation, the Danish Scientific Ethics Committee has, for this study, waived the need for specific informed consent in biomedical research based on existing biobanks. The clinical data were approved by the Ethics Committee at the University of Würzburg (Würzburg, Germany); in The Netherlands, by the regional Ethics Committee (Commissie Mensgebonden Onderzoek: CMO Regio Arnhem—Nijmegen; Protocol number III.04.0403 and 2014/290; ABR: NL47721.091.14), the Institutional Review Board of the Radboud University Medical Center (Nijmegen, The Netherlands). Participants were included at the department of Psychiatry at the Radboud University Nijmegen Medical Centre in Nijmegen, The Netherlands. All participants in the German and Dutch samples provided signed informed consent in accordance with the Declaration of Helsinki. |

Note that full information on the approval of the study protocol must also be provided in the manuscript.

# Field-specific reporting

Please select the one below that is the best fit for your research. If you are not sure, read the appropriate sections before making your selection.

☒ Life sciences ☐ Behavioural & social sciences ☐ Ecological, evolutionary & environmental sciences

For a reference copy of the document with all sections, see nature.com/documents/nr-reporting-summary-flat.pdf

# Life sciences study design

All studies must disclose on these points even when the disclosure is negative.

| | |
|---|---|
| Sample size | The sample size is based on findings from other studies e.g. in our previous study (Satterstrom et al. Nature Neuroscience, 2019) we analyzed 3,091 individuals with autism spectrum disorder (ASD), 3,206 with ADHD and 684 diagnosed with ASD+ADHD and 5002 ontrols. In the previous study we identifed one significant rare-variant risk gene for ASD+ADHD. In the current study the sample size for individuals with ADHD and controls is higher and thus we expected to have power to identify ADHD risk genes. |
| Data exclusions | We aimed at analyzing genetically homogeneous individuals. Genetic outliers were excluded based on principal component analyses and related individuals were removed (cases preferred kept over controls).Additionally individuals were excluded if the imputed sex did not match the sex in the registries. |
| Replication | Generalizability of our findings was assessed in 1,078 individuals with clinical ascertained persistent ADHD and 1,738 controls. We replicated the finding of increased load of class I variants in cases compared to controls. Top genes from our discovery analysis showed a tendency towards higher effect size point estimates than constrained genes. |
| Randomization | Allocation into groups was not random. Individuals were allocated to a phenotype based on diagnosis codes. |
| Blinding | In iPSYCH, diagnoses are drawn from registries based on ICD10 diagnosis codes. These are administrative data bases populated by data from the clinicians long before the current study. The blood samples are pulled from a biobank. Hence, the study participants and diagnosing clinicians are blinded with respect to this study. Genotyping is done on a massive scale overall for the full iPSYCH cohort, and the data is generated without a specif goal or effect in mind except for an overall goal of investigating the genetic and environmental effects on psychiatric disorders. |

# Reporting for specific materials, systems and methods

We require information from authors about some types of materials, experimental systems and methods used in many studies. Here, indicate whether each material, system or method listed is relevant to your study. If you are not sure if a list item applies to your research, read the appropriate section before selecting a response.

## Materials & experimental systems

| n/a | Involved in the study |
|---|---|
| ☐ | ☒ Antibodies |
| ☐ | ☒ Eukaryotic cell lines |
| ☒ | ☐ Palaeontology and archaeology |
| ☒ | ☐ Animals and other organisms |
| ☒ | ☐ Clinical data |
| ☒ | ☐ Dual use research of concern |
| ☒ | ☐ Plants |

## Methods

| n/a | Involved in the study |
|---|---|
| ☒ | ☐ ChIP-seq |
| ☒ | ☐ Flow cytometry |
| ☒ | ☐ MRI-based neuroimaging |

## Antibodies

| | |
|---|---|
| Antibodies used | Each antibody is listed with the following information in order: Gene ID: Vendor and catalog number; Host and clonality; Usage; Dilution used in immunofluorescence (IF).<br><br>NeuN: Abcam, ab104224; Mouse Monoclonal; Primary; 1:1000<br><br>MAP2: Abcam, ab183830; Rabbit Polyclonal; Primary; 1:1000<br><br>CUX2: Proteintech, 82933-1-RR; Rabbit Polyclonal; Primary; 1:1000<br><br>TUJ1: Invitrogen, MA1-118; Mouse Monoclonal; Primary; 1:1000<br><br>TUJ1: Invitrogen, 53-4510-82; Mouse Monoclonal; Conjugated Antibody; 1:500<br><br>Homer1: Synaptic Systems, 160003; Rabbit Polyclonal; Primary; 1:1000<br><br>Synaptophysin: Synaptic Systems, 101308; Guinea Pig; Primary; 1:1000<br><br>Vimentin: Abcam, ab24525; Rabbit Polyclonal; Primary; 1:1000<br><br>MSI1: Proteintech, 13512-1-AP; Rabbit Polyclonal; Primary; 1:1000<br><br>MAP2: Abcam, ab302547; Rabbit Monoclonal; Conjugated Antibody; 1:500<br><br>MAP2: Abcam, ab225316; Rabbit Monoclonal; Conjugated Antibody; 1:500 |

NeuN: NBP1-77686AF488; Rabbit Polyclonal; Conjugated Antibody; 1:500

Goat Anti-Rabbit IgG H&L (Alexa Fluor® 488): Invitrogen, A-11008; Goat Polyclonal; Secondary Antibody; 1:3000

Goat Anti-Mouse IgG H&L (Alexa Fluor® 488): Invitrogen, A28175; Goat Recombinant Superclonal; Secondary Antibody; 1:3000

Goat Anti-Rabbit IgG H&L (Alexa Fluor® 594): Invitrogen, A-11012; Goat Polyclonal; Secondary Antibody; 1:3000

Goat Anti-Mouse IgG H&L (Alexa Fluor® 594): Invitrogen, A-11032; Goat Polyclonal; Secondary Antibody; 1:3000

Goat anti-Chicken IgY (H+L) Secondary Antibody, Alexa Fluor® 647: Invitrogen, A-21449; Goat Polyclonal; Secondary Antibody; 1:3000

Goat anti-Guinea Pig IgG (H+L) Highly Cross-Adsorbed Secondary Antibody, Alexa Fluor® 488: Invitrogen, A-11073; Goat Polyclonal; Secondary Antibody; 1:3000

Goat anti-Guinea Pig IgG (H+L) Highly Cross-Adsorbed Secondary Antibody, Alexa Fluor® 647: Invitrogen, A-21450; Goat Polyclonal; Secondary Antibody; 1:3000

Validation

All antibodies used in this study have been validated by the respective manufacturers as indicated below
NeuN Abcam, ab104224, NeuN Immunocytochemistry/ Immunofluorescence staining of rat brain neurons using mouse Anti-NeuN antibody Rat brain neural cultures stained with ab104224 in pink with ab4674 (chicken polyclonal to GFAP) in green and DNA in blue. ab104224 reveals strong nuclear and distal cytoplasmic staining. It does not stain astrocytes and other non-neuronal cells.

MAP2 Abcam, ab183830, Immunofluorescent analysis of 4% Paraformaldehyde-fixed, 0.1% TritonX-100 permeabilized rat primary neural/glia cells labelling MAP2 with ab183830 at 1/1000 dilution, followed by ab150077 Goat Anti-Rabbit IgG H&L (Alexa Fluor® 488) antibody at 1/1000 dilution (2 µg/mL) (Green). Confocal image showing positive staining in rat primary neuron cell. Confocal scanning Z step was set as 0.3 µm followed by image processing with maximum Z projection. ab11267 Anti-MAP2 mouse monoclonal antibody was used to counterstain tubulin at 1/500 dilution (4 µg/mL) followed by ab150120 Goat Anti-Mouse IgG H&L (Alexa Fluor® 594) at 1/1000 dilution (2 µg/mL) (Red). The Nuclear counterstain was DAPI (Blue).

CUX2 Proteintech, 82933-1-RR, Immunofluorescent analysis of (4% PFA) fixed HeLa cells using CUX2 antibody (82933-1-RR, Clone: 230235G1 ) at dilution of 1:300 and CoraLite®488-Conjugated AffiniPure Goat Anti-Rabbit IgG(H+L).

TUJ1 Invitrogen, MA1-118, Immunofluorescence analysis of beta III tubulin (green) in the ectoderm derived from human ES cells. Embryoid bodies (EBs) were generated from the H9 embryonic stem cell line (WiCell Research Institute, WA09) using Gibco® KnockOut™ Serum Replacement. After four days in suspension culture, EBs were plated on Geltrex™-coated tissue culture-treated polystyrene and continuously cultured for 21 days. EB cultures were then fixed and permeabilized according to the 3-Germ Layer Immunocytochemistry Kit (Product # A25538) and stained with anti-beta III-tubulin monoclonal antibody (Product # MA1-118, 1:200 dilution, 5 uL/mL final) at 4°C overnight. Secondary staining was completed using Alexa Fluor™ 488-conjugated anti-mouse IgG (Product # A-11001) and DAPI (Product # D1306) for nuclear DNA (blue) for 1 h at room temperature. Images were taken on EVOS® FL Auto Imaging System at 10X magnification.

TUJ1 Invitrogen, 53-4510-82, Knockout of beta-3 Tubulin (TUBB3) was achieved by CRISPR-Cas9 genome editing. Immunofluorescence analysis was performed on wild type U-87 MG cells (panel a,d), U-87 MG Cas9 cells (panels b,e) and U-87 MG beta-3 Tubulin KO cells (panel c,f). Cells were fixed, permeabilized, and labelled with beta-3 Tubulin Monoclonal Antibody, Alexa Fluor® 488 (Product # 53-4510-82) (10 µg/mL). Nuclei (blue) were stained using ProLong™ Diamond Antifade Mountant with DAPI (Product # P36962), and Rhodamine Phalloidin (Product # R415) (1:300) was used for cytoskeletal F-actin (red) staining. Loss of signal (panel c,f) upon CRISPR mediated knockout (KO) confirms that antibody is specific to beta-3 Tubulin (green). The images were captured at 60X magnification.

Homer1 Synaptic Systems, 160003, Immunostaining of a hippocampus neuron with anti-homer (dilution 1 : 500, red) and anti-synaptophysin (cat. no. 101 011, dilution 1 : 500, green). Positive clusters (red) can be found on the postsynaptic neuron juxtaposed to presynaptic nerve terminals (green).

Synaptophysin Synaptic systems, 101308, Indirect immunostaining of PFA fixed rat hippocampus neurons with Guinea pig anti-Synaptophysin 1 (cat. no. 101 308, dilution 1 : 1000, red) and rabbit anti-MAP 2 (cat. no. 188 002, dilution 1 : 1000, green). Nuclei have been visualized by DAPI staining (blue).

Vimentin Abcam, ab24525, Immunocytochemistry/ Immunofluorescence analysis of neuron/glial cultures labeling Vimentin with ab24525 (green) and GFAP with ab7260 (red). Vimentin is the sole cytoplasmic intermediate filament subunit expressed in fibroblasts, microglial and endothelial cells. The flattened cells in the middle of the image which appear green are fibroblasts. Astrocytes may express primarily GFAP, or both GFAP and vimentin, and so appear red (GFAP only) or golden yellow (GFAP and Vimentin). In cells which express both GFAP and vimentin, the two proteins assemble to produce heteropolymer filaments.

MSl1 Proteintech, 13512-1-AP, Immunofluorescent analysis of (4% PFA) fixed HeLa cells using 13512-1-AP (SNRPB2 antibody) at dilution of 1:50 and Alexa Fluor 488-conjugated AffiniPure Goat Anti-Rabbit IgG

MAP2 Abcam, ab302547, Immunohistochemical analysis of 4% PFA-fixed, 0.2% Triton X-100 permeabilized frozen Mouse cerebellum (fresh) tissue labeling MAP2 with ab302547 at 1/100 (5.0 ug/ml) dilution (Green). Positive staining on mouse cerebellum is observed. The nuclear counterstain was DAPI (Blue).

MAP2 Abcam, ab225316, IHC image of MAP2 staining in a section of frozen normal human cerebral cortex. The section was fixed using 10% formaldehyde in 1XPBS for 10 minutes. No antigen retrieval step was performed prior to staining. Non-specific protein-protein interactions were then blocked in TBS containing 0.025% (v/v) Triton X-100, 0.3M (w/v) glycine and 1% (w/v) BSA for 1h at

room temperature. The section was then incubated overnight at +4°C in TBS containing 0.025% (v/v) Triton X-100 and 1% (w/v) BSA with ab225316 at 1/100 dilution (shown in green) and counterstained using ab195884, Rat monoclonal to Tubulin (Alexa Fluor® 647), at 1/250 dilution (shown in red). Nuclear DNA was labelled with DAPI (shown in blue).

Neun NBP1-77686AF488, RBFOX3/NeuN was detected in immersion fixed U-2 OS human osteosarcoma cell line using Rabbit anti-RBFOX3/NeuN Affinity Purified Polyclonal Antibody conjugated to Alexa Fluor® 488 (Catalog # NBP1-77686AF488) (green) at 10 μg/mL overnight at 4C. Cells were counterstained with DAPI (blue). Cells were imaged using a 100X objective and digitally deconvolved

Goat Anti-Rabbit IgG H&L (Alexa Fluor® 88) Invitrogen, Immunofluorescence analysis of Goat anti-Rabbit IgG (H+L) Cross-Adsorbed Secondary Antibody Alexa Fluor® 488 conjugate was performed using HeLa cells stained with alpha Tubulin Rabbit Polyclonal Antibody (Product # PA5-16891). The cells were fixed with 4% paraformaldehyde for 10 minutes, permeabilized with 0.1% Triton™ X-100 for 10 minutes, blocked with 1% BSA for 1 hour and labeled with 2 μg/mL Rabbit primary antibody for 3 hours at room temperature. Goat anti-Rabbit IgG (H+L) Cross-Adsorbed Secondary Antibody Alexa Fluor® 488 conjugate (Product # A-11008) was used at a concentration of 4 μg/mL in phosphate buffered saline containing 0.2% BSA for 45 minutes at room temperature, for detection of alpha Tubulin in the cytoplasm (Panel a: green). Nuclei (Panel b: blue) were stained with DAPI in SlowFade® Gold Antifade Mountant (Product # S36938). F-actin was stained with Rhodamine Phalloidin (Product # R415, 1:300) (Panel c: red). Panel d represents the composite image. No nonspecific staining was observed with the secondary antibody alone (panel f), or with an isotype control (panel e). The images were captured at 60X magnification.

Goat Anti-Mouse IgG H&L (Alexa Fluor® 488), Invitrogen, Immunofluorescence analysis of Goat anti-Mouse IgG (H+L) Secondary Antibody Alexa Fluor® 488 conjugate was performed using HeLa cells stained with alpha Tubulin (236-10501) Mouse Monoclonal Antibody (Product # A11126). The cells were fixed with 4% paraformaldehyde for 10 minutes, permeabilized with 0.1% Triton™ X-100 for 10 minutes, blocked with 1% BSA for 1 hour and labeled with 2 μg/mL Mouse primary antibody for 3 hours at room temperature. Goat anti-Mouse IgG (H+L)/IgM (L) Secondary Antibody Alexa Fluor® 488 conjugate (Product # A28175) was used at a concentration of 1 μg/mL in phosphate buffered saline containing 0.2% BSA for 45 minutes at room temperature, for detection of alpha Tubulin in the cytoplasm (Panel a: green). Nuclei (Panel b: blue) were stained with DAPI in SlowFade® Gold Antifade Mountant (Product # S36938). F-actin was stained with Rhodamine Phalloidin (Product # R415, 1:300) (Panel c: red). Panel d represents the composite image. No nonspecific staining was observed with the secondary antibody alone (panel f), or with an isotype control (panel e). The images were captured at 60X magnification.

Goat Anti-Rabbit IgG H&L (Alexa Fluor® 594), Invitrogen, Immunofluorescence analysis of Goat anti-Rabbit IgG (H+L) Cross-Adsorbed Secondary Antibody, Alexa Fluor 594 (Product # A-11012) was performed using HeLa cells stained with alpha Tubulin Rabbit Polyclonal Antibody (Product # PA5-16891). The cells were fixed with 4% paraformaldehyde for 10 minutes, permeabilized with 0.1% Triton™ X-100 for 10 minutes, blocked with 1% BSA for 1 hour and labeled with 2 μg/mL of rabbit primary antibody for 3 hours at room temperature. Goat anti-Rabbit IgG (H+L) Cross-Adsorbed Secondary Antibody, Alexa Fluor 594 (A-11012) was used at a concentration of 2 μg/mL in phosphate buffered saline containing 0.2 % BSA for 45 minutes at room temperature, for detection of alpha Tubulin in the cytoplasm (Panel a: red). Nuclei (Panel b: blue) were stained with DAPI in SlowFade® Gold Antifade Mountant (Product # S36938). F-actin was stained with Alexa Fluor® 488 Phalloidin (Product # A12379, 1:300) (Panel c: green). Panel d represents the composite image. No nonspecific staining was observed with the secondary antibody alone (panel f), or with an isotype control (panel e). The images were captured at 60X magnification.

Goat Anti-Mouse IgG H&L (Alexa Fluor® 594), Invitrogen, Immunofluorescence analysis of Goat anti-Mouse IgG (H+L) Highly Cross-Adsorbed Secondary Antibody, Alexa Fluor® 594 conjugate was performed using HeLa cells stained with alpha Tubulin (236-10501) Mouse Monoclonal Antibody (Product # A11126). The cells were fixed with 4% paraformaldehyde for 10 minutes, permeabilized with 0.1% Triton™ X-100 for 10 minutes, blocked with 1% BSA for 1 hour and labeled with 2 μg/mL primary antibody for 3 hours at room temperature. Goat anti-Mouse IgG (H+L) Highly Cross-Adsorbed Secondary Antibody, Alexa Fluor® 594 (Product # A-11032) was used at a concentration of 2 μg/mL in phosphate buffered saline containing 0.2% BSA for 45 minutes at room temperature, for detection of alpha Tubulin in the cytoplasm (Panel a: red). Nuclei (Panel b: blue) were stained with DAPI in SlowFade® Gold Antifade Mountant (Product # S36938). F-actin was stained with Alexa Fluor® 488 Phalloidin (Product # A12379), 1:300) (Panel c: green). Panel d represents the composite image. No nonspecific staining was observed with the secondary antibody alone (panel f), or with an isotype control (panel e). The images were captured at 60X magnification.

Goat anti-Chicken IgY (H+L) Secondary Antibody, Alexa Fluor® 647, Invitrogen, Verification of vagus nerve Schwann cell-specific Mpz-EPOR-KO in mice. A Characterization of mouse vagus nerve derived Schwann cells (VNSCs). The identity and purity of SCs were confirmed using IF staining of S100, p75NTR, and Mpz under a fluorescent microscope (ZEISS Apotome 2). The purity of the cultured SCs (99%) was analyzed by double positive staining of DAPI with S100/p75NTR/Mpz markers from 3 independent experiments. Each image represents 3 images from 3 independent experiments. Scale bar: 50A I 1/4m, naEUR%0=aEUR%03. B PCR genotyping with DNA isolated from the vagus nerve and VNSCs (passage zero, SC-P0; passage one, SC-P1) revealed the presence of the 220-bp PCR product, resulting from MpzCre mediated recombination in the Schwann cell of MpzCre-EPORflox/flox mice and not in wild-type (control; 390A bp) and flox/flox (428A bp) mice (naEUR%0=aEUR%03) - Image collected and cropped by CiteAb under a CC-BY license from the following publication: A critical role for erythropoietin on vagus nerve Schwann cells in intestinal motility. BMC Biotechnol (2023) Image collected and cropped by CiteAb from the following publication (https://pubmed.ncbi.nlm.nih.gov/37127673), licensed under a CC BY license.

Goat anti-Guinea Pig IgG (H+L) Highly Cross-Adsorbed Secondary Antibody, Alexa Fluor® 488, Invitrogen, KRT82 is expressed exclusively in anagen.a IF of KRT82 (green) in postnatal (p) mouse skin. Mouse HFs are in telogen at postnatal days 22 and 90 (p22, p90), in anagen at p31, and catagen at p43. Cuticular staining of KRT82 is only observed in the p31 (anagen) HF. White dashed line outlines the hair shaft cuticle. Weak staining observed in mouse epidermis may be a result of green autofluorescent properties of skin epidermal cells. IF staining was repeated at 3 other anagen timepoints and 2 additional telogen timepoints with similar results. b Western blot analysis of whole mouse skin at varying stages of the hair cycle shows that keratin protein is only present in the anagen phase (p31) of the mouse hair cycle. Western blot was repeated with similar results 2 additional times. Source data are provided as a Source data file. Image collected and cropped by CiteAb from the following publication (https://pubmed.ncbi.nlm.nih.gov/35145093), licensed under a CC BY license.

Goat anti-Guinea Pig IgG (H+L) Highly Cross-Adsorbed Secondary Antibody, Alexa Fluor® 647, Invitrogen, Identification of PPNs and of the recorded PPNs that were immunostained for MOR. (A1) sacral parasympathetic nucleus (SPN) located in the mediolateral border

of the gray matter that was identified under a lower magnification with fluorescent illumination. (A2) tetramethylrhodamine-dextran (TMR)-labeled PPNs were identified at a higher magnification with fluorescent illumination. (A3) The neuron shown in (A2), but viewed with infrared illumination during whole-cell recording. (B1,C1,D1) The recorded neuron shown in (A2,A3), filled with biocytin and visualized with FITC-conjugate avidin (green) at various magnifications. (B2,C2,D2) The same section shows TMR immunoreactivity (Alexa 594) and is shown at the same magnification as in (B1,C1,D1), respectively. (B3,C3,D4) Merged images show that the biocytin-filled neuron was the TMR-containing neuron. (D3) The neuron also shows MOR-Immunoreactive (IR) (Alexa 647). (D4) Merged image shows that the biocytin-filled neuron was the TMR-containing neuron that exhibited MOR-IR. Scale bars: 7 μm in (B); 20 μm in (C); 100 μm in (D). Image collected and cropped by CiteAb from the following publication (https://pubmed.ncbi.nlm.nih.gov/26074773), licensed under a CC BY license.

MAP1 Fortis A301-444A, Detection of human MAP1A by western blot of immunoprecipitates. Samples: Whole cell lysate (1 mg for IP, 20% of IP loaded) from HeLa cells. Antibodies: Affinity purified rabbit anti-MAP1A antibody A301-444A used for IP at 3 μg/mg lysate. MAP1A was also immunoprecipitated by rabbit anti-MAP1A antibody A301-445A, which recognizes a downstream epitope. For blotting immunoprecipitated MAP1A, A301-445A was used at 1 μg/ml. Detection: Chemiluminescence with an exposure time of 10 seconds.

ANO8 Biorbyt orb394715, mouse brain tissue were subjected to SDS PAGE followed by western blot with orb394715 (ANO8 antibody) at dilution of 1:500.

Rabbit  IgG polyclonal Abcam ab37415, validated by WB in human liver, mouse brain and rat kidney tissue lysates and HeLa, HepG2 and MCF7 whole cell lysates, for ChIP, with ChIP-qPCR for H3K36me2 on the BTF3 locus in NTKO and TKO cells. Methylation enrichment was tested 5 kb upstream of the TSS (5') and at the TSS (right). ab37415 was used as the isotype contol.

# Eukaryotic cell lines

Policy information about cell lines and Sex and Gender in Research

| Cell line source(s) | We used an induced pluripotent stem cell (iPSC) line (iPS hDFn 83/22 iNgn2#9 [iPS3]) from a neurotypical male donor. |
|---|---|
| Authentication | The cell line was not authenticated. |
| Mycoplasma contamination | Cell lines were tested negative for mycoplasma contamination as per cell culture protocols of the Broad Institute |
| Commonly misidentified lines (See ICLAC register) | No commonly misidentified cell lines were used |

# Plants

| Seed stocks | n/a |
|---|---|
| Novel plant genotypes | n/a |
| Authentication | n/a |

