## [Peer Review File · Nature]

Rare coding variants confer high ADHD risk and implicate neuronal biology

Corresponding Author: Professor Ditte Demontis

Version 0:

Reviewer comments:

Referee #1

(Remarks to the Author)

Review of "Rare coding genetic variants confer high risk of ADHD, implicate neuronal biology, and impact socioeconomic outcomes"

The authors present the largest study to date on the contributing of rare coding variants on ADHD risk. Data are primarily from iPsych (8,895 cases and 53,780 controls, although additional analyses were performed in smaller cohorts). The findings are of interest, e.g., the authors show that rare and common variants make independent contributions to ADHD risk. They argue that although rare variants explain less of the genetic variance compared to common variants, effect sizes of rare variants are large (nicely illustrated in Figure 2, facilitating the identification of biological mechanisms and causal genes). The analyses seem sound and the results present effect sizes and standard errors (the latter are quite large for some results, implying statistical power is sufficient but could be better for some of the analyses). The manuscript is generally well written (some suggestions for improvement are below). I have one major concern/question for the authors, and this relates to the representativeness of the iPsych sample for other populations of ADHD cases (see point 1 below). These are my main comments/suggestions:

- 1) As far as I'm aware, The iPsych sample is mostly based on psychiatric/medical records. I am wondering whether this selection, is biased towards the more severe cases of ADHD? I noticed that the prevalence of Intellectual Disability is close to 20% and the prevalence of ASD is even higher. These prevalences seem high to me. Can the authors comment on how this sample compares to "population" cohorts of ADHD cases and on whether they expect the contribution of rare coding variants to be stronger in iPsych cases because of the specific selection that was used?
- 2) I find the description of class I and class II variants a little confusing in the main text. Can the distinction be shown in a Figure? This could easily be added to Figure 1, maybe as a separate panel or at least described in the figure legends.
- 3) Line 92, I don't believe schizophrenia and bipolar disorder would be considered as neurodevelopmental disorders
- 4) Is there a reason why the X chromosome hasn't been included? (line 120)
- 5) Line 328: how robust is this association? Since overall enrichment is not significant, can these 4 nominally significant associations be due to type-II error?
- 6) Line 334: I assume iPsych was excluded from the discovery GWAS? What was the discovery GWAS? Briefly explain here please
- 7) I believe Figure 1 can be improved: Check colors of Panel A, these seem incorrect. Figure should also define class II variants.

Minor:

- please define the abbreviation rPTVs when it is used for the first time
- line 148: should this read rPTVs in constrained genes?
- Line 184: should this read class I or class II variants (instead of and)
- Line 193: this imply should read this implied
- Line 216: 5.2% of the class I or class II heritability?
- Line 302: What does the OR represent? What is the reference group?

Referee #2

(Remarks to the Author)

In this manuscript, Demontis, Duan, and colleagues perform a rare variant based approach to implicate three new genes in ADHD. As far as I can tell, this is the largest rare variant study in ADHD performed to date. First, the authors recapitulate well-established findings that ADHD cases are enriched for rare protein-coding variants in intolerant/constrained genes. They then perform gene-based burden tests. The novelty in this study mostly lies in the discovery of three novel risk genes: ANO8, ANK2, and MAP1A. Unfortunately, the authors chose to exclude the X chromosome from their analyses, despite the well-established importance of this chromosome in neurodevelopmental disease. The authors also present some experimental data (IP-MS in iPSC-derived NPCs and excitatory neurons) to further interrogate these novel loci, though these experiments would benefit from additional validation, as detailed in my comments below.

Major comments:

- A recent paper in Nat Comms (PMID: 38997333) performed a large scale de novo mutation analysis in ADHD, identifying KDM5B as a risk gene. The authors should cite this paper in their revision. Also, can the authors comment on whether the KDM5B association replicates in their dataset, and vice versa, whether ANO8, ANK2, and MAP1A show any enrichment in the de novo study? This would provide nice orthogonal evidence.
- The authors show that there is an enrichment of rare PTVs and missense variants in constrained genes in fig 1a. They show that there is no enrichment of rare synonymous variants (rSYN) in constrained genes, which is good. However, it is strange that they didn't also show the distribution of rSYN variants exome-wide as they did for other the other variant consequences. It seems to me that this should be included as another negative control.
- There still appears to be inflation even after excluding the ~3000 genes with increased syn variation in cases vs. controls (Supp. Figure 12A). What was the lambda?
- The description of the cohort demographic could use more details. In particular, what was the ancestral breakdown of the fully sequenced cohort, prior to sample selection? How many individuals were from non-European ancestries? Excluding non-European individuals is not ideal. Furthermore, what was the sex breakdown of controls?
- I am a little confused about the gene burden test performed. Am I interpreting it correctly that they only performed the burden test on the genes that had higher rates of damaging variants in cases vs. controls? The authors ought to perform an exome-wide analysis and include all the summary stats in the supplement. 1) This would be helpful to potentially identify protective signals, and 2) would allow the community to perform meta-analyses in future work.
- It is a shame that the authors excluded the X chromosome, particularly since there are genes on this chromosome that clearly contribute to neurodevelopmental disease. I strongly urge the authors to include the X chromosome in the revised manuscript.
- As far as I can tell, I don't think the authors included a recessive model, which also seems like a missed opportunity.
- The replication analysis is unconvincing. It is unclear whether the authors attempted to replicate the MAP1A, ANO8, and ANK2 associations. Even if not nominally significant, do these genes show effect sizes in a consistent direction? It is possible that they are limited by statistical power, but if this is the case, the authors should at least include a power analysis. Alternatively, the authors could perform a meta-analysis across the two cohorts.
- Per the supp. note, the controls in the replication dataset are also in gnomAD, meaning it's not truly an independent set. Why not pull controls from another publicly available datasets (UK Biobank, TopMED, All of US, etc.)?
- The IP-MS studies are a nice addition to the paper. Given how prone IP-MS can be to false positives, I urge the authors to knockdown or knock out the three proteins as negative controls.
- The authors should perform immunofluorescence to demonstrate the purity of their NPC and ExN populations.
- Why do the authors switch from defining constrained genes as pLI < 0.9 to LOEUF < 0.6 for the PPI analyses? I suggest they stick to one definition or test both sets in each constrained gene analysis (including fig. 1).
- For any type of gene ontology analysis, it is critical to define a background set. How did the authors define a background set for their PPI analysis? Unless they are including genes expressed in neurons at baseline, it's hard to interpret.
- Why did the authors only focus on midbrain regions for their single-cell enrichment analysis? Given the number of cortical single-cell datasets (both from human embryos, post-mortem, and iPSC-derived neurons), I'd urge the authors to consider these regions as well.
- In addition to showing the single-cell and BrainSpan expression patterns for the top 100 genes, I recommend the authors specifically plot the expression patterns of the three significant genes.

• Finally, I think the authors could provide more details on the phenotypes of MAP1A, ANO8, and ANK2 carriers. For example, were these individuals enriched for more severe forms of disease/early-onset disease?

Minor comments

- There are lots of different cohorts/analyses in this paper, and it becomes a bit hard to follow. I would recommend a schematic to show the overall study design.
- The main text lists Supp Table 8 as the replication analysis table, but this seems incorrect.
- I would suggest you include the summary stats for each gene MAP1A, ANO8, and ANK2 in the main text for readability.
- I was intrigued that ANO8 had way fewer interactions than ANK2 and MAP1A. Do the authors have any idea why this might be?
- Figures 5 and 6 would benefit from having color legends in the figures themselves.

Referee #3

(Remarks to the Author)

A. Summary of key results: This article presents results from an analysis of exome sequence data from individuals diagnosed with ADHD (n ~ 9K) and controls (n ~ 54K). They identify three genes with rare deleterious variants that are associated with substantially elevated odds of ADHD diagnosis and conduct a series of bioannotation and follow-up analyses to probe biological mechanisms implicated by these genes.

B. Originality and significance: Understanding the genetic etiology of ADHD is a topic of substantial public health significance, but the significance of the question is not very well described in the paper. Right now, the Introduction reads as if it is written for a genetics or psychiatric journal; that is, it is written as if the audience is already familiar with the public health impact of ADHD and the importance of genetic studies. ADHD is often trivialized/stigmatized as merely a label for "bad kids" rather than a psychiatric condition that results not just in lower education and lower socioeconomic attainments, but also with injuries, criminal justice system involvement, earlier mortality, etc. There is a need for novel and more effective pharmacologies with lower potential for abuse. All of that to say, I am sold on the significance of this research study, but I wish the authors would make the case of its importance more clearly.

Instead, they present the rationale for studying rare variation as "necessary to explain the heritability more fully." In the very next sentence, they report that rare variants have been found to explain only small fractions of the population variance in SZ and BD, so they likely knew going into this project that they were not going to substantially account for the population heritability with rare variants. The rationale for why these analyses are important needs to be described more effectively.

(On a more minor note, I would caution the authors against using "genetic" (second sentence of the Introduction) as a synonym for "heritable", especially in a general interest journal.)

Regarding originality, these results extend their previous analysis of rare variants in ADHD individuals from the iPSYCH cohort (2019 paper in Nature Neuroscience). They have added more ADHD cases (a little more than doubled) and continue to use gnomAD as a control data set. The 2019 paper also identified MAP1A as an exome-wide significant gene in relation to ADHD, which seems to contradict their claim that this paper identifies "the first three genes implicated in ADHD by rare coding variants from exome sequencing."

C. Data and methodology: These analyses use Danish register data to identify adults who have been diagnosed with ADHD and compared them to adults with no major psychiatric disorder diagnosis within the same cohort, plus other controls from gnomAD, which removes samples that have been in psychiatric studies. My major question is to what extent they are analyzing the genetic etiology of ADHD, specifically. They conduct several tests of whether their results are driven by comorbid ID, some additional tests for ADHD comorbid with autism or schizophrenia – but none of these are the psychiatric conditions most commonly comorbid with ADHD. Twin studies consistently find that about $\frac{1}{3}$ to $\frac{1}{2}$ of the heritable variation in ADHD is shared with disruptive behavior disorders of childhood (CD, ODD) and with symptoms of ASPD and SUDs in adulthood. Genomic SEM models have found converging results. Is this paper an analysis of ADHD, or of disinhibited/disruptive behavior disorders more broadly, of which ADHD is just the most readily available indicator in register data?

Relatedly, they analyze the association between rPTVs and socioeconomic attainments within individuals with ADHD – but some have argued that ADHD is under-diagnosed, particularly among girls/women who are more likely to have inattentive compared to hyperactive subtypes, and among those whose school performance does not fall below what is "typical" for their age. Have the authors considered analyzing the relationship between ADHD-associated rPTVs and socioeconomic status in individuals who have not been diagnosed with ADHD (or in the full sample regardless of ADHD diagnosis)? Are there any sex differences in this relationship?

D. Statistical tests and treatment of uncertainties: The authors conduct many statistical tests and correct for multiple testing, throughout. My question concerns their "ADHD gene discovery" analyses. They restricted their analyses to consider only genes with a higher rate of variants in cases versus controls, which was 24% of genes for "class I" variants and 7% of genes for "class II" variants – because they expected was that variants that disrupt biological processes would be increased in cases compared to controls. Perhaps I'm just misunderstanding something basic here, but this feels like selecting the data that fits your hypothesis before you test your hypothesis. (If I want to prove that needles are more common in haystack #1 versus haystack #2, wouldn't taking out the needles that are more common in haystack #2 bias that test?) The authors then use the total number of genes (15,603), not just the ones that were "considered", for their Bonferroni correction – are there

Bonferroni significant genes that have a higher rate of variants in controls versus cases? Overall, I found this approach confusing and would appreciate a clearer explanation and rationale.

E-F. Conclusions: robustness, validity, reliability. Suggestions for improvement. Beyond the considerations I've already described (ADHD comorbidity, statistical strategy for discovery analyses), the biggest challenge to the robustness of this study's results is its minimal consideration of sex differences. In population studies, males are diagnosed with ADHD 2:1 compared to females; in clinical samples, that sex ratio can be 4:1. Basic information on sex is lacking (e.g., what percentage of ADHD cases versus controls are male and female?) Even if there is not sufficient power to conduct sex-stratified analyses, or if the authors expect minimal sex differences in the etiology of ADHD, that would be important to note and explain.

G. References. The references are appropriate. Overlap with their 2019 paper in terms of sample and findings (including MAP1A results) could be described more clearly.

H. Clarity and context: lucidity of abstract/summary, appropriateness of abstract, introduction and conclusions. I found the paper to be clearly written, overall. It has, at times, a bit of a "laundry list" feel, where results are itemized with little guidance about what the reader is supposed to take away as the most important point. For instance, in the "Biological Implications" section, the authors end the section by reporting that the top genes were not more expressed pre-natally compared to post-natally, and had elevated expression within 11 of 12 brain developmental stages. Is this surprising, or different than what is found when examining common variation? How does this fit with models of ADHD as a "neurodevelopmental" disorder? This is just one example where the *implications* of the results could be expressed with greater lucidity.

Referee #4

(Remarks to the Author)

A. Summary of the key results

B. Originality and significance: if not novel, please include reference
>> new analysis of previous data, so some novelty in insights gained

C. Data & methodology: validity of approach, quality of data, quality of presentation
(I will only comment on technological/methodological aspects, as the biological field is not my own)

>> methods sections, in particular those pertaining to reanalysis of snRNAseq data, are completely missing. There is no information about:

1. which datasets were used - a basic reference and comment that data were obtained from the GEO is inappropriate - for e.g., the Sarkar dataset contains 7,584 samples, while the La Manno dataset contains 6,179 - which of these 13,763 samples were chosen for reanalysis? (this is not including all other datasets reanalysed in this manuscript). There is also additional concerns/missing information about how data was normalized for different sequencing chemistries, sequencing platforms, sample collection methods, and reference genome use. There was no description of QC metrics, or where they shown, exclusion criterion, etc.

2. complete lack of orthogonal validation that key ADHD DEGs are in fact real and not artifacts of sampling error or sequencing

>> there are also profound errors in data presentation - in particular a Manhattan plot with no individual SNP data plotted, no statistical cutoff, or other standard representation points.

>> missing data in figures also concerning - e.g. what are heatmap/scales representing in UMAP plots of snRNAseq data - z-score, expression values, something else?

>> no details of what 'significant' or 'background' genes are in Figure 4, panel A

>> The proteomics sample prep is mostly fine. A general lack of clear methods permeates this section of the manuscript as well. And a lack of clear descriptions of how previous datasets are normalized for integration here. I am also unclear about the power of these datasets as, like the sequencing integration, no clear statement is made about which samples from the numerous published datasets are used.

>> There is no QC metrics about sample inclusion, not statement about sex, age, genetic predisposition, etc., in addition to sample sizes, which make interpretation of the comparison data difficult.

>> On the whole, the level of ambiguity in what data was actually analyzed in this manuscript make review very difficult - the reader has no idea what is included in the analysis.

>> minor point: Fig 2 scale bar has an error in the title. I think the square (before $\log_{10}(P)$) should be 'delta'? hard to say.

D. Appropriate use of statistics and treatment of uncertainties

>> inappropriate due to lack of methods descriptions. Unable to review

E. Conclusions: robustness, validity, reliability

>> unable to review or comment on due to numerous errors in generation and presentation of data

F. Suggested improvements: experiments, data for possible revision

>> the authors need to include appropriate methods in their manuscript - simply stating the same metrics and approaches were used as in previous studies (e.g. line 577) is inappropriate - particularly given this new manuscript is asking a completely different question with a large patient cohort and as such likely has different exclusion criteria. This omission alone should exclude publication

>> additional lacking experimental details for reanalysis of scRNAseq data (which was actually snRNAseq data) is another major concern. These datasets were generated on different single cell platforms, are aligned to different reference genomes, are sequenced on different Illumina platforms and had different baseline QC metrics applied (which is not surprising given they were generated 2 years apart in a rapidly evolving field). For the analysis presented in this paper to be scientifically sound, raw fastq files should be obtained for all samples, they should be aligned to the same reference genome and appropriate and standardized QC metrics/exclusion criterion applied (and presented in the manuscript). They should be normalized, integrated, and then clustered to ensure appropriate reporting of results. The authors are strongly urged to also orthogonally validate results as output from such analysis.

>> Figure 3, panel a is not helpful without mapping/detailing connecting genes, completing GSEA or other appropriate analyses and attempting to build out a mechanistic framework.

G. References: appropriate credit to previous work?

>> appropriate, but limited

H. Clarity and context: lucidity of abstract/summary, appropriateness of abstract, introduction and conclusions

>> the manuscript should be carefully read for numerous grammatical errors and confusion sentence structure

>>unable to review conclusion appropriateness due to numerous major concerns with methods for generation of data

Version 1:

Reviewer comments:

Referee #1

(Remarks to the Author)

The authors have been responsive to the majority of my comments. I have just one remaining comment. My first key point concerned whether the selection of samples in this study will be representative of ADHD in the general population (e.g., high rates of ASD and ID). The authors responded showing extensive comparisons between this specific study with the general population of iPsych participants. Still, I would be interested in some data and perspective on how the iPsych sample compares to other cohorts of ADHD patients, beyond the iPsych general population sample. This concerns comorbidity with ASD and ID, but also the question whether iPsych potentially includes more severe ADHD cases compared to other population cohorts.

Referee #2

(Remarks to the Author)

In their revised article, the authors have addressed most of my major comments. Namely, they have performed burden tests on the X chromosome and replication analyses that overall make their story more complete. However, I remain concerned about the functional data:

- The authors chose not to perform immunofluorescence on their iPSC-derived neurons. While they emphasize that their differentiation protocol is standardized and well-established—which is accurate—this does not eliminate the need to validate the identity and purity of the resulting cell types. It is standard practice in the field to provide basic characterization, irrespective of the protocol used. In this manuscript, the authors do not even include brightfield images, leaving the reader without any confirmation that the cells under study are indeed NPCs or neurons. This omission weakens confidence in the functional assays and is particularly concerning given that their conclusions appear to be drawn from a single differentiation in a single genetic background. I disagree that such basic validation via immunofluorescence is out of scope, especially for a journal of Nature's caliber.

Referee #3

(Remarks to the Author)

The authors have responded to my previous comments. I especially appreciate the extra work they did to consider comorbidities with disruptive behavior disorders. I do not have further revisions to suggest.

Referee #4

(Remarks to the Author)

The authors have made extensive revisions and added a great deal of text both in the response to reviewers as well as in

the text, that does a great job of clearing up much of the confusion present in the initial review of the manuscript. My initial review was unnecessarily aggressive and I apologise for this, the GWAS components of the study remain high quality, and the use of the enlarged dataset will no doubt be widespread." However, my main concerns remain, largely focused on the way the snRNAseq data have been obtained/analyzed/and interpreted - which continue to be the weakest part of the manuscript. For simplicity, I respond to the authors comments to each initial concern separately:

Comment #2:

-the authors provide a link to the GEO repository for the La Manno dataset (GSE76381) from the Linnarsson lab. This repository contains both mouse and human samples (as described by the authors: Human embryo ventral midbrain cells between 6 and 11 weeks of gestation, mouse ventral midbrain cells at six developmental stages between E11.5 to E18.5, Th+ neurons at P19-P27, and putative dopaminergic neurons at P28-P56), and there are 6179 deposited datasets for download from this repository. Have the authors use all samples in their reanalysis (a quick review of the original manuscript described 10 human embryos were used) - if not, which of the subsamples uploaded to the GEO repository have been used?
-the additional clarifying comments on pages 33 and 35 are appreciated.

Comment #3

-this was an error on my part, the authors are correct that they did not identify DEGs in their study.
-As an expansive GWAS the current study fits the brief.
-the title alluding to an implication of neuron biology would be strengthened by some orthogonal experimentation. Perhaps this could be edited given this is not a result, but more an interpretation and hypothesis given the sequencing data/analysis.

Comment #5

-edits to Figure 4 to improve clarity are appreciated and address the original concern.

Comment #6

-additions to statistical testing throughout, in particular Fig 4 multiple testing is appreciated
-improvements and inclusion in methods, in particular for proteomics section is appreciated

Comment #8

-Supplementary Figure 1 a nice addition.
-Some concern still about a lack of details on sex - important also for the reanalyzed data (e.g. La Manno) which contains human samples of mixed sex - I could not find any details about which samples were used for the reanalysis. Does this matter? Given the authors comment throughout the manuscript about sex-specific responses suggests it does (?), but this is unclear.

Comment #9

-Supplementary Figure 2 (original Supplementary Figure 1) does a nice job of detailing how sample/datasets are integrated, but still lacks the standardization and preanalysis steps (e.g. ambient RNA cleanup in snRNAseq data as described above).

Comment #11

-I was unclear in the original comments, and I apologize for the ambiguity of the comment. The authors are correct that they have extensive methods descriptions for a lot of the manuscript, however the weakness remains about how the published datasets were collected, reanalyzed using modern methods (still missing), and integrated. This may seem like a small concern to the authors, but the onus is on the authors to update these historical datasets to ensure artifacts are not incorporated due to usage of errors (non egregious or planned) creeping in due to older computational pipelines.

Comment #12

-reviewing other reviewer comments and author responses, it is true that the GWAS analysis and much of the manuscript is well-prepared. There does remain concerns in the snRNAseq integration and analysis, and given the assertion in the title of a neuronal dysfunction, the lack of orthogonal validation, and the weaknesses in the cell-type specific snRNAseq data remains, making these cell type specific conclusions weak.

Comment #15

-I appreciate the authors statement in response that the current study provides an important dataset for hypothesis generation and follow-up functional studies. It achieves this in volumes, and the authors are to be applauded for this. My concern about the attribution of function/cell type specific biology remains, and given it is listen prominently in the title of the paper, these weaknesses are a major concern.

Version 2:

Reviewer comments:

Referee #1

(Remarks to the Author)

Thank you for the additional comments around ADHD diagnosis in iPsych. I have no further comments or suggestions.

Referee #2

(Remarks to the Author)

The reviewers have sufficiently answered my concerns.

Referee #4

(Remarks to the Author)

Referees' comments:

We thank the editor and the reviewers for their valuable comments and suggestions, which we believe have greatly improved the manuscript. We have added and modified several analyses, with major additions including:

- Analysis of the X chromosome, including overall burden analyses (with and without ID), gene-burden analysis, and sex-specific analyses (with and without ID).
- Sex-stratified X chromosome analyses.
- Burden heritability analysis of ADHD without co-occurring ID.
- Analyses of rare variant load across an extended set of comorbidities, now including also comorbid substance use disorders and disruptive behavior disorders (CD and ODD).
- Categorizing of non-European ancestry groups in the exome sequenced subcohort of iPSYCH and listing of the limited number of individuals for each of these groups (to facilitate contribution to cross- and multi-ancestry studies).

We believe these analyses together with new figure and text revisions have led to a clearer and more impactful version of the paper.

Referee #1 (Remarks to the Author):

Review of “Rare coding genetic variants confer high risk of ADHD, implicate neuronal biology, and impact socioeconomic outcomes”

The authors present the largest study to date on the contributing of rare coding variants on ADHD risk. Data are primarily from iPsych (8,895 cases and 53,780 controls, although additional analyses were performed in smaller cohorts). The findings are of interest, e.g., the authors show that rare and common variants make independent contributions to ADHD risk. They argue that although rare variants explain less of the genetic variance compared to common variants, effect sizes of rare variants are large (nicely illustrated in Figure 2, facilitating the identification of biological mechanisms and causal genes. The analyses seem sound and the results present effect sizes and standard errors (the latter are quite large for some results, implying statistical power is sufficient but could be better for some of the analyses). The manuscript is generally well written (some suggestions for improvement are below).

We thank the reviewer for the positive feedback on our analyses, results and interpretation of the results.

I have one major concern/question for the authors, and this relates to the representativeness of the iPsych sample for other populations of ADHD cases (see point 1 below).

These are my main comments/suggestions:

Comment #1

As far as I'm aware, The iPsych sample is mostly based on psychiatric/medical records. I am wondering whether this selection, is biased towards the more severe cases of ADHD? I noticed that the prevalence of Intellectual Disability is close to 20% and the prevalence of ASD is even higher. These prevalences seem high to me. Can the authors comment on how this sample compares to

“population” cohorts of ADHD cases and on whether they expect the contribution of rare coding variants to be stronger in iPsych cases because of the specific selection that was used?

Response

Regarding comorbid ASD:

In this study 37% of the individuals with ADHD have comorbid ASD, which is higher than in the full population-based iPSYCH cohort where 22% of individuals with ADHD have comorbid ASD, as pointed out by the reviewer. This could potentially increase power to detect rare variant genes shared with autism compared to analyzing a sample reflecting ADHD-ASD comorbidity in the general population. In the c-alpha test (described on page 18) we specifically evaluated if the distributions of constrained genes with rare variants in ADHD and ASD have the same underlying distribution, reinforcing the c-alpha results reported in our ASD-ADHD cross-disorder paper on a smaller cohort¹. No difference in the gene distributions was identified suggesting a substantial overlap in rare variant risk genes between ADHD and ASD. This is also consistent with results of the rare variant burden analyses of comorbid ASD and other comorbid conditions (excluding ID) on page 15 and 16, showing no differences between comorbid and non-comorbid individuals in the load of class I and class II variants in constrained genes (Supplementary Figure 13.b-d,g,h, Supplementary Table 19).

We modified the text on page 19 to address this issue:

“We note that the prevalence of co-occurring ASD was higher in this study (37%) compared to the full population-based iPSYCH case-cohort sample (22%), which potentially could increase power to identify risk genes shared across ADHD and ASD. However, it has probably not influenced the gene discovery much since (i) the c-alpha results suggested substantial sharing of rare variant risk genes across ADHD and autism regardless of including comorbid ADHD-ASD cases, and (ii) the rare variant burden analysis across comorbidities showed no difference between ADHD with ASD and ADHD without comorbid conditions in the load of class I and class II variants in constrained genes (Supplementary Figure 13, Supplementary Table 19).”

Furthermore, we have now included a new supplementary figure (Supplementary Figure 6) that outlines the comorbidities of individuals with ADHD carrying a class I variant in the three exome-wide significant genes (see figure below). There is a high comorbidity with ASD among carriers of class I variants in *ANK2* (64%), but for the other two genes the prevalence of comorbid ASD is at the same level as the prevalence in the sequenced cohort (36% for *MAP1A* and 40% for *ANO8*) and thus, for the latter two genes ASD does not contribute to the signal beyond what could be expected based on its prevalence in the analyzed sample. We have discussed the comorbidities of individuals with class I variants in the three significant genes in the Discussion, page 19:

“For *MAP1A* around 50% and for *ANO8* around 30% of the association signal was driven by rare deleterious variants in individuals diagnosed with ADHD only (without comorbid schizophrenia, ID or autism), while the *ANK2* signal was largely driven by ADHD with co-occurring autism or ID, as around 90% of ADHD individuals with *ANK2* rare deleterious variants had these comorbidities. This is consistent with *ANK2* being a known rare variant risk gene in autism². *MAP1A* and *ANO8* have not been associated with ASD or other neurodevelopmental conditions in other studies^{2,3}, although *MAP1A* was significantly associated when combining rare deleterious variants in both ADHD and ASD, but not significant in the disorders separately in our previous study¹.”

New Supplementary Figure 6:

“Supplementary Figure 6. Comorbidities of individuals with class I variants in *MAP1A*, *ANO8* and *ANK2*

Phenotypic breakdown of individuals carrying at least one rare class I variant. The categories ADHD_ASD_SZ_ID indicate the presence (1) or absence (0) of diagnoses for ADHD, ASD, schizophrenia (SZ), and intellectual disability (ID), respectively. For example, 0_0_0_0 represents controls without any of the listed diagnoses, 1_0_0_0 indicates individuals diagnosed with ADHD only, 1_0_0_1 represents individuals diagnosed with both ADHD and ID, and 1_1_1_1 corresponds to individuals diagnosed with ADHD, ASD, SZ, and ID.”

Regarding comorbid ID:

We agree with the reviewer’s observation that the prevalence of comorbid intellectual disability (ID) in individuals with ADHD is high (18,4%) in this subset of the iPSYCH cohort, which exceeds what is observed in the full population-based iPSYCH cohort (9%). We have found that the load of rare variant risk is significantly higher in ADHD with ID compared to ADHD without ID (Figure 1b, Supplementary Figure 13.a, Supplementary Table 19) and the increased prevalence of comorbid ID should therefore increase the power for gene discovery (all things being equal).

More generally, comorbid ID might have an impact on several results (regardless of the specific comorbidity rate in the study sample), which is why we consistently have performed all variant burden and gene-set analyses both with– and without cooccurring ID. We have included a sentence on page 5 to make this clearer:

“There was a high comorbidity with ID among included individuals with ADHD (18.4%; Supplementary Table 1) and the impact of co-occurring ID was evaluated by doing all variant burden and gene set analyses both with and without ADHD individuals also diagnosed with ID.”

As written above the impact of co-diagnosed ID was consistently evaluated in all variant-load and gene set analyses, but we did not evaluate the impact of co-diagnosed ID on the burden heritability, which we have included now, and added the following to the results section, page 9:

“When excluding individuals with ID, the burden heritability for class I and class II variants decreased to 1.43% (s.e. = 0.74%) and 0.26% (s.e. = 0.27) respectively.”

Furthermore, it can be seen from the new Supplementary Figure 6 (shown above), that the prevalence of comorbid ID is slightly higher (compared to the overall study sample) among individuals that carry a class I variant in *ANK2* (27.3%) but slightly underrepresented among individuals that carry a class I variant in *MAP1A* (16.7%) and *ANO8* (10%). Thus, the exome-wide significant gene associations are not primarily driven by comorbid ID.

Finally, we note that we identify an increased burden of rare deleterious variants in ADHD without comorbid ID, compared to controls (demonstrated in Figure 1.b), and about one out of five individuals with ADHD without ID had a class I variant in a constrained gene, while it was one in three individuals with ADHD and comorbid ID (described on page 4). These findings are consistent with what is also observed in rare variant studies of ASD and substantiate our conclusion that ADHD, regardless of comorbidity with ID, is associated with a higher load of rare deleterious variants.

Comment #2

I find the description of class I and class II variants a little confusing in the main text. Can the distinction be shown in a Figure? This could easily be added to Figure 1, maybe as a separate panel or at least described in the figure legends.

Response

We thank the reviewer for the excellent suggestion. We have updated Figure 1A to clearly illustrate that class I variants consist of rPTV + rSevereDMV, while class II variants are exclusively rModerateDMV. Additionally, we have expanded the figure legend to include a detailed explanation of the grouping (highlighted in red below).

“Figure 1. ADHD risk across rare variant categories and average load across analyzed groups

1.a. Odds ratio of rare protein truncating variants (rPTV), rare severe damaging missense variants (rSevereDMV), rare moderate damaging missense variants (rModerateDMV) and rare synonymous variants (rSYN) in all genes (marked in yellow) and in constrained genes (pLI >= 0.9; marked in red) in individuals with ADHD (N = 8,895) and controls (N = 9,001). The dots represent the odds ratio and the vertical lines 95% confidence intervals (CI). **Due to similar effect sizes of rPTVs and rSevereDMVs in constrained genes (pLI >= 0.9) these variants were grouped into Class I variants while rModerateDMV were categorized as Class II variants** Note: number of rSYN in each individual, is used as a covariate in the analyses, it is therefore not possible to test for differences in rSYN load across all autosomal genes.”

Comment#3

Line 92, I don't believe schizophrenia and bipolar disorder would be considered as neurodevelopmental disorders.

Response

We have deleted "neurodevelopmental component" on page 3, according to the comment from the reviewer.

Comment#4

Is there a reason why the X chromosome hasn't been included? (line 120)

Response

We agree with the reviewer that including the X chromosome is highly relevant, and we have now included X chromosome analyses. We have conducted a comprehensive analysis of the X chromosome, accounting for differences between its pseudoautosomal regions (PAR) and non-pseudoautosomal regions (nonPAR). The corresponding results and methods have been added to the revised manuscript.

We have added the following to the Results section pages 8-9:

"X chromosome analyses

Due to the distinct properties of the X chromosome, with its pseudoautosomal (PAR) and non-pseudoautosomal (nonPAR) regions, we applied region-specific definitions of rare variants. In PAR regions, rare variants were defined as those with an allele count ≤ 5 , following what we did for the autosomal chromosomes. For the nonPAR regions, we used a stricter threshold of allele count ≤ 3 to account for the hemizyosity in males for these regions (see methods).

Significant differences in the load of deleterious variants on the X chromosome were observed only for class II variants in ADHD with ID (Supplementary Figure 8, Supplementary Table 8), and in the sex-stratified analysis comparing males with ADHD to controls (Supplementary Figure 9), but not after excluding comorbid ID. No genes on the X chromosome were associated with ADHD in the gene-based burden test after Bonferroni correction (P-value threshold = 1.34×10^{-4} , corresponding to 0.05 corrected for 372 tests; see Methods; Supplementary Table 9)."

We have added two new Supplementary Figures with X chromosome results:

“Supplementary Figure 8 ADHD risk across variant categories on the X chromosome

Burden (odds ratio [OR]) of class I, class II and rare synonymous variants (SYN) in chromosome X genes stratified by their pLI score. Results are from logistic regression (results in Supplementary Table 8). Intellectual disability (ID). *Indicates nominal significant association $P < 0.05$, **Indicates significant association after Bonferroni correction correcting for three gene sets (P-values less than $P = 0.0167$ are considered significant).”

“Supplementary Figure 9 Sex-specific X chromosome rare variant analyses

Sex-specific rare variant analyses for chromosome X. The load of rare class I, class II and rare synonymous variants (SYN) across pLI bins of genes on chromosome X in multiple comparisons. Results are from logistic regression. *Indicates nominal significant association $P < 0.05$, **Indicates significant association after correction for multiple testing (a P-value = 0.0167 is considered significant, correcting for three gene sets).”

We have added the following to the Methods section pages 30-31:

“X chromosome analyses

Given the distinct regions of the X chromosome, we applied region-specific thresholds to define rare variants: In the pseudoautosomal regions (PAR), rare variants were defined as those with an allele count ≤ 5 following what was done for the autosomes. In non-pseudoautosomal regions (nonPAR), the rare variant threshold was adjusted to an allele count ≤ 3 to account for the hemizyosity in males. This adjustment was derived using a scaling factor based on the relative contributions of males and females in iPSYCH (6,002 females and 11,894 males) and the non-psych non-Finnish European subset of gnomAD (19,916 females and 24,863 males):

$$5 \times \frac{(\text{num}_{\text{males}} + 2 \times \text{num}_{\text{females}})}{2 \times (\text{num}_{\text{males}} + \text{num}_{\text{females}})} = 3.53$$

Variants were categorized into **class I**, **class II**, and **synonymous** variants, using the same criteria applied for autosomal variants. In the burden test of males, the number of alleles for nonPAR variants was counted as two to account for hemizyosity.

Logistic regression was performed on three gene sets stratified by their pLI score, comparing the following groups of individuals: ADHD versus controls, ADHD without ID versus controls, ADHD with ID versus controls, ADHD with ID versus ADHD without ID. Covariates included birth year, sex, the first ten principal components (PCs) from PCA (non-European samples excluded), total variant count, rare synonymous variant count, exome target coverage (percentage $\geq 20\times$), mean read depth at target sites, and sequencing batch (one-hot encoded). In addition, sex-stratified analyses were performed using the same approach.

For gene discovery on the X chromosome, we applied the same approach as for autosomes when combining iPSYCH data with gnomAD. We excluded genes with higher number of synonymous variants in individuals with ADHD compared to controls and retained genes with a higher rate of class I variants in cases compared to controls (78 genes included) or higher number of class II variants (44 genes). We performed gene-based burden analysis separately for males and females, where we, for each gene, compared the number of individuals with ADHD carrying at least one class I or class II variant to the number found for controls, using a two-tailed Fisher’s exact test. Notably, six genes had both increased class I and II variants in cases compared to controls, for these genes, a meta-analysis was used to combine the effect of class I and class II variants using the weighted Z-score method. The weight was determined as the ratio of the standardized effect sizes observed for class I and class II variants in the enrichment analysis of constrained genes on chromosome X (weight = $1.03/1.86 = 0.55$). In total, we performed 78 x 3 tests for class I variants (female, male, and combined), 44 x 3 tests for class II variants, and six tests for combining class I and II, resulting in a total of 372 tests. The Bonferroni correction threshold for statistical significance was set at $P < 1.34 \times 10^{-4}$.”

Comment#5

Line 328: how robust is this association? Since overall enrichment is not significant, can these 4 nominally significant associations be due to type-II error?

Response

We think the reviewer means type-I error (i.e., the finding of a significant difference when there is no true difference). We agree that the results are too weak for hypothesizing on potential common and rare variant convergence. We have therefore modified the sentence on page 14, which now reads:

“The gene set was not enriched in common variant association signals overall (Supplementary Table 16), but we note that four out of the 20 genes demonstrated nominally significant gene-based association with ADHD, including *MAP1A* ($P = 0.005$, Supplementary Table 16)”.

The last part of the sentence “suggesting a tendency towards convergence of common and rare variant risk” has been deleted.

Comment#6

Line 334: I assume iPsych was excluded from the discovery GWAS? What was the discovery GWAS? Briefly explain here please

Response

The iPSYCH cohort represents the largest contribution to the most recent GWAS meta-analysis of ADHD, and as such, the ADHD polygenic score (PGS) was partially derived internally to maximize statistical power. However, it is important to note that the target sample was never included in the training data. We have provided a detailed explanation of this approach in the Methods section, page 41:

“The PGS was constructed by splitting the relatedness pruned GWAS dataset (25,895 with ADHD; 37,148 controls) in 50 random subsets of roughly even size. For each of these a GWAS was run on the complimentary 49 subsets using 10 principal components, and the results were meta-analyzed with PGC and deCODE ADHD GWAS summary statistics described elsewhere⁴. The resulting summary statistics were then used to generate PGSs in the index set using the SBayesR algorithm implemented in LDAK⁵. Finally, the scores from the 50 subsets, which together cover the full data set was assembled into one data set.”

While we are happy to provide further details regarding the generation of the PGS in the main text, we believe that such a level of detail may not align with the preferred style of the journal in this section, but if the reviewer/editor prefer we could elaborate also in the main text.

Comment#7

I believe Figure 1 can be improved: Check colors of Panel A, these seem incorrect. Figure should also define class II variants.

Response

We have reviewed the colours in Panel A and confirm their correctness: orange represents autosomal genes with $pLI \geq 0.9$, while purple represents all autosomal genes. To enhance clarity, we have added two additional bars in Panel A and included a description in the figure legend to clearly define Class I and Class II variants (text marked in red below):

“Figure 1. ADHD risk across rare variant categories and average load across analyzed groups

a. Odds ratio of rare protein truncating variants (rPTV), rare severe damaging missense variants (rSevereDMV), rare moderate damaging missense variants (rModerateDMV) and rare synonymous variants (rSYN) in all genes (marked in yellow) and in constrained genes ($pLI \geq 0.9$; marked in red) in individuals with ADHD ($N = 8,895$) and controls ($N = 9,001$). The dots represent the odds ratio and the vertical lines 95% confidence intervals (CI). **Due to similar effect sizes of rPTVs and rSevereDMVs in constrained genes ($pLI \geq 0.9$) these variants were grouped into Class I variants while rModerateDMV were categorized as Class II variants.** Note: Count of rSYN variants in each individual, is used as a covariate in the analyses, and thus it is not possible to test for differences in rSYN load across all autosomal genes.”

Comment#8

- please define the abbreviation rPTVs when it is used for the first time

Response

It was already defined the first time on page 3.

Comment#9

- line 148: should this read rPTVs in constrained genes?

Response

We assume the reviewer is referring to the sentence:

“Consequently, rPTVs and rSevereDMVs were grouped together (referred to as class I variants; Supplementary Table 2) in the gene discovery analysis.”

rPTVs and rSevere DMVs were analysed together as class I variants for all genes, not only for constrained genes. Thus, we believe the sentence is correct as it is.

Comment#10

- Line 184: should this read class I or class II variants (instead of and)

Response

When analysing class I variants, genes with a higher load of class I variants in cases compared to controls were included, no matter the load of class II variants. Likewise, genes with a higher load of class II variants in cases compared to controls were included in tests of class II variants no matter the load of class I variants. To make it clear how the filtering was done we have modified the text on page 7, which now reads:

“Therefore, the analysis was restricted to include only genes with a higher rate of class I variants (3,698 out of 15,603 genes; 24% of the genes) or class II variants (1,026 genes out of 15,603; 7% of the genes) in cases compared to controls. A small number of genes had both increased class I and class II variants (347 genes). For these genes a higher number of both types of variants in cases compared to controls was required.”

Comment#11

- Line 193: this imply should read this implied

Response

Thank you for noting this, we have corrected the sentence as suggested.

Comment#12

- Line 216: 5.2% of the class I or class II heritability?

Response

It is 5.2% of the class I heritability. We have added this information on page 13, which now reads:

“...(MAP1A, ANO8, ANK2) explained 5.2% (s.e. = 3.4%) of the class I burden heritability, suggesting...”

Comment#13

- Line 302: What does the OR represent? What is the reference group?

Response

To enhance clarity, we have modified the paragraph on page 13, which now reads:

“We found that rPTVs had a negative impact on education level in individuals with ADHD. rPTVs in constrained genes were significantly associated with only finishing primary school (OR=1.24; standard deviation [s.d.] = 5.23×10^{-2} ; $P = 3.68 \times 10^{-5}$) when comparing education level in those with ADHD having one (or more) rPTVs to those with ADHD without rPTVs in constrained genes (N = 6,488 only primary school; N = 1,436 higher education; Figure 5.a, Supplementary Table 14).”

Referee #2 (Remarks to the Author):

In this manuscript, Demontis, Duan, and colleagues perform a rare variant based approach to implicate three new genes in ADHD. As far as I can tell, this is the largest rare variant study in ADHD performed to date. First, the authors recapitulate well-established findings that ADHD cases are enriched for rare protein-coding variants in intolerant/constrained genes. They then perform gene-based burden tests. The novelty in this study mostly lies in the discovery of three novel risk genes: ANO8, ANK2, and MAP1A.

Unfortunately, the authors chose to exclude the X chromosome from their analyses, despite the well-established importance of this chromosome in neurodevelopmental disease.

The authors also present some experimental data (IP-MS in iPSC-derived NPCs and excitatory neurons) to further interrogate these novel loci, though these experiments would benefit from additional validation, as detailed in my comments below.

Major comments:

Comment#1

A recent paper in Nat Comms (PMID: 38997333) performed a large scale de novo mutation analysis in ADHD, identifying *KDM5B* as a risk gene. The authors should cite this paper in their revision. Also, can the authors comment on whether the *KDM5B* association replicates in their dataset, and vice versa, whether *ANO8*, *ANK2*, and *MAP1A* show any enrichment in the de novo study? This would provide nice orthogonal evidence.

Response

In the study by Olfson et al. *KDM5B* was identified by merging their exome-sequencing data on 147 ADHD parent-child trios and 780 unaffected parent-child trios with our previously published iPSYCH data on 3,206 ADHD cases and 5,002 controls (Satterstrom et al. Nature Neuroscience 2019). The previously published iPSYCH data are also included in the present study. Their study is therefore not independent from ours, and any consistent findings cannot be considered as replications.

To address the reviewer's comment, we analysed the top 20 genes from our study (P-values ≤ 0.001) alongside the top four genes from the de novo analysis of the Olfson et al paper (FDR < 0.3). The respective P-values from our study (x-axis) were plotted against the FDR/q-values reported in the de novo study (y-axis), as shown below. Genes marked in blue represent the top 20 genes from our study, while those in red represent the top genes from the de novo study. Note that *GNB2L1*, one of the top four genes in the Olfson study, is not shown as it was excluded from our gene discovery analysis.

[FIGURE REDACTED]

In the paper by Olfson et al. *KDM5B* was identified as risk gene for ADHD with an FDR = 0.04. Thus, it did not surpass the more stringent exome-wide significance threshold applied in our study and the lack of significant association observed in our larger data set is therefore not unexpected. *KDM5B* was not among our top 20 genes, but it showed a relatively low gene-based P-value (P = 1.23×10^{-3}), indicating a potential involvement in ADHD.

We have included a small discussion of the findings from the Olfson et al. study in the discussion, page 19:

“A previous study has identified *KDM5B* as a potential risk gene for ADHD (FDR = 0.04)⁶ by combing data on de-novo mutations in 147 ADHD parent–child trios and 780 unaffected parent–child trios with the previously published ADHD iPSYCH data¹. Although *KDM5B* was not significantly associated with ADHD in this larger data set and not among our top 20 genes, it did show a moderate association signal with OR of 2.93 (P = 1.23x10⁻³)”

Comment#2

The authors show that there is an enrichment of rare PTVs and missense variants in constrained genes in fig 1a. They show that there is no enrichment of rare synonymous variants (rSYN) in constrained genes, which is good. However, it is strange that they didn't also show the distribution of rSYN variants exome-wide as they did for other the other variant consequences. It seems to me that this should be included as another negative control.

Response

We are using the count of rSYN variants per individual as a covariate in the analyses, to correct for potential differences in the general variant count across individuals. So, it is not possible to include the analysis the reviewer is requesting. We have added text to the Figure 1.a legend, to make it clear to the reader why this analysis is not included:

“Note: Count of rSYN variants in all genes, in each individual, is used as a covariate in the analyses, and thus it is not possible to test for differences in rSYN load across all autosomal genes”.

Comment#3

There still appears to be inflation even after excluding the ~3000 genes with increased syn variation in cases vs. controls (Supp. Figure 12A). What was the lambda?

Response

The lambda was calculated to 1.53 and we have included the value in the legend of Supplementary Figure 16.a.

The inflation we observe is consistent with ADHD being highly polygenic also for rare variants and is comparable to what has been found in the rare variant study of schizophrenia by Singh et al. (Nature 2022). The QQ-plot in Singh et al. (Figure 2.b in the paper is shown below) demonstrates a large inflation based on a sample size around three times as large as ours (24,248 schizophrenia cases and 97,322 controls and 3,402 trios).

[FIGURE REDACTED]

Figure 2.b from Singh et al. Nature 2022. The observed $-\log_{10}$ P values plotted against the expected P values given a uniform distribution. The per-gene P values were calculated by meta-analysing the two-sided burden test P values from rare coding variants in 24,248 cases and 97,322 controls and 3,402 trios.

Comment#4

The description of the cohort demographic could use more details. In particular, what was the ancestral breakdown of the fully sequenced cohort, prior to sample selection? How many individuals were from non-European ancestries? Excluding non-European individuals is not ideal. Furthermore, what was the sex breakdown of controls?

Response

The sex break-down of controls was already in Supplementary Table 1 for iPSYCH and the clinical samples. We have now included this information also for gnomAD (19,916 females and 24,863 males).

Regarding non-European individuals, we have identified ancestry information for all controls and individuals diagnosed with ADHD in the iPSYCH dataset prior to applying quality control filters at the sample level. Ancestry assignments were conducted using UK Biobank as a global reference for worldwide population⁷. The identified subpopulations are illustrated in the figure below (new Supplementary Figure 3) and listed in a new Supplementary Table 23 (inserted below). Sample sizes of other genetic ancestries were low, particularly for controls (see Table below) and we have therefore not sufficient statistical power to perform robust and properly controlled analyses for these subpopulations. However, findings from a recent study on ASD⁸, which often co-occurs with ADHD, demonstrated that deleterious rare variant risk for autism were consistent across European and admixed Latin American populations. This indicates that the risk genes identified in our study (based on individuals with genetic European ancestry) may be valid and have a similar biological impact in non-European populations.

We strongly agree with the importance of investigating non-European ancestries and have therefore included the table below as a new Supplementary 23, so that other researchers can see which samples are available in iPSYCH for inclusion in future rare-variant studies in other ancestries.

“Supplementary Figure 3. PCA plot of iPSYCH samples

PCA plot of iPSYCH samples **a.** before quality control (11,051 with ADHD and 10,448 controls), individuals are marked in colour depending on the ancestry group they have been assigned to in the PCA (See Supplementary Table 1, for sample sizes of non-European ancestries.”

“Supplementary Table 23.

Genetic ancestry	Number of cases	Number of controls	Total sample size
Africa (East)	63	36	99
Africa (North)	74	43	117
Africa (South)	<5	<5	<5
Africa (West)	5	<5	7
Ashkenazi	9	<5	10
Asia (East)	23	<5	27
Bangladesh	<5	<5	<5
Finland	5	6	11
Japan	<5	<5	<5
Middle East	306	167	473
Pakistan	95	39	134
Philippines	<5	<5	5
South America	5	<5	9
Sri Lanka	24	7	31
non-Finnish Europe	10183	9952	20135
NA	253	181	434
Total	11051	10448	21499

Sample size of individuals in iPSYCH with ADHD and controls across other ancestries, which are not included in this study.”

Comment#5

I am a little confused about the gene burden test performed. Am I interpreting it correctly that they only performed the burden test on the genes that had higher rates of damaging variants in cases vs. controls? The authors ought to perform an exome-wide analysis and include all the summary stats in the supplement. 1) This would be helpful to potentially identify protective signals, and 2) would allow the community to perform meta-analyses in future work.

Response

Yes, correct, we are only reporting results for genes with a higher rate of damaging variants in cases compared to controls.

We need to be very strict in our QC when combining iPSYCH data with gnomAD data, to be sure that an observed higher load of rare deleterious variants in ADHD is not driven by a generally higher load of variants in cases compared to controls caused by technical differences, e.g. in sequencing platforms or QC. We therefore included the filtering step where we require the number of synonymous variants in a gene to be higher in controls compared to cases. This strategy, however,

also infers that we cannot report results for genes with a higher number of rare deleterious variants in controls compared to cases (i.e., potential protective signals) since such a result could be caused by the controls having a generally higher load of variants (due to the filtering strategy).

We have added the following text to the Methods section, page 28, to explain better why genes with a higher load of rare deleterious variants in controls compared to cases were not considered:

“We did not consider genes with higher number of class I and II variants in controls, since such observations could be caused by a generally higher number of variants in the gene in controls due to the previous filtering step where only genes with a higher number of rare synonymous variants in controls compared to cases were retained.”

We agree with the reviewer that identification of protective signals would be very interesting but that would require another filtering strategy than we have applied, and we think it is beyond the scope of this study to identify resilience genes.

We also agree with the reviewer that data sharing is important for the research community. Due to Danish data protection rules the full summary statistics cannot be shared publicly but the full summary for all genes can be accessed by contacting the iPSYCH data access committee, which is stated in the data availability statement.

Comment#6

It is a shame that the authors excluded the X chromosome, particularly since there are genes on this chromosome that clearly contribute to neurodevelopmental disease. I strongly urge the authors to include the X chromosome in the revised manuscript.

Response

We agree with the reviewer that including the X chromosome is highly relevant, and we have now included X chromosome analyses. We have conducted a comprehensive analysis of the X chromosome, accounting for differences between its pseudoautosomal regions (PAR) and non-pseudoautosomal regions (nonPAR). The corresponding results and methods have been added to the revised manuscript.

We have added the following to the Results section pages 8-9:

“X chromosome analyses

Due to the distinct properties of the X chromosome, with its pseudoautosomal (PAR) and non-pseudoautosomal (nonPAR) regions, we applied region-specific definitions of rare variants. In PAR regions, rare variants were defined as those with an allele count ≤ 5 , following what we did for the autosomal chromosomes. For the nonPAR regions, we used a stricter threshold of allele count ≤ 3 to account for the hemizyosity in males for these regions (see methods).

Significant differences in the load of deleterious variants on the X chromosome were observed only for class II variants in ADHD with ID (Supplementary Figure 8, Supplementary Table 8), and in the sex-stratified analysis comparing males with ADHD to controls (Supplementary Figure 9), but not after excluding comorbid ID. No genes on the X chromosome were associated with ADHD in the gene-based burden test after Bonferroni correction (P-value threshold = 1.34×10^{-4} , corresponding to 0.05 corrected for 372 tests; see Methods; Supplementary Table 9).”

We have added two new Supplementary Figures with X chromosome results:

“Supplementary Figure 8. ADHD risk across variant categories on the X chromosome

Burden (odds ratio [OR]) of class I, class II and rare synonymous variants (SYN) in chromosome X genes stratified by their pLI score. Results are from logistic regression (results in Supplementary Table 8). Intellectual disability (ID). *Indicates nominal significant association $P < 0.05$, **Indicates significant association after Bonferroni correction correcting for three gene sets (P -values less than $P = 0.0167$ are considered significant).”

“Supplementary Figure 9. Sex-specific X chromosome rare variant analyses

Sex-specific rare variant analyses for chromosome X. The load of rare class I, class II and rare synonymous variants (SYN) across pLI bins of genes on chromosome X in multiple comparisons. Results are from logistic regression. *Indicates nominal significant association $P < 0.05$, **Indicates significant association after correction for multiple testing (a P-value = 0.0167 is considered significant, correcting for three gene sets).”

We have added the following to the Methods section pages 30-31:

“X chromosome analyses

Given the distinct regions of the X chromosome, we applied region-specific thresholds to define rare variants: In the pseudoautosomal regions (PAR), rare variants were defined as those with an allele count ≤ 5 following what was done for the autosomes. In non-pseudoautosomal regions (nonPAR), the rare variant threshold was adjusted to an allele count ≤ 3 to account for the hemizyosity in males. This adjustment was derived using a scaling factor based on the relative contributions of males and females in iPSYCH (6,002 females and 11,894 males) and the non-psych non-Finnish European subset of gnomAD (19,916 females and 24,863 males):

$$5 \times \frac{(\text{num}_{\text{males}} + 2 \times \text{num}_{\text{females}})}{2 \times (\text{num}_{\text{males}} + \text{num}_{\text{females}})} = 3.53$$

Variants were categorized into class I, class II, and synonymous variants, using the same criteria applied for autosomal variants. In the burden test of males, the number of alleles for nonPAR variants was counted as two to account for hemizyosity.

Logistic regression was performed on three gene sets stratified by their pLI score, comparing the following groups of individuals: ADHD versus controls, ADHD without ID versus controls, ADHD with ID versus controls, ADHD with ID versus ADHD without ID. Covariates included birth year, sex, the first ten principal components (PCs) from PCA (non-European samples excluded), total variant count, rare synonymous variant count, exome target coverage (percentage $\geq 20\times$), mean read depth at target sites, and sequencing batch (one-hot encoded). In addition, sex-stratified analyses were performed using the same approach.

For gene discovery on the X chromosome, we applied the same approach as for autosomes when combining iPSYCH data with gnomAD. We excluded genes with higher number of synonymous variants in individuals with ADHD compared to controls and retained genes with a higher rate of class I variants in cases compared to controls (78 genes included) or higher number of class II variants (44 genes). We performed gene-based burden analysis separately for males and females, where we, for each gene, compared the number of individuals with ADHD carrying at least one class I or class II variant to the number found for controls, using a two-tailed Fisher’s exact test. Notably, six genes had both increased class I and II variants in cases compared to controls, for these genes, a meta-analysis was used to combine the effect of class I and class II variants using the weighted Z-score method. The weight was determined as the ratio of the standardized effect sizes observed for class I and class II variants in the enrichment analysis of constrained genes on chromosome X (weight = $1.03/1.86 = 0.55$). In total, we performed 78×3 tests for class I variants (female, male, and combined), 44×3 tests for class II variants, and six tests for combining class I and II, resulting in a total of 372 tests. The Bonferroni correction threshold for statistical significance was set at $P < 1.34 \times 10^{-4}$.”

Comment#7

As far as I can tell, I don’t think the authors included a recessive model, which also seems like a missed opportunity.

Response

A recent study on UKBB on exome-sequencing data from 176,935 individuals investigated the recessive impact on 311 phenotypes and identified six gene associations across all 311 diseases⁹. It is therefore highly unlikely that we will discover any significant genes based on our sample size, which is more than an order of magnitude smaller (it is not possible to include gnomAD data in this analysis as we don’t have individual level data available).

The framework for this study was not to identify variants with recessive effects. It would require phased genotypes to be able to reliably detect compound heterozygotes. That phased genotypes are a prerequisite to detect compound heterozygotes was demonstrated in the recent UKBB study showing that for 36% of individuals carrying ≥ 2 pLoF + damaging missense/protein-altering variants, the variants were located on a single haplotype (hence, not true compound heterozygotes).

It is not straight forward to phase genotypes in unrelated individuals, but it could be done e.g. by following the approach used in the UKBB study where the phasing of the unrelated individuals was benchmarked against phasing of 96 trios. It would require a substantial amount of work and would probably not add additional value to our study, due to the low power.

We have, however, performed an analysis based on non-phased genotypes to get an impression of the potential information we could gain. We conducted a compound heterozygosity (CH) analysis to evaluate the recessive model using the iPSYCH dataset. Specifically, we identified class I variants with a minor allele frequency (MAF) ≤ 0.01 and defined CH as individuals carrying two or more class I variants within the same gene. We applied two-tailed Fisher's exact test to compare the number of individuals with CH between cases and controls. As expected, no genes reached statistical significance. We have included the details on methods and results below. Since this analysis was done on unphased genotypes the results are unreliable (many of the associations do not reflect differences in true CH as shown in the UKBB study) and we would prefer not to include the results in the manuscript.

Methods and Results

Compound heterozygosity analysis on unphased genotypes in iPSYCH

To evaluate the recessive model, we performed a compound heterozygosity (CH) analysis in iPSYCH on unphased genotypes. We analysed class I variants with a minor allele frequency (MAF) ≤ 0.01 based on their frequency in iPSYCH. CH was defined as individuals carrying two or more class I variants in the same gene. We used two-tailed Fisher's exact tests to compare the number of individuals with CH between ADHD and controls.

In the analysis 478 genes had at least one individual being CH for class I variants, and differences in the number of CH individuals with ADHD and controls was tested for each gene using Fisher's method. No genes were significantly associated after Bonferroni correction (adjusted p-value cutoff is $0.05/478 = 0.0001$). *AGTR2* demonstrated the lowest P-value ($P = 0.00243$). Note that the stronger signal observed for the X chromosome is caused by the hemizyosity in males in the non-PAR regions. To conclude, we did not observe a significant recessive signal in ADHD cases compared to controls.

Figure. The Manhattan plot demonstrating the P values for a recessive model analysing CH for class I variants in individuals in iPSYCH.

Comment#8

The replication analysis is unconvincing. It is unclear whether the authors attempted to replicate the *MAP1A*, *ANO8*, and *ANK2* associations. Even if not nominally significant, do these genes show effect sizes in a consistent direction? It is possible that they are limited by statistical power, but if this is the case, the authors should at least include a power analysis. Alternatively, the authors could perform a meta-analysis across the two cohorts.

Response

In our previous rare-variant study of ADHD (cited in the introduction), we did not identify any exome-wide significant genes for ADHD when analysing 4,378 individuals with ADHD and 5,002 controls¹. We did not attempt to replicate the specific risk genes identified in the present study due to lack of power as the clinical sample size is much lower than our 2019 study. We agree with the reviewer that this is not clearly stated in the manuscript. We have therefore edited the text on page 8, which now reads:

“We found a significantly increased overall load of class I variants in cases compared to controls that increased when restricting to constrained genes (OR = 1.24, CI = [1.07, 1.45], P = 0.005; Supplementary Figure 7, Supplementary Table 5). Due to the small sample size, we had no power to discover exome-wide significant genes in this sample, but when focusing on top genes from our discovery analysis we observed a higher effect size point estimate than for constrained genes (OR = 1.42, CI = [1.08, 1.85], P = 0.012, Supplementary Table 6; see also Supplementary Note), suggesting that the identified top genes overall confer more ADHD risk than constrained genes.”

We also agree with the reviewer that it is relevant to report the findings for *MAP1A*, *ANO8* and *ANK2* in the clinical sample. We have added this information to Supplementary table 7 (Supplementary Table 20 in previous version) and added the following to the text on page 8:

“It is noteworthy that the number of deleterious variants in *MAP1A* and *ANK2* was higher in individuals with ADHD compared to controls (no rare deleterious variants were found in *ANO8*; Supplementary Table 7).”

Even though we do not replicate the specific risk genes in ADHD, it is worth noting that nine of the top 20 genes have been identified as rare variant risk genes in autism and neurodevelopmental disorders² (Supplementary Table 4). This is mentioned in the discussion section page 19.

It is not possible to perform the meta-analysis the reviewer suggests because the German controls used in analyses of the clinical sample are included in gnomAD data which are combined with the control data for gene discovery in iPSYCH.

Comment#9

Per the supp. note, the controls in the replication dataset are also in gnomAD, meaning it's not truly an independent set. Why not pull controls from another publicly available datasets (UK Biobank, TopMED, All of US, etc.)?

Response

We wanted to work with high quality data and minimize the effect of population stratification as much as possible. We have therefore specifically chosen to use the German individuals as controls because they have the same genetic ancestry as the clinically ascertained individuals with ADHD.

Comment#10

The IP-MS studies are a nice addition to the paper. Given how prone IP-MS can be to false positives, I urge the authors to knockdown or knock out the three proteins as negative controls.

Response

We thank the reviewer for recognizing the value of our IP-MS studies and for their thoughtful suggestion. While we appreciate the rigor of using knockout models as negative controls, generating such systems is labor-intensive and extends beyond the scope of this study. Additionally, IgG controls are widely regarded in the field as effective in minimizing false positives (doi.org/10.1007/978-1-0716-3445-5_18). Even if knockouts were feasible, *ANO8* is highly loss-of-function (LoF) intolerant, and both *ANK2* and *MAP1A* are similarly constrained, with pLI scores of 1.00 for ANK2, 1.00 for MAP1A, and 0.96 for ANO8¹⁰. Thus, we would likely only achieve heterozygous iPSC lines, which, due to immunoprecipitations inherently enriching for the bait protein, would not serve as effective negative controls. We believe that our IgG controls and the rigor of our protocol robustly address potential false positives in our IP-MS data.

Comment#11

The authors should perform immunofluorescence to demonstrate the purity of their NPC and ExN populations.

Response

We appreciate the reviewer's suggestion to confirm the purity of our NPC and ExN populations via immunofluorescence. However, our differentiation protocol is derived from a widely established and amply published method, initially developed by the Südhof lab (Zhang *et al*, 2013)¹¹ and further validated in our specific cell line by Nehme *et al*. 2018¹². Additionally, Nehme and colleagues have conducted single-nucleus RNA sequencing (snRNA-seq) on these cultures across diverse genetic backgrounds and confirmed both the identity and homogeneity of NPC and ExN populations¹³. Given the well-documented robustness of this protocol and these recent molecular analyses, we believe additional immunofluorescence assays would be redundant and beyond the scope of this manuscript.

Comment#12

Why do the authors switch from defining constrained genes as $pLI < 0.9$ to $LOEUF < 0.6$ for the PPI analyses? I suggest they stick to one definition or test both sets in each constrained gene analysis (including fig. 1).

Response

We initially used the LOEUF scores (without any cutoff) and KS test to assess if the PPI networks were enriched for constrained genes, since LOEUF better captures the whole distribution of constrained genes and thus are more appropriate for the KS test. However, it is true that in the follow-up constraint subletting analysis, binarizing constrained genes with $LOEUF < 0.6$ or $pLI \geq 0.9$ would have served the same purpose. In the revision, we have updated all rare variant enrichment analyses for the PPI networks to use the same definition of constrained genes ($pLI \geq 0.9$ in gnomAD v2.1.1) to be consistent with the rest of the paper as the reviewer suggested. We have updated the results on page 9, the methods on page 30 and added a new Figure 3.c reflecting that pLI scores are used instead of LOEUF scores. The new Figure 3.c is also displayed in the response to comment#14.

Comment#13

For any type of gene ontology analysis, it is critical to define a background set. How did the authors define a background set for their PPI analysis? Unless they are including genes expressed in neurons at baseline, it's hard to interpret.

Response

We previously used the entire protein-coding genome as the background set in both the rare and common variant enrichment analyses of the PPI network genes. Based on this very relevant comment, we have rerun the analyses using ~13k protein-coding genes expressed in neurons as the background set. We defined this background by reanalysing RNA-seq data generated from our iPSC-derived neuronal model (GEO: GSE178896; doi: 10.1016/j.xgen.2022.100250). We only observed minor changes in the results, which means that all enrichment signals we observed were above what we would expect for genes generally expressed in a neuronal cellular context.

We have updated the text in the Results section, page 9, regarding the background genes:

“...and combined network of all three index proteins were significantly enriched (FDR < 0.05, one-tailed Kolmogorov-Smirnov tests) for rare-variant risk genes associated with ASD and DD in both NPC and ExN compared to other protein-coding genes expressed in the neuronal cell model (Figure 3.b; Supplementary PPI-Tables 7-8)”.

We have added a new Figure 3.c demonstrating the results from using neuron expressed protein coding genes as background:

New Figure 3.c demonstrating results from the PPI networks enrichment analyses when using neuron expressed protein coding genes as background.

We have updated the Methods section, page 30, with the description of how the new background set of genes expressed in neurons was identified:

“For subsequent network enrichment analyses, we compared the network genes against a background set of protein-coding genes expressed in neurons (hereafter “neuronal background”). To define the neuronal background, we reanalyzed RNA-seq data derived from the same ExN cellular model used in this study (days 21 and 51 data from GEO: GSE178896)¹⁴. We first performed transcript quantification from FASTQ files using salmon (v1.10.2)¹⁵ and GENCODE v43¹⁶ reference files. We summarized the quantification results to gene-level counts using tximport (v1.26.1), then removed

non-protein-coding genes and low-count genes using the filterByExpr function in edgeR (v3.40.2). This resulted in a list of 13,018 neuronal background genes (Supplementary PPI-Table 7) to be used in downstream analyses.”

We have included a new table, Supplementary PPI-Table 7, demonstrating the list of ~13k protein-coding genes expressed in neurons used as background genes, and updated Supplementary PPI-Tables 8-10 with the results from the enrichment analyses using neuronally expressed genes as background.

Finally, we have updated the description of the mass spectrometry methods to better reflect how the data for this study were generated, in the Supplementary Information pages 5-6.

Comment#14

Why did the authors only focus on midbrain regions for their single-cell enrichment analysis? Given the number of cortical single-cell datasets (both from human embryos, post-mortem, and iPSC-derived neurons), I'd urge the authors to consider these regions as well.

Response

We agree with the reviewer that several studies have highlighted the involvement of the cortex in ADHD. However, in the present study, we were underpowered to conduct hypothesis-free analyses across many cell types, as our rare-variant association signals were relatively modest, with only three genes reaching exome-wide significance. Therefore, we opted to perform a hypothesis-driven single-cell risk score (scDRS) analysis, focusing specifically on dopaminergic neurons, to attempt replication of findings associated with common variants.

Studying dopaminergic neurons derived from midbrain iPSCs is highly relevant, as the cell bodies of these neurons are predominantly located in the midbrain, with axons projecting to various other regions of the brain. E.g. dopaminergic neurons in the ventral tegmental area of the midbrain project to the prefrontal cortex¹⁷, and have been associated with motivation¹⁸, learning^{19,20}, and memory²¹.

However, based on the suggestion from the reviewer we have complemented our hypothesis-based analyses with a hypothesis free analysis exploring the scDRS across a snRNA-seq data set comprising 382 neuronal cell type clusters from the entire human brain²². We have added the following to the result section pages 12-13:

“We also explored the scDRS across 382 cell type clusters representing a spectrum of neurons from the entire human brain²², with no significant findings (Supplementary Table 13). One cluster represented dopaminergic neurons (scDRS P-value = 0.054), which did not reflect the strong signal observed for the iPSC derived dopaminergic neurons. This could be due to differences in neuronal age across data sets (neurons from postmortem brains²² vs iPSC²³).”

We have added the following to the Methods section, page 36: “Furthermore, we did a hypothesis free approach where we evaluated the scDRS in 382 cell type clusters (2,480,956 cells analysed) representing a spectrum of neurons from the entire human brain (Study III)²². This data set consists of snRNA-seq data of neurons desiccated from four adult postmortem human brains from 105 locations across the forebrain (cerebral cortex, hippocampus, cerebral nuclei, hypothalamus, and thalamus), midbrain, and hindbrain (the pons, medulla, and cerebellum). We used the cell type cluster definitions as reported in the original paper²².”

We added the results from scDRS analysis of the 382 clusters to Supplementary Table 13.

Comment#15

In addition to showing the single-cell and BrainSpan expression patterns for the top 100 genes, I recommend the authors specifically plot the expression patterns of the three significant genes.

Response

We have added a new plot (Figure 4.b) which demonstrates the expression of each of the three significant genes across brain developmental stages. We have added the following to the result section page 11: "..., but the three significant genes demonstrated different expression patterns across brain developmental stages with *MAP1A* being expressed significantly higher postnatally than prenatally ($P = 2.2 \times 10^{-16}$) and the opposite was observed for *ANO8* ($P = 2.2 \times 10^{-16}$, Figure 4.c)".

“Figure 4.b.

b. Expression of the three exome-wide significant genes across neocortical brain developmental stages in data from BrainSpan. “*” next to the gene name at the right side indicates significant differences in pre- and postnatal expression”

We have added the following to the Methods section, page 35:

“Likewise, each of the three exome-wide significant genes (*MAP1A*, *ANK2*, *ANO8*) were tested for differences in pre- and postnatal expressions. “

Furthermore, we have performed the BrainSpan analysis testing for increased expression of the top associated genes across brain developmental stages using genes expressed in neurons (listed in the new Supplementary PPI-Table 7) as background. Only minor changes to the results were observed, which are presented in the new Supplementary Figure 10 (new results added to Supplementary Table 12). We have added the following to the Results section, on page 11:

“These ADHD risk genes demonstrated significantly higher mean expression within 11 of the 12 brain developmental stages (prenatal to adult) compared to the average gene expression and in 10 out of 12 stages when compared to genes expressed in neurons (Supplementary Table 12; Figure 4.a, Supplementary Figure 10). These results contrasts....”

New Supplementary Figure 10:

“Supplementary Figure 10. Expression of ADHD risk genes across brain developmental stages compared to neuronal expressed genes

Mean Expression ln(Reads Per Kilobase Million [RPKM] + 1)) of 17 ADHD risk genes and background genes across neocortical brain developmental stages in data from BrainSpan. Background genes include genes expressed in neurons (the genes listed in Supplementary PPI-table 7) but not the 17 tested genes. “*” Indicate significant difference between the two genes sets at a given developmental stage (P-value = 4.17×10^{-3} was considered significant correcting for 12 brain developmental stages).”

Comment#16

Finally, I think the authors could provide more details on the phenotypes of MAP1A, ANO8, and ANK2 carriers. For example, were these individuals enriched for more severe forms of disease/early-onset disease?

Response

We already had a brief description and discussion of comorbidities observed in individuals with rare deleterious variants in the three significant genes in the previous version of the manuscript, but we agree with the reviewer that it will be informative to give more information about these carriers. We have therefore investigated this further and included the following in the Supplementary Information, page 2:

“Phenotype of individuals with rare deleterious variants in MAP1A, ANO8 or ANK2

We identified 33 individuals carrying at least one rare class I variant in the three significant genes (MAP1A, ANO8, and ANK2) and conducted a detailed breakdown of their commodities. Specifically, we examined the presence of ADHD, ASD, schizophrenia (SZ), and intellectual disability (ID)

diagnoses in these carriers. More than half of the carriers were comorbid with at least one of ASD, SZ, and/or ID (Supplementary Figure 6).

To explore disease severity further, we compared “age of ADHD diagnosis” of individuals who were carriers with non-carriers. No significant difference in age of ADHD diagnosis was identified (carrier ADHD: mean age at diagnosis = 12.6 years, standard error (s.e.) = 1.07; non-carrier ADHD: mean = 12.6 years, s.e. = 0.067; Wilcoxon rank-sum test, P value = 0.96.”

We have included a new figure, Supplementary Figure 6:

“Supplementary Figure 6. Comorbidities of individuals with class I variants in *MAP1A*, *ANO8* and *ANK2*

Phenotypic breakdown of individuals carrying at least one rare class I variant. The categories ADHD_ASZ_ID indicate the presence (1) or absence (0) of diagnoses for ADHD, ASD, schizophrenia (SZ), and intellectual disability (ID), respectively. For example, 0_0_0_0 represents controls without any of the listed diagnoses, 1_0_0_0 indicates individuals diagnosed with ADHD only, 1_0_0_1 represents individuals diagnosed with both ADHD and ID, and 1_1_1_1 corresponds to individuals diagnosed with ADHD, ASD, SZ, and ID.”

We have linked the supplementary text to the phenotype evaluation in the main text, page 7:

“More details on the phenotypes of individuals with class I or class II variants in the three risk genes, can be found in the Supplementary Information and Supplementary Figure 6.”

The discussion pages 18-19, have been updated to include information about comorbid schizophrenia (besides ASD and ID):

“For *MAP1A* around 50% and for *ANO8* around 30% of the association signal was driven by rare deleterious variants in individuals diagnosed with ADHD only (without comorbid schizophrenia, ID or autism), while the *ANK2* signal was largely driven by ADHD with co-occurring autism or ID, as around 90% of ADHD individuals with *ANK2* rare deleterious variants had these comorbidities.”

Comment#17

Minor comments

There are lots of different cohorts/analyses in this paper, and it becomes a bit hard to follow. I would recommend a schematic to show the overall study design.

Response

We have added a new figure (Supplementary Figure 1) with an overview of the study design. We hope that this new figure makes it easier to follow what has been done:

“Supplementary Figure 1. Overview of study design and quality control

a. Gene discovery

In total, 8,895 ADHD cases (entirely iPSYCH) and 53,780 controls (mostly gnomAD)

b.

c.

Study design and analytical approach for gene discovery. (a) Sample sizes used in the gene discovery analysis (NFE = non-Finnish European). (b) Definition of rare variant classes. (c) Overview of the analytical approach for gene discovery. The numbers of genes illustrated correspond to autosomal genes. The same approach was applied to genes on the X chromosome, and the corresponding results are described in the Results section.”

Comment#18

The main text lists Supp Table 8 as the replication analysis table, but this seems incorrect.

Response

We thank the reviewer for noticing this inconsistency. The table number has been corrected in the text.

Comment#19

I would suggest you include the summary stats for each gene MAP1A, ANO8, and ANK2 in the main text for readability.

Response

We have added the information on page 7.

Comment#20

I was intrigued that ANO8 had way fewer interactions than ANK2 and MAP1A. Do the authors have any idea why this might be?

Response

We appreciate the reviewer's interest in the differential interaction counts for ANO8, ANK2, and MAP1A. This discrepancy may partly reflect antibody specificity, as variations in recognition, accessibility, and binding affinity for each epitope can influence interaction capture. However, the primary factors are likely the expression levels and distinct biological roles of each protein. ANK2 and MAP1A are highly abundant, large proteins with prominent structural roles—ANK2 (220KDa) in association with actin, and MAP1A (320KDa) with microtubules. In contrast, ANO8 (130KDa) is involved in a more specialized calcium signalling pathway, which may inherently limit the breadth of its interactions.

Comment#21

Figures 5 and 6 would benefit from having colour legends in the figures themselves.

Response

We have made new versions of Figure 5 and 6, with colour bars included in the figures.

Referee #3 (Remarks to the Author):

A. Summary of key results: This article presents results from an analysis of exome sequence data from individuals diagnosed with ADHD (n ~ 9K) and controls (n ~ 54K). They identify three genes with rare deleterious variants that are associated with substantially elevated odds of ADHD diagnosis and conduct a series of bioannotation and follow-up analyses to probe biological mechanisms implicated by these genes.

Comment#1

B. Originality and significance: Understanding the genetic etiology of ADHD is a topic of substantial public health significance, but the significance of the question is not very well described in the paper. Right now, the Introduction reads as if it is written for a genetics or psychiatric journal; that is, it is written as if the audience is already familiar with the public health impact of ADHD and the importance of genetic studies. ADHD is often trivialized/stigmatized as merely a label for "bad kids" rather than a psychiatric condition that results not just in lower education and lower socioeconomic attainments, but also with injuries, criminal justice system involvement, earlier mortality, etc.

There is a need for novel and more effective pharmacologies with lower potential for abuse. All of that to say, I am sold on the significance of this research study, but I wish the authors would make the case of its importance more clearly.

Response

Thank you for the comment on the significance of the study, we appreciate that.

We agree with the reviewer that it would be informative to the general reader of Nature to have more information about the severe outcome associated with ADHD. We have therefore added the following text to the introduction, page 3:

“The disorder is linked to a variety of serious outcomes, including higher risks of substance use disorder^{24,25}, accidents²⁶, premature death²⁷, unemployment²⁸, incarceration and crime²⁹, suicide³⁰, and

metabolic conditions^{31,32}. Gaining insight into the biological mechanisms that drive the disorder is crucial for understanding how it develops and how it may be treated in the future."

Comment#2

Instead, they present the rationale for studying rare variation as "necessary to explain the heritability more fully." In the very next sentence, they report that rare variants have been found to explain only small fractions of the population variance in SZ and BD, so they likely knew going into this project that they were not going to substantially account for the population heritability with rare variants. The rationale for why these analyses are important needs to be described more effectively.

Response

The rationale for focusing on rare variants is described in the introduction, page 3:

"Although rare coding variants only explain a minor part of the overall liability, they can confer substantial risk individually and, in contrast to common variants, identifying them often directly pinpoints the affected causal gene and the likely functional consequence (e.g. predicted protein loss-of-function), providing clues to the underlying etiology of ADHD."

In other words, rare variant analysis often pinpoints the causal genes, the likely genetic consequences and affected biological mechanisms. The identified genes and pathways might also be affected in individuals without rare deleterious variants, but due to other risk factors such as common variants or environmental factors.

We have corrected the sentence the reviewer highlights (on page 3) to reflect that rare variants only explain a small proportion of the heritability. We have deleted "more fully", so the sentence now reads:

"...however, an investigation of rare variants is also necessary to explain more of the heritability".

Comment#3

(On a more minor note, I would caution the authors against using "genetic" (second sentence of the Introduction) as a synonym for "heritable", especially in a general interest journal.)

Response

We have modified the sentence which now reads:

"A large proportion of ADHD risk can be explained by genetics, with an estimated twin heritability of 77 - 88%³³, rooted in both rare and common risk variants."

Comment#4

Regarding originality, these results extend their previous analysis of rare variants in ADHD individuals from the iPSYCH cohort (2019 paper in Nature Neuroscience). They have added more ADHD cases (a little more than doubled) and continue to use gnomAD as a control data set. The 2019 paper also identified MAP1A as an exome-wide significant gene in relation to ADHD, which seems to contradict their claim that this paper identifies "the first three genes implicated in ADHD by rare coding variants from exome sequencing."

Response

In our previous study (Satterstrom et al. Nature Neuroscience 2019), we identified *MAP1* as a cross-disorder risk gene for ASD and ADHD. This result was based on the observation of 11 rare deleterious

variants in cases with five observed in ASD, five in ADHD and one in ASD+ADHD. The previous significant association was therefore not generated exclusively by rare variant analysis in individuals diagnosed with ADHD and no significant association was observed when analysing ADHD separately.

Comment#5

C. Data and methodology: These analyses use Danish register data to identify adults who have been diagnosed with ADHD and compared them to adults with no major psychiatric disorder diagnosis within the same cohort, plus other controls from gnomAD, which removes samples that have been in psychiatric studies. My major question is to what extent they are analyzing the genetic etiology of ADHD, specifically. They conduct several tests of whether their results are driven by comorbid ID, some additional tests for ADHD comorbid with autism or schizophrenia – but none of these are the psychiatric conditions most commonly comorbid with ADHD. Twin studies consistently find that about $\frac{1}{3}$ to $\frac{1}{2}$ of the heritable variation in ADHD is shared with disruptive behavior disorders of childhood (CD, ODD) and with symptoms of ASPD and SUDs in adulthood. Genomic SEM models have found converging results. Is this paper an analysis of ADHD, or of disinhibited/disruptive behavior disorders more broadly, of which ADHD is just the most readily available indicator in register data?

Response

This is a very relevant comment thank you for addressing this. We have now performed gene set analyses evaluating the load of rare deleterious variants in individuals with comorbid SUD or disruptive behaviour disorders (DBDs) which include CD and ODD. This has been done for (1) gene sets stratified by their pLI score and (2) the gene-sets representing risk genes identified in rare-variant studies of SCZ, DD and autism.

We have not specifically evaluated ASPD as the sample size was too small, only 112 individuals with ADHD have comorbid ASPD.

We have added the following two sentences to the results section page 16, regarding the new results:

“A significantly increased load of class I variants was observed in less constrained genes ($pLI \leq 0.5$) for ADHD comorbid with SUD (Supplementary Figure 13.h., Supplementary Table 19).”

“The seven rare-variant risk gene sets showed increased load with some specificity towards comorbidity for the disorders they were most related to, but not for other co-occurring conditions. To specify, there were no significant findings in analyses of comorbidity with DBDs or SUD, but when comparing ADHD comorbid with schizophrenia we found”

We have added two new Supplementary Figures (Supplementary Figures 13.g and 13.h) demonstrating the load of rare deleterious variants in pLI gene-sets across comorbidities with SUD and DBDs (and no ID).

New added figures to Supplementary Figure 13:

We have added two new Supplementary Figures (Supplementary Figures 14.g and 14.h) demonstrating the load of rare deleterious variants in ASD, SCZ and DD risk gene-sets across comorbidities with SUD and DBDs (no ID):

New added figures to Supplementary Figure 14:

Additionally, we have redone all the analyses (and figures) evaluating the impact of multimorbidities where we now have included diagnoses of DBDs and ASPD (SUD was already considered together with other psychiatric disorders in the previous version). Only minor differences in the results were observed compared to the previous analyses.

We have added information about the new ICD10 codes used in the comorbidity analyses to the Methods section, page 23 and 24:

“For the comorbidity analyses we identified individuals with the following ICD10 diagnosis codes in the Danish Psychiatric Central Research Registry: ID (ICD10: F70, F71, F72, F73, F78, F79), autism spectrum disorder (as above), schizophrenia (as above), disruptive behavior disorders (including conduct disorder and oppositional defiant disorder, ICD10: F91, F90.1), substance use disorders (ICD10: F10.1-9; F11.1-9; F12.1-9; F13.1-9; F14.1-9; F15.1-9; F16.1-9; F17.1-9; F18.1-9; F19.1-9) and multi-comorbidities which, besides the comorbidities already listed, also included comorbid anxiety (ICD10: F40.0-F40.2, F41.0-F41.1, F42, F43.0-F43.1), Tic disorder (ICD10: F95), bipolar disorder (ICD10: F30-F31), major depressive disorder (ICD10: F32-F33), anorexia nervosa (ICD10:

F50.0), developmental disorders (ICD10: F80-F83) and anti-social personality disorder (ICD10: F60.2).”

Comment#6

Relatedly, they analyze the association between rPTVs and socioeconomic attainments within individuals with ADHD – but some have argued that ADHD is under-diagnosed, particularly among girls/women who are more likely to have inattentive compared to hyperactive subtypes, and among those whose school performance does not fall below what is “typical” for their age. Have the authors considered analyzing the relationship between ADHD-associated rPTVs and socioeconomic status in individuals who have not been diagnosed with ADHD (or in the full sample regardless of ADHD diagnosis)? Are there any sex differences in this relationship?

Response

The analysis the reviewer is proposing is indeed interesting and relevant, but it is outside the scope of the present study to perform an investigation of the role of rPTVs and socioeconomic status in individuals not diagnosed with ADHD.

Additionally, data on socioeconomic status is considered very sensitive and are only available for analysis on the servers of Statistics Denmark (<https://www.dst.dk/en/TilSalg/Forskningsservice>). The server we have available at Statistics Denmark cannot handle the full iPSYCH rare variant data, which is why only rPTVs on individuals with ADHD have been uploaded and analysed. Besides that, the genetic data also needs to go through a time-consuming approval process before the data are uploaded (can take several months). All together this means that it is not technically possible for us to perform the suggested analyses of control individuals.

Comment#7

D. Statistical tests and treatment of uncertainties: The authors conduct many statistical tests and correct for multiple testing, throughout. My question concerns their “ADHD gene discovery” analyses. They restricted their analyses to consider only genes with a higher rate of variants in cases versus controls, which was 24% of genes for “class I” variants and 7% of genes for “class II” variants – because they expected was that variants that disrupt biological processes would be increased in cases compared to controls. Perhaps I’m just misunderstanding something basic here, but this feels like selecting the data that fits your hypothesis before you test your hypothesis. (If I want to prove that needles are more common in haystack #1 versus haystack #2, wouldn’t taking out the needles that are more common in haystack #2 bias that test?) The authors then use the total number of genes (15,603), not just the ones that were “considered”, for their Bonferroni correction – are there Bonferroni significant genes that have a higher rate of variants in controls versus cases? Overall, I found this approach confusing and would appreciate a clearer explanation and rationale.

Response

We need to be very strict in our QC when combining iPSYCH data with gnomAD data, to be sure that an observed higher load of rare deleterious variants in ADHD is not driven by a generally higher detection/inclusion rate of variants in cases compared to controls caused by differences in sequencing platforms or QC. We therefore included the filtering step where we require the number of synonymous variants in a gene to be higher in controls compared to cases. This strategy, however, also infers that we cannot report results for genes with a higher number of rare deleterious variants in controls compared to cases since such a result could be caused by the controls having a generally higher load of variants (due to the filtering). Consequently, a second filtering step excluded genes with more rare deleterious variants in controls compared to cases. In other words, we are not selecting the data that fits our hypothesis before we do the test. The data included for association analysis are a consequence of our stringent filtering strategy to avoid false positives. We have added the following

text to the Methods section, page 28, to explain better why only genes with a higher load of rare deleterious variants in cases compared to controls were considered:

“We did not consider genes with higher number of class I and II variants in controls, since such observations could be caused by a generally higher number of variants in controls due to the previous filtering step where only genes with a higher number of rare synonymous variants in controls compared to cases were retained.”

We could reduce our significance threshold, but we prefer to be conservative and correct for the number of genes before filtering when claiming exome-wide significance.

Comment#8

E-F. Conclusions: robustness, validity, reliability. Suggestions for improvement. Beyond the considerations I’ve already described (ADHD comorbidity, statistical strategy for discovery analyses), the biggest challenge to the robustness of this study’s results is its minimal consideration of sex differences. In population studies, males are diagnosed with ADHD 2:1 compared to females; in clinical samples, that sex ratio can be 4:1. Basic information on sex is lacking (e.g., what percentage of ADHD cases versus controls are male and female?) Even if there is not sufficient power to conduct sex-stratified analyses, or if the authors expect minimal sex differences in the etiology of ADHD, that would be important to note and explain.

Response

Information about the number of males and females in iPSYCH and the clinical sample were already included in Supplementary Table 1. We have now included this information also for gnomAD. Sex stratified analyses are already included and described in the main text page 6 and results presented in Supplementary Figure 5:

“No difference in the number of class I, class II, or synonymous variants was observed between males and females with ADHD, either before or after excluding individuals with ID (Supplementary Figure 5). This suggests a similar overall burden of rare variants in the two sexes, in line with what is observed for the common variants^{34,35}.”

Additionally, we have now added X chromosome analyses, which also include sex-stratified analysis, and the results are shown in a new Supplementary Figure 9.

Comment#9

G. References. The references are appropriate. Overlap with their 2019 paper in terms of sample and findings (including MAP1A results) could be described more clearly.

Response

The sample overlap of the current study with the previous is already stated in the results section, page 4:

“Our analysis included 8,479 individuals (4,378 with ADHD and 5,002 controls) previously analyzed in our 2019 study of rare variant risk in autism and ADHD¹.”

We have included more information in the discussion section, page 19, about the previous *MAP1A* results:

“*MAP1A* and *ANO8* have not been associated with ASD or other neurodevelopmental conditions in the most recent studies^{2,3}, although *MAP1A* was significantly associated when combining rare deleterious variants in both ADHD and ASD, but not significant in the disorders separately¹.

Comment#10

H. Clarity and context: lucidity of abstract/summary, appropriateness of abstract, introduction and conclusions. I found the paper to be clearly written, overall. It has, at times, a bit of a “laundry list” feel, where results are itemized with little guidance about what the reader is supposed to take away as the most important point. For instance, in the “Biological Implications” section, the authors end the section by reporting that the top genes were not more expressed pre-natally compared to post-natally, and had elevated expression within 11 of 12 brain developmental stages. Is this surprising, or different than what is found when examining common variation? How does this fit with models of ADHD as a “neurodevelopmental” disorder? This is just one example where the *implications* of the results could be expressed with greater lucidity.

Response

We have added a takeaway message regarding analysis of BrainSpan data on page 11:

“This contrasts findings for genes identified by common variants, which were significantly enriched among genes expressed prenatally (19th post conceptual week)⁴.”

How the BrainSpan analyses fit with ADHD being a neurodevelopmental disorder is addressed in the discussion, page 20/21:

“Additionally, we found that the top 20 risk genes had a high expression across all brain developmental stages suggesting the genes play important roles in both development and function of the brain across the lifespan. In accordance with this, we identified both immature neurons (GABAergic and medial neuroblasts) and mature GABAergic and dopaminergic neurons to be enriched for expression of rare-variant risk genes, as mentioned above.”

We have added a concluding remark on the sex specific analyses on page 6:

“This suggests a similar overall burden of rare variants in the two sexes, in line with what is observed for the common variants^{34,35}.”

Referee #4 (Remarks to the Author):

A. Summary of the key results

Comment#1

B. Originality and significance: if not novel, please include reference
>> new analysis of previous data, so some novelty in insights gained

Response

The reviewer writes that we performed new analyses of previous data. This does not capture the breath of our study. We have more than doubled the number of individuals with ADHD in the

iPSYCH cohort compared to the previous study and included unpublished exome-sequencing data from more than thousand clinically ascertained individuals with persistent ADHD. A strength that has been highlighted by two other reviewers.

Comment#2

C. Data & methodology: validity of approach, quality of data, quality of presentation (I will only comment on technological/methodological aspects, as the biological field is not my own) >> methods sections, in particular those pertaining to reanalysis of snRNAseq data, are completely missing. There is no information about:

1. which datasets were used - a basic reference and comment that data were obtained from the GEO is inappropriate - for e.g., the Sarkar dataset contains 7,584 samples, while the La Manno dataset contains 6,179 - which of these 13,763 samples were chosen for reanalysis? (this is not including all other datasets reanalysed in this manuscript). There is also additional concerns/missing information about how data was normalized for different sequencing chemistries, sequencing platforms, sample collection methods, and reference genome use. There was no description of QC metrics, or where they shown, exclusion criterion, etc.

Response

We used already QCed single cell/single nucleus sequencing data from published papers, made available for download by the authors of these publications. We believe it is in line with common practice to focus on what we have done with the data in our manuscript, and not go into details on data generation and QC if we have not done anything beyond what is already described in the published papers.

We have described all our procedures with respect to data handling and additional QC performed after download of the data; e.g., we have described the additional filtering we have performed in two of the data sets so that only genes represented in a minimum of 30 cells were included and described in detail how UMAPs were generated (Methods page 36).

We are not using a data set from Sarkar, so we are not sure what the reviewer is referring to. Additionally, the reviewer is mentioning 6,179 samples in the La Manno data set. We are not sure what this number refers to – we have already stated in the manuscript the number of cells analysed (1,695 cells), and these cells comes from ten human embryos (6–11 weeks old). We have added information about the number of embryos from which the cells were derived on page 35:

“... one scRNA-seq data set from a study (study I) of gene expression in prenatal human brain cells³⁶ derived from ten human embryos 6 -11 week old (1,695 cells analyzed),...”

We have also added more information about the Jerber data used, on page 33:

“These data were generated from 215 pluripotent stem cell (iPSC) lines each derived from a single healthy donor for differentiation towards midbrain neuronal cell types.”

When linking ADHD risk genes to the single cell sequencing data for calculating scDRS, the data were linked by gene name, so potential differences in the reference genomes used has no influence on the analysis.

Comment#3

2. complete lack of orthogonal validation that key ADHD DEGs are in fact real and not artifacts of sampling error or sequencing.

Response

We have not done DEG analyses, so we are not sure what the reviewer is referring to. Maybe the reviewer is referring to the variably expressed genes identified and used for PCA and subsequent generation of UMAPs. Generation of these plots are not related to analysis of ADHD risk genes, only for clustering and visualization of cell types.

Comment#4

>> there are also profound errors in data presentation - in particular a Manhattan plot with no individual SNP data plotted, no statistical cutoff, or other standard representation points.

Response

Figure 2 presents gene-based P-values and is a standard way of presenting results from rare-variant analyses; e.g. our figure is similar to Figure 2.a in the Nature paper reporting results from rare-variant analyses of schizophrenia³.

We have not included a figure demonstrating results from a single variant analysis, because we are underpowered to detect single-rare-variant associations.

Comment#5

>> missing data in figures also concerning - e.g. what are heatmap/scales representing in UMAP plots of snRNAseq data - z-score, expression values, something else?

Response

The scale of the heatmap represents the scDRS and it is already explained in the Figure 4 legend how it should be interpreted:

“The strength of the scDRS, is indicated by the bar at the right side. Red indicates a positive score reflecting increased expression of ADHD rare variant risk genes compared to the distribution of expressions of control gene sets, blue indicates a negative score reflecting decreased expression of ADHD rare variant risk genes compared to the distribution of the expression of control gene sets in a cell.”

We have made a new version of Figure 4.c where the right-hand y-axis has been streamlined across the three plots and added information in the figures, about what the scale bar on the right-hand y-axis represents.

Comment#6

>> no details of what 'significant' or 'background' genes are in Figure 4, panel A.

Response

The background genes are explained in the Methods section and in the Figure 4 legend. We have now added information about the number of genes/transcripts used as background to the legend:

“Background genes (22,402 genes/transcripts) include genes expressed in BrainSpan but not the 17 tested genes”.

We would like to thank the reviewer for pointing out that information about correction for multiple testing is missing. We have therefore added information about the P-value threshold for significance to the Method section page 35:

”..and a two-sided paired t-test was used to test for differential expression of sets of the 17 ADHD risk genes against a background gene-set (22,402 genes/transcripts) for all developmental stages merged and across developmental stages (P-value = 4.17×10^{-3} was considered significant correcting for 12 brain developmental stages).”

Additionally, we made a new version of Figure 4, which now indicates brain developmental stages with significant increased expression of ADHD risk genes compared to background genes, and added the following text to the Figure 4 legend:

“*Indicate significant difference between the two genes sets at a given developmental stage (P-value = 4.17×10^{-3} was considered significant correcting for 12 brain developmental stages).”

Furthermore, we have added information to the legend of Supplementary Table 12, about the P-value threshold for declaring significance.

Comment#7

The proteomics sample prep is mostly fine. A general lack of clear methods permeates this section of the manuscript as well. And a lack of clear descriptions of how previous datasets are normalized for integration here. I am also unclear about the power of these datasets as, like the sequencing integration, no clear statement is made about which samples from the numerous published datasets are used.

Response

We have updated and reorganized the PPI networks methods section (and corresponding Supplementary Information) to improve clarity. Specifically, the ANK2 PPI networks were derived from four IP-MS datasets previously published by our group (Table S2 of Pintacuda et al¹⁴), while the MAP1A and ANO8 networks were derived from four new IP-MS experiments performed in this study (Supplementary PPI-Tables 1-4). Each IP-MS dataset was analysed independently to define significant interactors for the index protein ($\log_2 \text{FC} > 0$ and $\text{FDR} \leq 0.1$, based on 2-3 replicates), before all identified interactors were consolidated into networks for downstream analyses.

Comment#8

>> There is no QC metrics about sample inclusion, not statement about sex, age, genetic predisposition, etc., in addition to sample sizes, which make interpretation of the comparison data difficult.

We are not sure which data the reviewer is referring to. The QC of the sequencing data is described in detail in the method section pages 23 – 25 and outlined in Supplementary Figure 2. Sample size and number of females and males are stated in Supplementary Table 1, comorbidity information in Supplementary Table 18, and we have now also added information about number of females and males in gnmAD.

We have included a new Supplementary Figure 1, with an overview of the study design.

Comment#9

>> On the whole, the level of ambiguity in what data was actually analyzed in this manuscript make review very difficult - the reader has no idea what is included in the analysis.

Response

We are sorry to hear that this is unclear, since we have described in detail how we have done our QC and analyses. The QC procedures are also outlined in Supplementary Figure 2 (Supplementary Figure 1 in the previous version). It would be a help if the reviewer could pinpoint which information is missing/unclear, since the other reviewers do not have the same critique.

We have generated a new figure (Supplementary Figure 1) to improve readability and clarity of the study design.

Comment#10

>> minor point: Fig 2 scale bar has an error in the title. I think the square (before $\log_{10}(P)$) should be 'delta'? hard to say.

Response

In our version of the figure the scale bar is in good quality and the text is clear ($-\log_{10}(P\text{value})$) but if the editor wants a larger font, we will be happy to provide that.

Comment#11

D. Appropriate use of statistics and treatment of uncertainties

>> inappropriate due to lack of methods descriptions. Unable to review

Response

We are very surprised by this critique. In our opinion the method section is describing in detail what has been done at a level similar to our previous rare-variant publication in Nature Neuroscience (Satterstrom et al., 2019¹) and what has been done in other exome-sequencing papers, e.g. the schizophrenia rare-variant Nature paper³ and the bipolar rare-variant Nature Genetics paper³⁷.

Comment#12

E. Conclusions: robustness, validity, reliability

>> unable to review or comment on due to numerous errors in generation and presentation of data.

Response

We disagree with this critique, which also contrasts with the other reviewers. We have performed stringent QC of the data generated, analysed all data using state-of-the-art methods and carefully checked the presentation of data and results for errors. We would appreciate if the reviewer would specify the errors they have observed.

Comment#13

F. Suggested improvements: experiments, data for possible revision

>> the authors need to include appropriate methods in their manuscript - simply stating the same metrics and approaches were used as in previous studies (e.g. line 577) is inappropriate - particularly given this new manuscript is asking a completely different question with a large patient cohort and as such likely has different exclusion criteria. This omission alone should exclude publication.

Response

The manuscript includes a full description of the data generation and QC also outlined in Supplementary Figure 2 (Supplementary Figure 1 in the previous version). We have deleted the word "briefly" from the sentence "To recap briefly", which wrongly could indicate that we have left out some parts of the data processing. The sentence on page 25, now reads:

“In this study, we have applied the same methods for quality control (QC) as explained in our previous study¹ to an updated data set including new exome-sequenced individuals. To recap, the sequencing of 34,544 individuals from the iPSYCH cohort was performed using the Illumina Nextera capture kit and the Illumina HiSeq sequencer. The sequencing process...”

Comment#14

>> additional lacking experimental details for reanalysis of scRNAseq data (which was actually snRNAseq data) is another major concern. These datasets were generated on different single cell platforms, are aligned to different reference genomes, are sequenced on different Illumina platforms and had different baseline QC metrics applied (which is not surprising given they were generated 2 years apart in a rapidly evolving field). For the analysis presented in this paper to be scientifically sound, raw fastq files should be obtained for all samples, they should be aligned to the same reference genome and appropriate and standardized QC metrics/exclusion criterion applied (and presented in the manuscript). They should be normalized, integrated, and then clustered to ensure appropriate reporting of results. The authors are strongly urged to also orthogonally validate results as output from such analysis.

Response

We have corrected the text describing the Jerber data set, which now states that we use data from single nucleus sequencing.

We performed scDRS analyses separately for each of the sc/snRNA-seq data sets, which is explained in the Method section pages 35-36, so there was no merging of data. We have included a sentence on page 37, to make it clear that the data sets were analysed separately:

“We used the P-values from the top 100 most associated genes from the gene-discovery analysis of iPSYCH+gnomAD samples as input for scDRS analysis which was done separately for the above-described data from study I, II and III.”

We downloaded QCed data as explained in response to Comment#2. Additional filtering of data done by us is already described in the method section on page 36. Information about which genes that are ADHD risk genes was alone based on gene name and therefore potential differences in reference genomes is not influencing the analyses.

Comment#15

Figure 3, panel a is not helpful without mapping/detailing connecting genes, completing GSEA or other appropriate analyses and attempting to build out a mechanistic framework.

Response

We included Figure 3.a as a visual summary of the PPI data we generated, emphasizing that there was convergence both between the ADHD risk proteins and between the two neuronal cell types, and that many of the observed PPI have not been reported in the literature. We agree with the reviewer that this panel alone does not provide details that explain the biological implications of the PPI data. However, we have also included extensive supplementary data (Supplementary PPI-Tables 1-10) to further annotate the PPI networks, including prioritizing network genes that have been implicated in genetic studies of ADHD and other neurodevelopmental disorders, and various genetic or gene set enrichment analyses just like the reviewer suggested. We believe these results will facilitate hypothesis generation for follow-up functional studies that focus on understanding the mechanistic links of these PPI.

Comment#16

G. References: appropriate credit to previous work?

>> appropriate, but limited.

Response

We have added the published ADHD *de novo* variant paper to the references: “Olfson, E. *et al.* Rare de novo damaging DNA variants are enriched in attention-deficit/hyperactivity disorder and implicate risk genes. *Nature communications* **15**, 5870 (2024). <https://doi.org:10.1038/s41467-024-50247-7>”.

Comment#17

H. Clarity and context: lucidity of abstract/summary, appropriateness of abstract, introduction and conclusions

>> the manuscript should be carefully read for numerous grammatical errors and confusion sentence structure.

Response

We have gone through the manuscript and corrected grammatical errors, additionally we have made new versions of Supplementary Figures 5, 13 and 14 (the numbering refers to the revised version) so the header of each figure explains more clearly which individuals are analysed.

Comment#18

>>unable to review conclusion appropriateness due to numerous major concerns with methods for generation of data.

Response

We do not think that there are major concerns regarding data generation and applied methods and hope that we have cleared any outstanding issues in present revision.

References

- 1 Satterstrom, F. K. *et al.* Autism spectrum disorder and attention deficit hyperactivity disorder have a similar burden of rare protein-truncating variants. *Nature neuroscience* **22**, 1961-1965 (2019). <https://doi.org:10.1038/s41593-019-0527-8>
- 2 Fu, J. M. *et al.* Rare coding variation provides insight into the genetic architecture and phenotypic context of autism. *Nature genetics* **54**, 1320-1331 (2022). <https://doi.org:10.1038/s41588-022-01104-0>
- 3 Singh, T., Neale, B. M., Daly, M. J. & Consortium, o. b. o. t. S. E. M.-A. S. Exome sequencing identifies rare coding variants in 10 genes which confer substantial risk for schizophrenia. *medRxiv* (2020).
- 4 Demontis, D. *et al.* Genome-wide analyses of ADHD identify 27 risk loci, refine the genetic architecture and implicate several cognitive domains. *Nature genetics* (2023). <https://doi.org:10.1038/s41588-022-01285-8>
- 5 Zhang, Q., Prive, F., Vilhjalmsson, B. & Speed, D. Improved genetic prediction of complex traits from individual-level data or summary statistics. *Nature communications* **12**, 4192 (2021). <https://doi.org:10.1038/s41467-021-24485-y>
- 6 Olfson, E. *et al.* Rare de novo damaging DNA variants are enriched in attention-deficit/hyperactivity disorder and implicate risk genes. *Nature communications* **15**, 5870 (2024). <https://doi.org:10.1038/s41467-024-50247-7>

- 7 Prive, F. Using the UK Biobank as a global reference of worldwide populations: application to measuring ancestry diversity from GWAS summary statistics. *Bioinformatics* **38**, 3477-3480 (2022). <https://doi.org:10.1093/bioinformatics/btac348>
- 8 Avila, M. N. *et al.* Deleterious coding variation associated with autism is consistent across populations, as exemplified by admixed Latin American populations. *medRxiv* (2025).
- 9 Lassen, F. H. *et al.* Exome-wide evidence of compound heterozygous effects across common phenotypes in the UK Biobank. *medRxiv* (2023). <https://doi.org:10.1101/2023.06.29.23291992>
- 10 Chen, S. *et al.* A genomic mutational constraint map using variation in 76,156 human genomes. *Nature* **625**, 92-100 (2024). <https://doi.org:10.1038/s41586-023-06045-0>
- 11 Zhang, Y. *et al.* Rapid single-step induction of functional neurons from human pluripotent stem cells. *Neuron* **78**, 785-798 (2013). <https://doi.org:10.1016/j.neuron.2013.05.029>
- 12 Nehme, R. *et al.* Combining NGN2 Programming with Developmental Patterning Generates Human Excitatory Neurons with NMDAR-Mediated Synaptic Transmission. *Cell reports* **23**, 2509-2523 (2018). <https://doi.org:10.1016/j.celrep.2018.04.066>
- 13 Nehme, R. *et al.* The 22q11.2 region regulates presynaptic gene-products linked to schizophrenia. *Nature communications* **13**, 3690 (2022). <https://doi.org:10.1038/s41467-022-31436-8>
- 14 Pintacuda, G. *et al.* Protein interaction studies in human induced neurons indicate convergent biology underlying autism spectrum disorders. *Cell Genom* **3**, 100250 (2023). <https://doi.org:10.1016/j.xgen.2022.100250>
- 15 Patro, R., Duggal, G., Love, M. I., Irizarry, R. A. & Kingsford, C. Salmon provides fast and bias-aware quantification of transcript expression. *Nature methods* **14**, 417-419 (2017). <https://doi.org:10.1038/nmeth.4197>
- 16 Frankish, A. *et al.* GENCODE: reference annotation for the human and mouse genomes in 2023. *Nucleic acids research* **51**, D942-D949 (2023). <https://doi.org:10.1093/nar/gkac1071>
- 17 Rosenthal, A. & Lin, J. C. *Encyclopedia of Neuroscience, Dopaminergic Differentiation*. 609-613 (Academic Press, 2009).
- 18 Hughes, R. N. *et al.* Ventral Tegmental Dopamine Neurons Control the Impulse Vector during Motivated Behavior. *Curr Biol* **30**, 2681-2694 e2685 (2020). <https://doi.org:10.1016/j.cub.2020.05.003>
- 19 Solie, C., Girard, B., Righetti, B., Tapparel, M. & Bellone, C. VTA dopamine neuron activity encodes social interaction and promotes reinforcement learning through social prediction error. *Nature neuroscience* **25**, 86-97 (2022). <https://doi.org:10.1038/s41593-021-00972-9>
- 20 Keiflin, R., Pribut, H. J., Shah, N. B. & Janak, P. H. Ventral Tegmental Dopamine Neurons Participate in Reward Identity Predictions. *Curr Biol* **29**, 93-103 e103 (2019). <https://doi.org:10.1016/j.cub.2018.11.050>
- 21 Titulaer, J. *et al.* The Importance of Ventral Hippocampal Dopamine and Norepinephrine in Recognition Memory. *Front Behav Neurosci* **15**, 667244 (2021). <https://doi.org:10.3389/fnbeh.2021.667244>
- 22 Siletti, K. *et al.* Transcriptomic diversity of cell types across the adult human brain. *Science* **382**, eadd7046 (2023). <https://doi.org:10.1126/science.add7046>

- 23 Jerber, J. *et al.* Population-scale single-cell RNA-seq profiling across dopaminergic neuron differentiation. *Nature genetics* **53**, 304-312 (2021).
<https://doi.org/10.1038/s41588-021-00801-6>
- 24 Brandt, A., Rehm, J. & Lev-Ran, S. Clinical Correlates of Cannabis Use Among Individuals With Attention Deficit Hyperactivity Disorder. *The Journal of nervous and mental disease* **206**, 726-732 (2018). <https://doi.org/10.1097/NMD.0000000000000877>
- 25 Sundquist, J., Ohlsson, H., Sundquist, K. & Kendler, K. S. Attention-deficit/hyperactivity disorder and risk for drug use disorder: a population-based follow-up and co-relative study. *Psychological medicine* **45**, 977-983 (2015).
<https://doi.org/10.1017/S0033291714001986>
- 26 Ruiz-Goikoetxea, M. *et al.* Risk of unintentional injuries in children and adolescents with ADHD and the impact of ADHD medications: A systematic review and meta-analysis. *Neurosci Biobehav Rev* **84**, 63-71 (2018).
<https://doi.org/10.1016/j.neubiorev.2017.11.007>
- 27 Dalsgaard, S., Ostergaard, S. D., Leckman, J. F., Mortensen, P. B. & Pedersen, M. G. Mortality in children, adolescents, and adults with attention deficit hyperactivity disorder: a nationwide cohort study. *Lancet* **385**, 2190-2196 (2015).
[https://doi.org/10.1016/S0140-6736\(14\)61684-6](https://doi.org/10.1016/S0140-6736(14)61684-6)
- 28 Fleming, M. *et al.* Educational and Health Outcomes of Children Treated for Attention-Deficit/Hyperactivity Disorder. *JAMA Pediatr* **171**, e170691 (2017).
<https://doi.org/10.1001/jamapediatrics.2017.0691>
- 29 Mohr-Jensen, C., Muller Bisgaard, C., Boldsen, S. K. & Steinhausen, H. C. Attention-Deficit/Hyperactivity Disorder in Childhood and Adolescence and the Risk of Crime in Young Adulthood in a Danish Nationwide Study. *Journal of the American Academy of Child and Adolescent Psychiatry* **58**, 443-452 (2019).
<https://doi.org/10.1016/j.jaac.2018.11.016>
- 30 Septier, M., Stordeur, C., Zhang, J., Delorme, R. & Cortese, S. Association between suicidal spectrum behaviors and Attention-Deficit/Hyperactivity Disorder: A systematic review and meta-analysis. *Neurosci Biobehav Rev* **103**, 109-118 (2019).
<https://doi.org/10.1016/j.neubiorev.2019.05.022>
- 31 Chen, Q. *et al.* Common psychiatric and metabolic comorbidity of adult attention-deficit/hyperactivity disorder: A population-based cross-sectional study. *PloS one* **13**, e0204516 (2018). <https://doi.org/10.1371/journal.pone.0204516>
- 32 Akmatov, M. K., Ermakova, T. & Batzing, J. Psychiatric and Nonpsychiatric Comorbidities Among Children With ADHD: An Exploratory Analysis of Nationwide Claims Data in Germany. *J Atten Disord* **25**, 874-884 (2021).
<https://doi.org/10.1177/1087054719865779>
- 33 Faraone, S. V. & Larsson, H. Genetics of attention deficit hyperactivity disorder. *Molecular psychiatry* (2018). <https://doi.org/10.1038/s41380-018-0070-0>
- 34 Martin, J. *et al.* Examining Sex-Differentiated Genetic Effects Across Neuropsychiatric and Behavioral Traits. *Biological psychiatry* **89**, 1127-1137 (2021).
<https://doi.org/10.1016/j.biopsych.2020.12.024>
- 35 Martin, J. *et al.* A Genetic Investigation of Sex Bias in the Prevalence of Attention-Deficit/Hyperactivity Disorder. *Biological psychiatry* **83**, 1044-1053 (2018).
<https://doi.org/10.1016/j.biopsych.2017.11.026>
- 36 La Manno, G. *et al.* Molecular Diversity of Midbrain Development in Mouse, Human, and Stem Cells. *Cell* **167**, 566-580 e519 (2016). <https://doi.org/10.1016/j.cell.2016.09.027>

- 37 Palmer, D. S. *et al.* Exome sequencing in bipolar disorder identifies AKAP11 as a risk gene shared with schizophrenia. *Nature genetics* **54**, 541-547 (2022).
<https://doi.org/10.1038/s41588-022-01034-x>

Referees' comments:

We thank the editor and the reviewers for their valuable comments and suggestions, and hope that the additional analyses we have added, and our responses below will clear any remaining doubts.

Referee #1 (Remarks to the Author):

Comment#1

The authors have been responsive to the majority of my comments. I have just one remaining comment. My first key point concerned whether the selection of samples in this study will be representative of ADHD in the general population (e.g., high rates of ASD and ID). The authors responded showing extensive comparisons between this specific study with the general population of iPsych participants. Still, I would be interested in some data and perspective on how the iPsych sample compares to other cohorts of ADHD patients, beyond the iPsych general population sample. This concerns comorbidity with ASD and ID, but also the question whether iPsych potentially includes more severe ADHD cases compared to other population cohorts.

Response

The reviewer is raising an interesting point, which is not straightforward to answer. Comparing comorbidity prevalences across cohorts and studies is challenging. These estimates can vary substantially due to differences in case ascertainment, clinical practices, assessment procedures, access to healthcare, and even within-country heterogeneity in diagnostic approaches¹. Additionally, factors such as the age distribution of the sample and the period of data collection can further influence prevalence estimates^{2,3}.

In the population-based iPSYCH cohort, the prevalence of ADHD is 1.8%, based on diagnoses made by the end of 2016 (N cohort = 1,657,449)⁴. This estimate reflects diagnostic practices in Denmark at that time. National data show that the prevalence of ADHD in Denmark increased from 0.10% in 2000 to 3.03% in 2022 (N = 7,748,837)⁵, following trends observed in other countries, including the US⁶ and Sweden⁷. These temporal changes underscore the difficulty in directly comparing prevalence and comorbidity rates across settings and time periods.

We have previously examined ADHD and comorbidities in a Swedish population-based cohort (1987–2006 births), identifying a higher ADHD prevalence (4%) and lower comorbidity with ASD (13.9%) and ID (5.2%) compared to iPSYCH. Given the lower ADHD prevalence but higher comorbidity rates in iPSYCH, one hypothesis is that individuals diagnosed in iPSYCH may represent more severe ADHD cases (at least compared to the Swedish cohort) - potentially those at the extreme end of the liability distribution. However, we do not have clinical data on ADHD symptom severity in iPSYCH to evaluate this.

Importantly, while diagnostic approaches differ across countries (e.g., DSM vs. ICD) and inclusion criteria vary between cohorts, the genetic architecture of ADHD appears highly consistent. We have observed strong genetic correlations ($r_g = 0.82–0.93$)⁸ between iPSYCH and other European-ancestry ADHD cohorts, supporting the

validity and generalizability of iPSYCH for genetic analyses of ADHD despite diagnostic differences between cohorts.

So, even though comparison of comorbidity rates between cohorts is interesting we would prefer not to discuss this in the current manuscript, due to the uncertainties and heterogeneities listed above. Nonetheless, we now include the overall ADHD prevalence in the base cohort of 1,657,449 individuals on page 23 to provide context for readers:

“All individuals diagnosed with major psychiatric disorders by the end of 2016 according to the ICD-10 criteria (ADHD [1.8% of the cohort],...)”

Referee #2 (Remarks to the Author):

In their revised article, the authors have addressed most of my major comments. Namely, they have performed burden tests on the X chromosome and replication analyses that overall make their story more complete. However, I remain concerned about the functional data:

Comment#1

• The authors chose not to perform immunofluorescence on their iPSC-derived neurons. While they emphasize that their differentiation protocol is standardized and well-established—which is accurate—this does not eliminate the need to validate the identity and purity of the resulting cell types. It is standard practice in the field to provide basic characterization, irrespective of the protocol used. In this manuscript, the authors do not even include brightfield images, leaving the reader without any confirmation that the cells under study are indeed NPCs or neurons. This omission weakens confidence in the functional assays and is particularly concerning given that their conclusions appear to be drawn from a single differentiation in a single genetic background. I disagree that such basic validation via immunofluorescence is out of scope, especially for a journal of Nature’s caliber.

Response

We thank the reviewer for this suggestion. Although we performed extensive QC of cell identity before our biochemical assays, we omitted those details from the initial manuscript to avoid distracting from its main focus. However, we agree that validation is important and have implemented a two-pronged approach to address the reviewer’s concern.

First, we analyzed single-nucleus RNA-seq data derived from our neuronal cell model to confirm expression of cell type marker genes across iPSC differentiation and demonstrated the results in the new Supplementary Figure 17.a (see figure below). The pluripotency (*POU5F1*, *MKI67*), progenitor (*NEUROD1*, *MSI1*), and neuronal (*MAPT*, *SLC17A6*) markers showed elevated expression in the iPSC, NPC, and ExN samples, respectively, supporting the robustness of our differentiation protocol. Details on the full snRNA-seq dataset are available in our recent preprint where we extended the same proteomic approach applied here to the study of rare variant genes of

schizophrenia (DOI: <https://doi.org/10.1101/2025.05.02.25326757>). We have also added the relevant methods for generating the new Supplementary Figure 17. a in the Supplementary Information, page 4:

“Marker gene expression plots

The snRNA-seq datasets of differentiating neurons spanning iPSC (day 0), NPC (day 4) and ExN (day 31) stages were previously described in [REF⁹]. The data were processed with CellRanger (v6.1.2)¹⁰ using the refdata-gex-GRCh38-2020-A reference from 10x Genomics. CellRanger was run with the following parameters: --expect-cells=10000, --localmem=64, --nosecondary, --chemistry=SC3Pv3, and --include-introns. Nuclei with fewer than 300 detected genes and more than 5% mitochondrial genes were excluded from further analysis. For the remaining nuclei, the percentage of mitochondrial genes was included as a technical variable when performing data normalization using the SCTransform method in Seurat (v5.0.3)¹¹. Data integration across multiple 10x runs were performed via STACAS (v2.2.2)¹², using 3000 anchor features, 30 dimensions, and cell type labels (iPSC, NPC, ExN) for semi-supervised alignment. UMAP projection plots showing log-transformed counts per million (CPM) expression for genes of interest were generated using Seurat.”

Second, we performed immunofluorescence staining on NPC and fully differentiated ExNs to confirm their identities as immature neural progenitors and upper-layer prefrontal cortex neurons, respectively. For NPCs, we used established markers of early neuronal identity, including VIM, a cytoskeletal protein enriched in proliferating neural progenitor cells, and MSI1, an RNA-binding protein involved in maintaining progenitor self-renewal. For ExNs, we used general neuronal markers (NeuN, MAP2) alongside well-characterized markers of upper cortical identity (CUX2, BRN2), the result are demonstrated in the new Supplementary Figure 17 b-e (also shown below). Related methods have also been included in the Supplementary Information page 4:

“Immunofluorescence (IF)

Cells were plated on 96-well microplates (PhenoPlates, Revvity) at the iPSC stage. On the day of the experiment, cells were washed with PBS and then fixed with 2% formaldehyde for 15 min, followed by 5 min of permeabilisation in 0.4% Triton X-100 (TBST). Cells were briefly washed with PBS before blocking with a 1% BSA-TBST solution for 30 min. Primary antibody dilutions were prepared in blocking solution. Cells were incubated with primary antibodies overnight in a humid chamber at 4°C. Slides were washed three times in PBS. Secondary antibodies and conjugated antibodies were diluted in blocking solution and incubated with cells for 45 min in a humid chamber at room temperature. After incubation, wells were washed twice with PBS before incubating with 4,6-diamidino-2-phenylindole (DAPI) for 10 minutes. Before imaging, cells were washed three times with PBS. All antibodies used are listed below.

Gene ID	Vendor & Catalog	Host & Clonality	Usage	Amount in IF
---------	------------------	------------------	-------	--------------

NeuN	Abcam, ab104224	Mouse Monoclonal	Primary	1:1000
MAP2	Abcam, ab183830	Rabbit polyclonal	Primary	1:1000
CUX2	Proteintech, 82933-1-RR	Rabbit polyclonal	Primary	1:1000
TUJ1	Invitrogen, MA1-118	Mouse monoclonal	Primary	1:1000
TUJ1	Invitrogen, 53-4510-82	Mouse Monoclonal	Conjugated Antibody	1:500
Homer1	Synaptic Systems, 160003	Rabbit polyclonal	Primary	1:1000
Synaptophysin	Synaptic systems, 101308	Guinea Pig	Primary	1:1000
Vimentin	Abcam, ab24525	Rabbit Polyclonal	Primary	1:1000
MSI1	Proteintech, 13512-1-AP	Rabbit Polyclonal	Primary	1:1000
MAP2	Abcam, ab302547	Rabbit Monoclonal	Conjugated Antibody	1:500
MAP2	Abcam, ab225316	Rabbit Monoclonal	Conjugated Antibody	1:500
Neun	NBP1- 77686AF488	Rabbit Polyclonal	Conjugated Antibody	1:500
Goat Anti-Rabbit IgG H&L (Alexa Fluor® 488)	Invitrogen, # A-11008	Goat Polyclonal	Secondary Antibody	1:3000
Goat Anti-Mouse IgG H&L (Alexa Fluor® 488)	Invitrogen, #A28175	Goat Recombinant Superclonal	Secondary Antibody	1:3000
Goat Anti-Rabbit IgG H&L (Alexa Fluor® 594)	Invitrogen, # A-11012	Goat Polyclonal	Secondary Antibody	1:3000
Goat Anti-Mouse IgG H&L (Alexa Fluor® 594)	Invitrogen, #A-11032	Goat Polyclonal	Secondary Antibody	1:3000
Goat anti-Chicken IgY (H+L) Secondary Antibody, Alexa Fluor® 647	Invitrogen, #A-21449	Goat Polyclonal	Secondary Antibody	1:3000
Goat anti-Guinea Pig IgG (H+L) Highly Cross- Adsorbed Secondary	Invitrogen #A- 11073	Goat Polyclonal	Secondary Antibody	1:3000

Antibody, Alexa Fluor® 488				
Goat anti-Guinea Pig IgG (H+L) Highly Cross-Adsorbed Secondary Antibody, Alexa Fluor® 647	Invitrogen #A-21450	Goat Polyclonal	Secondary Antibody	1:3000

Fluorescence images acquisition

Cells were imaged on an Opera Phenix High-Content Screening System (PerkinElmer) using Harmony software (v4.9). Initial setup was in wide-field mode with a water-immersion 20x objective. For data acquisition, at least 20 fields per well were captured in confocal mode, acquiring a minimum of six Z-stacks per field at 2 µm intervals. Optimal exposure times for each field and channel were determined manually and then applied uniformly across all experiments. Image processing and refinement were performed in Fiji/ImageJ.”

Additionally we have added the following to the methods section in the main text, page 33:

“We evaluated the expression of cell type marker genes using snRNA-seq and performed immunofluorescence staining on NPC and fully differentiated ExNs to confirm their identities as immature neural progenitors and upper-layer prefrontal cortex neurons, respectively (results demonstrated in Supplementary Figure 17).”

New Supplementary Figure 17 (See supplementary Information for a better quality):

“Supplementary Figure 17. iPSC-derived neurons recapitulate features of human neural progenitor cells (NPC) and excitatory neurons (ExN) *in vitro*

(a) Single-nucleus RNA-seq profiling of iPSC-derived neural cultures reveals expected expression of stage-specific markers: *POU5F1* and *MKI67* for pluripotent cells (iPSC, day 0 of differentiation), *NEUROD1* and *MS11* for neural progenitors (NPC, day 4), and *MAP2* and *SLC17A6* for excitatory neurons (ExN, day 31). CPM = counts per million reads. (b) Immunofluorescence at day 4 of differentiation highlights early neural identity (VIM, MS11). Scale bar: 50 μ m. (c) Quantification of five biological replicates (100 cells per replicate) for each IF condition, normalized to cells with DAPI+ nuclei. (d) Immunofluorescence at day 30 of differentiation shows expression of mature neuronal markers (NeuN, MAP2) and upper-layer identity (CUX2, BRN2). Scale bar: 50 μ m. (e) Quantification of five biological replicates (100 cells per replicate) for each IF condition, normalized to cells with DAPI+ nuclei.”

Together, both the snRNA-seq and immunofluorescence staining data strongly support the cell identity of the NPC and ExN we used to identify protein interaction partners of the prioritized ADHD risk genes.

Referee #3 (Remarks to the Author):

Comment#1

The authors have responded to my previous comments. I especially appreciate the extra work they did to consider comorbidities with disruptive behavior disorders. I do not have further revisions to suggest.

Response

We would like to thank the reviewer for acknowledging the work we have done to improve the manuscript.

Referee #4 (Remarks to the Author):

Comment#1

The authors have made extensive revisions and added a great deal of text both in the response to reviewers as well as in the text, that does a great job of clearing up much of the confusion present in the initial review of the manuscript.

My initial review was unnecessarily aggressive and I apologise for this, the GWAS components of the study remain high quality, and the use of the enlarged dataset will no doubt be widespread."

However, my main concerns remain, largely focused on the way the snRNAseq data have been obtained/analyzed/and interpreted - which continue to be the weakest part of the manuscript.

For simplicity, I respond to the authors comments to each initial concern separately:

Response

We would like to thank the reviewer for appreciating the additional work we have done to improve the manuscript and for acknowledging the high quality of the data generated and the genetic analyses.

Comment #2

-the authors provide a link to the GEO repository for the La Manno dataset (GSE76381) from the Linnarsson lab. This repository contains both mouse and human samples (as described by the authors: Human embryo ventral midbrain cells between 6 and 11 weeks of gestation, mouse ventral midbrain cells at six developmental stages between E11.5 to E18.5, Th+ neurons at P19-P27, and putative dopaminergic neurons at P28-P56), and there are 6179 deposited datasets for download from this repository. Have the authors use all samples in their reanalysis (a quick review of the original manuscript described 10 human embryos were used) - if not, which of the subsamples uploaded to the GEO repository have been used?

-the additional clarifying comments on pages 33 and 35 are appreciated.

Response

Thank you for addressing this, we agree that it is important to be specific about which files we have analysed. For the La Manno data we have used one single file containing processed data from the 10 embryos (we specify page 36 that we included data from all 10 embryos). The data are downloaded as a cell-gene matrix in CEF format (GSE76381_EmbryoMoleculeCounts.cef.txt.gz) containing data processed as described in the publication¹³. We have added information about this under URLs:

"La Manno et al. scRNA-seq data (GSE76381_EmbryoMoleculeCounts.cef.txt.gz) was obtained from: <https://www.ncbi.nlm.nih.gov/geo/query/acc.cgi?acc=GSE76381>"

Additionally, we have added information, under URLs about the version of the Jerber data sets (we used the latest version 3):

“Jerber et al. scRNA-seq datasets were obtained (version v310.5281/zenodo.4651413) from: <https://zenodo.org/record/4651413#.ZAcxXbMJEZ>”

We would also like to apologize for the missing information about where to download the Silette data. This has now been added to URLs:

“Silletti et al. snRNA-seq data were obtained from the CZ CELLxGENE platform: <https://datasets.cellxgene.cziscience.com/f9ecb4ba-b033-4a93-b794-05e262dc1f59.h5ad>”

Furthermore, we have specified, that the data used are the publicly available data processed by the original authors of the published papers, on page 37:

“We used processed sc/sn RNA-seq count data and cell-type annotations generated as described in the two papers^{13,14}. For study I⁹ the data was downloaded as a Collapsible Expression Format (CEF) file and converted into a Hierarchical Data Format version 5 Annotated Data (h5ad) file format using the python package AnnData (see URLs), while h5ad file formats were available for study II¹⁰.”

And on page 38:

“We used a h5ad file with processed data (see URLs) and cell cluster definitions reported in [Ref¹⁵].”

We have added information about the python package “AnnData” to URLs on page 46:

“Python package AnnData: <https://anndata.readthedocs.io/en/latest/generated/anndata.AnnData.html>”

Comment #3

-this was an error on my part, the authors are correct that they did not identify DEGs in their study.

-As an expansive GWAS the current study fits the brief.

-the title alluding to an implication of neuron biology would be strengthened by some orthogonal experimentation. Perhaps this could be edited given this is not a result, but more an interpretation and hypothesis given the sequencing data/analysis.

Response

We would like to emphasize that the wording of the title is based on several lines of evidence, primarily from the presented results, pointing to neuron biology:

- 1) The three identified risk genes have been found to be expressed in both excitatory and inhibitory neurons (<https://www.proteinatlas.org/>) and a more

detailed role of ANK2 and MAP1A in neuronal development and function of neurons have been reported in [REF]^{16,17}.

- 2) The top 20 genes (P -value $< 1 \times 10^{-3}$) are significantly enriched among genes expressed the initial segment of the axon
- 3) The scDRS analyses points to several neuronal cell types including GABAergic and dopaminergic neurons in independent data sets.
- 4) The PPI-network enrichment analyses have now been expanded further as outlined below

We have now added additional analyses to provide further evidence that the rare variant risk genes affect neuronal biology. We have tested for enrichment of the top 20 rare-variant risk genes ($P < 1 \times 10^{-3}$) and the three PPI-network genes among genes involved in synapse functions using SynGO¹⁸. We identified strong enrichment of genes in all three PPI-networks among genes involved in a range of synaptic functions (see new results added to Supplementary Table 11). We have added the following to the results section page 11:

“For all three PPI-network gene sets a high proportion of the genes mapped to genes with synaptic annotations in SynGO¹⁸ (MAP1A 52.7%; ANK2 44.30%; ANO8 57.14%) and demonstrated significant enrichment among genes involved in several synaptic processes (Supplementary Table 11). Especially the MAP1A PPI-network genes demonstrated a specific enrichment signal among genes involved in synaptic ribosome processes; out of the 184 MAP1A PPI-network genes, 37 and 49 genes mapped to pre- and post-synaptic functions respectively ($P_{\text{presynaptic ribosome}} = 3.02 \times 10^{-51}$; $P_{\text{postsynaptic ribosome}} = 6.94 \times 10^{-68}$).”

We have added the following to the results section on page 11:

“Five of the 20 genes were mapped to synaptic functions, but no significant enrichments were observed for synaptic functions (Supplementary Table 11).”

We have added the following to the discussion section, page 21:

“Enrichment pointed towards several biological mechanisms or cell components that potentially could be affected in individuals with ADHD, including both pre- and post-synapse functions (all three networks),...”

We have added the following to the methods section regarding SynGO enrichment analyses, page 35:

“(2) genes related to synapse function using synaptic annotations based on published, expert-curated evidence for 1602 genes in SynGO¹⁸ (version 20231201 release 1.2), the list of brain expressed genes provided by SynGO was used as background”

“All the enrichment analyses were done using a one-sided Fisher’s exact test and the within database correction for multiple testing was done using the Benjamini-Hochberg method for correction for multiple testing.”

The SynGO analyses provide strong support for involvement of the PPI-network genes in synapse function. There was limited power to detect any enrichment signals for the

top associated genes due to the small number of genes (20 genes), but five of the genes were mapped to synaptic functions and we identified a within database significant enrichment among genes involved in post- and pre-synaptic processes ($P_{\text{presynaptic ribosome}} = 3.98 \times 10^{-2}$; $P_{\text{postsynaptic ribosome}} = 2.72 \times 10^{-3}$, Supplementary Table 11), however this was no longer significant when we corrected for all 13 databases that were used in the enrichment analyses, and thus not discussed on the manuscript.

Overall, the SynGO results add to the evidence suggesting that rare variant risk genes contribute to ADHD risk by impacting neuronal biology. We believe the current title accurately reflects our findings and would prefer to retain it as is. However, if the reviewer and editor consider the term 'implicate' to be too strong, we are open to adjusting it.

Comment #5

-edits to Figure 4 to improve clarity are appreciated and address the original concern.

Response

We are pleased that the steps we took to improve the figure have resolved the concerns.

Comment #6

-additions to statistical testing throughout, in particular Fig 4 multiple testing is appreciated
-improvements and inclusion in methods, in particular for proteomics section is appreciated

Response

We are glad to hear that the concerns have been clarified.

Comment #8

-Supplementary Figure 1 a nice addition.
-Some concern still about a lack of details on sex - important also for the reanalyzed data (e.g. La Manno) which contains human samples of mixed sex - I could not find any details about which samples were used for the reanalysis. Does this matter? Given the authors comment throughout the manuscript about sex-specific responses suggests it does (?), but this is unclear.

Response

Thank you for appreciating the new Supplementary Figure 1.

Regarding the details about the La Manno data, please see response to comment#2 above. as noted in our response to comment#2 above.

Regarding potential sex differences, we would like to reiterate that we did not observe any genetic differences between males and females with ADHD in our genetic analyses. We did not correct for sex in the scDRS analyses in the previous version of the manuscript, due to the following reasons:

La Manno et al.: The dataset includes data from 10 human embryos, but there is no available metadata on sex of these embryos, nor were sex differences analysed or reported in the original study.

Siletti et al.: This dataset consists of snRNA-seq data from four adult postmortem human brains (three males and one female). Given the small number of individuals meaningful conclusions about sex differences are not feasible, and such analyses were therefore not included in the original publication or in our study. We have included sex information for this data set on page 37 of the manuscript:

“This dataset consists of snRNA-seq data from neurons dissected from four adult postmortem human brains (three males and one female) from 105 locations across...”

Jerber et al.: The dataset includes iPSCs derived from a relatively large and sex-diverse cohort: 88 male and 127 female donors. We have now included this information on page 37:

“These data were generated from 215 pluripotent stem cell (iPSC) lines, each derived from a single healthy donor (88 males, 127 females).”

The original study by Jerber et al. did not include analyses of sex differences in gene expression between cells derived from male vs. female donors and there is no available information about sex of the samples in the publicly available data from the study. The iPSC lines analysed in that study were drawn from the Human Induced Pluripotent Stem Cells Initiative (HipSci), and a related study by Kilpinen et al. (*Nature, 2017*)¹⁹ has investigated gene expression variability across 711 HipSci iPSC lines, including 169 of the 215 lines analysed in Jerber et al. (and in our study). Kilpinen et al. found that donor effects accounted for the largest proportion of gene expression variance for 53.3% of genes, while sex accounted for the largest proportion of variance for only 1.9% of the genes. This suggests that sex has a minimal impact on gene expression variation in this context.

Nevertheless, we acknowledge that sex-based differences, even if subtle, may still contribute to variability. We have therefore merged information about sex of the donors obtained from HipSci with the downloaded data and redone the analyses of the Jerber data including sex as a covariate. The results remained practically unchanged. We have updated Supplementary Table 13 with the results from analyses including sex as a covariate and replaced the Figure 4.C with a new figure based on the updated results. Furthermore, we have added information about the sex-covariate to the methods section, page 39:

“Additionally, sex was used as a covariate in the analyses of data from study II. Information about sex of the donors was obtained from the Human Induced Pluripotent Stem Cell Initiative (HipSci) data browser (see URLs) and merged with the downloaded data.”

Comment #9

-Supplementary Figure 2 (original Supplementary Figure 1) does a nice job of

detailing how sample/datasets are integrated, but still lacks the standardization and preanalysis steps (e.g. ambient RNA cleanup in snRNAseq data as described above).

Response

Thank you for appreciating Supplementary Figure 2.

What the reviewer is commenting on, as we understand it, is a missing description of the generation and processing of the publicly available single cell sequencing data. We have added all details in the manuscript about the modifications and additional QC steps we have applied to the data. We believe that a detailed summary of the data generation and processing performed by the original authors of the publicly available datasets falls outside the scope of our study, as those methods are already extensively described in the respective publications.

To make it clearer that Supplementary Figure 2, represents an overview of the QC work done in this study we have modified the title to be more specific, and it now reads:

“Supplementary Figure 2. Overview of quality control of whole-exome sequencing data”

Comment #11

-I was unclear in the original comments, and I apologize for the ambiguity of the comment. The authors are correct that they have extensive methods descriptions for a lot of the manuscript, however the weakness remains about how the published datasets were collected, reanalyzed using modern methods (still missing), and integrated. This may seem like a small concern to the authors, but the onus is on the authors to update these historical datasets to ensure artifacts are not incorporated due to usage of errors (non egregious or planned) creeping in due to older computational pipelines.

Response

In our recent GWAS of ADHD we identified a significant association between common variant ADHD risk genes and genes with high expression in dopaminergic neurons in analyses of the La Manno data. This is why we included these (relatively old) data in this study, as we wanted to evaluate if rare-variant risk genes demonstrate the same pattern in the same data (explained on page 12). We are fully aware of the lower quality of the La Manno data, which was the reason why we included the newer data set from Jerber et al. We used donor ID as a covariate in our scDRS analyses, to account for inter-individual biological variability, potential technical differences linked to donor samples, and unequal donor representation among cells. However, this was not possible for the La Manno dataset, as donor ID information was not available in their data due to the way the data was generated. We have now made this limitation clear in the revised manuscript on page 13, with the following statement:

“The results could be influenced by variables which we were not able to correct for in this older data (see methods) and thus we validated our findings in a more recent, larger dataset”

An added the following to the methods section page 39:

“We included donor ID as a covariate in analyses of study II and III. Information on donor ID was not available for study I, and we were therefore not able to correct for the potential impact of factors captured by the donor ID covariate i.e., differences in biological variance between individuals, potential technical variation linked to donor samples, or unequal donor representation among cells.”

Based on the comment from the reviewer we revisited our analyses of the La Manno data to evaluate if there was something we could do to improve the analyses. The cells are collected from week 6 to 11, and we evaluated that it would be appropriate to include a covariate to correct for potential differences in gene expression caused by collection day. This only had smaller effects on the results; but it did change some of the significant findings so that the ADHD rare-variant risk genes now have a significant increased expression in dopaminergic neurons type 1 (we observed a nominal significant expression in the previous analysis). GABAergic neuroblasts were only nominal significant in the new analyses. The dopaminergic signal replicates what we previously found for common variants in the same dataset.

Supplementary Table 13 was updated with the new results, and we replaced Supplementary Figure 11 with a new version. We have updated the text on page 13 with the new results:

“.. we identified a significant association (i.e., increased scDRS across a cell types) between rare variant risk genes and dopaminergic neurons type 1 ($P = 9.99 \times 10^{-4}$), replicating what was found for common variant ADHD risk genes⁸. Furthermore, a significant association was observed for GABAergic neurons ($P = 9.99 \times 10^{-4}$), and medial neuroblasts ($P = 2 \times 10^{-3}$) (Supplementary Figure 11, Supplementary Table 13).”

In the discussion section page 21, we have removed the mentioning of GABAergic neuroblasts since this result was no longer significant after Bonferroni correction.

We have information about the covariate in the methods section page 39:

“Day of cell collection (collected from week 6 to 11) was also used as a covariate in analyses of data from study I.”

Regarding the two other data sets used in the scDRS analyses. We do not consider those to be “historical” as the data are from recent projects which were published in Nature Genetics in 2021 (Jerber et al.¹⁴) and in Science in 2023 (Siletti et al.¹⁵). The data have been generated and grouped into cell types by leading researchers in the field, and we prefer to rely on the expertise of the original authors on data processing rather than redoing this ourselves. This is in line with recent papers also using these data published in Nature²⁰, and Nature Communications²¹. Nevertheless, we have done additional QC on some of the data as outlined on page 37:

“For study II, the 11 days and 30 days downloaded data sets were used without any modifications, while the 52 days dataset was filtered to remove cells that were treated with rotenone (following what was done in the published study¹⁴). Additionally

the raw read counts from study I¹³ and the 52 days data from study II¹⁴ were filtered so only genes represented in a minimum 30 cells were included.”

Regarding the BrainSpan data we have done additional QC of these data by samples with low quality (RIN \leq 7) individuals, as explained in page 36.

Comment #12

-reviewing other reviewer comments and author responses, it is true that the GWAS analysis and much of the manuscript is well-prepared. There does remain concerns in the snRNAseq integration and analysis, and given the assertion in the title of a neuronal dysfunction, the lack of orthogonal validation, and the weaknesses in the cell-type specific snRNAseq data remains, making these cell type specific conclusions weak.

Response

As outlined above, there are several lines of evidence from our analyses pointing towards the involvement of neuronal biology in ADHD, not only the sc/snRNA analyses, and the inclusion of the new SynGO analyses, have provided further evidence (see answer to comment#3 above).

We think our results suggest that rare variant risk genes contribute to ADHD risk by impacting neuronal biology, and we would therefore prefer to retain the title as is. However, if the reviewer and editor consider the term 'implicate' to be too strong, we are open to adjusting it."

Comment #15

-I appreciate the authors statement in response that the current study provides an important dataset for hypothesis generation and follow-up functional studies. It achieves this in volumes, and the authors are to be applauded for this. My concern about the attribution of function/cell type specific biology remains, and given it is listen prominently in the title of the paper, these weaknesses are a major concern.

Response

We thank the reviewer for the positive assessment of the generated data and hope that we have addressed the reviewer's remaining concerns satisfactorily.

References

- 1 Widding-Havneraas, T. *et al.* Geographical variation in ADHD: do diagnoses reflect symptom levels? *European child & adolescent psychiatry* **32**, 1795-1803 (2023). <https://doi.org:10.1007/s00787-022-01996-7>
- 2 Cortese, S. *et al.* Incidence, prevalence, and global burden of ADHD from 1990 to 2019 across 204 countries: data, with critical re-analysis, from the Global Burden of Disease study. *Molecular psychiatry* **28**, 4823-4830 (2023). <https://doi.org:10.1038/s41380-023-02228-3>

- 3 Ayano, G., Demelash, S., Gizachew, Y., Tsegay, L. & Alati, R. The global prevalence of attention deficit hyperactivity disorder in children and adolescents: An umbrella review of meta-analyses. *Journal of affective disorders* **339**, 860-866 (2023). <https://doi.org:10.1016/j.jad.2023.07.071>
- 4 Bybjerg-Grauholm, J. *et al.* The iPSYCH2015 Case-Cohort sample: updated directions for unravelling genetic and environmental architectures of severe mental disorders. *medRxiv* (2020).
- 5 Grontved, S. *et al.* Prevalence and Incidence of Attention Deficit/Hyperactivity Disorder in Denmark. A National Register-Based Open Cohort Study. *Acta Psychiatr Scand* (2025). <https://doi.org:10.1111/acps.13804>
- 6 Xu, G., Strathearn, L., Liu, B., Yang, B. & Bao, W. Twenty-Year Trends in Diagnosed Attention-Deficit/Hyperactivity Disorder Among US Children and Adolescents, 1997-2016. *JAMA Netw Open* **1**, e181471 (2018). <https://doi.org:10.1001/jamanetworkopen.2018.1471>
- 7 Polyzoi, M., Ahnemark, E., Medin, E. & Ginsberg, Y. Estimated prevalence and incidence of diagnosed ADHD and health care utilization in adults in Sweden - a longitudinal population-based register study. *Neuropsychiatr Dis Treat* **14**, 1149-1161 (2018). <https://doi.org:10.2147/NDT.S155838>
- 8 Demontis, D. *et al.* Genome-wide analyses of ADHD identify 27 risk loci, refine the genetic architecture and implicate several cognitive domains. *Nature genetics* (2023). <https://doi.org:10.1038/s41588-022-01285-8>
- 9 Pintacuda, G. *et al.* A foundational neuronal protein network model unifying multimodal genetic, transcriptional, and proteomic perturbations in schizophrenia. *medRxiv* (2025). <https://doi.org:10.1101/2025.05.02.25326757>
- 10 Zheng, G. X. *et al.* Massively parallel digital transcriptional profiling of single cells. *Nature communications* **8**, 14049 (2017). <https://doi.org:10.1038/ncomms14049>
- 11 Hao, Y. *et al.* Dictionary learning for integrative, multimodal and scalable single-cell analysis. *Nat Biotechnol* **42**, 293-304 (2024). <https://doi.org:10.1038/s41587-023-01767-y>
- 12 Andreatta, M. *et al.* Semi-supervised integration of single-cell transcriptomics data. *Nature communications* **15**, 872 (2024). <https://doi.org:10.1038/s41467-024-45240-z>
- 13 La Manno, G. *et al.* Molecular Diversity of Midbrain Development in Mouse, Human, and Stem Cells. *Cell* **167**, 566-580 e519 (2016). <https://doi.org:10.1016/j.cell.2016.09.027>
- 14 Jerber, J. *et al.* Population-scale single-cell RNA-seq profiling across dopaminergic neuron differentiation. *Nature genetics* **53**, 304-312 (2021). <https://doi.org:10.1038/s41588-021-00801-6>
- 15 Siletti, K. *et al.* Transcriptomic diversity of cell types across the adult human brain. *Science* **382**, eadd7046 (2023). <https://doi.org:10.1126/science.add7046>
- 16 Jin, X. *et al.* In vivo Perturb-Seq reveals neuronal and glial abnormalities associated with autism risk genes. *Science* **370** (2020). <https://doi.org:10.1126/science.aaz6063>
- 17 Liu, Y., Lee, J. W. & Ackerman, S. L. Mutations in the microtubule-associated protein 1A (Map1a) gene cause Purkinje cell degeneration. *The Journal of neuroscience : the official journal of the Society for Neuroscience* **35**, 4587-4598 (2015). <https://doi.org:10.1523/JNEUROSCI.2757-14.2015>

- 18 Koopmans, F. *et al.* SynGO: An Evidence-Based, Expert-Curated Knowledge Base for the Synapse. *Neuron* **103**, 217-234 e214 (2019). <https://doi.org:10.1016/j.neuron.2019.05.002>
- 19 Kilpinen, H. *et al.* Common genetic variation drives molecular heterogeneity in human iPSCs. *Nature* **546**, 370-375 (2017). <https://doi.org:10.1038/nature22403>
- 20 O'Connell, K. S. *et al.* Genomics yields biological and phenotypic insights into bipolar disorder. *Nature* **639**, 968-975 (2025). <https://doi.org:10.1038/s41586-024-08468-9>
- 21 Yao, S. *et al.* Connecting genomic results for psychiatric disorders to human brain cell types and regions reveals convergence with functional connectivity. *Nature communications* **16**, 395 (2025). <https://doi.org:10.1038/s41467-024-55611-1>